# A robust and interpretable machine learning approach using multimodal biological data to predict future pathological tau accumulation

Joseph Giorgio[1], William J. Jagust [2,3], Suzanne Baker [3], Susan M. Landau[2], Peter Tino[4], Zoe Kourtzi [1✉] & Alzheimer's Disease Neuroimaging Initiative*

The early stages of Alzheimer's disease (AD) involve interactions between multiple pathophysiological processes. Although these processes are well studied, we still lack robust tools to predict individualised trajectories of disease progression. Here, we employ a robust and interpretable machine learning approach to combine multimodal biological data and predict future pathological tau accumulation. In particular, we use machine learning to quantify interactions between key pathological markers (β-amyloid, medial temporal lobe atrophy, tau and APOE 4) at mildly impaired and asymptomatic stages of AD. Using baseline non-tau markers we derive a prognostic index that: (a) stratifies patients based on future pathological tau accumulation, (b) predicts individualised regional future rate of tau accumulation, and (c) translates predictions from deep phenotyping patient cohorts to cognitively normal individuals. Our results propose a robust approach for fine scale stratification and prognostication with translation impact for clinical trial design targeting the earliest stages of AD.

[1] Department of Psychology, University of Cambridge, Cambridge, UK. [2] Helen Wills Neuroscience Institute, University of California, Berkeley, CA, USA. [3] Molecular Biophysics & Integrated Bioimaging, Lawrence Berkeley National Laboratory, Berkeley, CA, USA. [4] School of Computer Science, University of Birmingham, Birmingham, UK. *A list of authors and their affiliations appears at the end of the paper. ✉email: zk240@cam.ac.uk

Alzheimer's Disease (AD) develops gradually with multiple pathophysiological events occurring well before clinical manifestations[1]. The proposed sequence of events begins with the deposition of β-amyloid (Aβ) which promotes widespread pathological tau protein accumulation that in turn leads to neurodegeneration and cognitive impairment[2]. Quantifying these interactions is critical for establishing a mechanistic account of the events that lead to initiation and progression of AD.

The availability of PET imaging of Aβ plaques and neurofibrillary tau pathology in the brain, has enabled the detection of the core features of the neuropathology of AD in-vivo, and largely supported the proposed sequence of events from Aβ through tau and neurodegeneration (for reviews:[3–5]). In light of these developments, the staging of AD has shifted from a clinical syndromic diagnosis requiring pathological verification post-mortem[6,7] to a disorder reflecting a continuum of biomarker characteristics[8]. The clinical syndromic classification framework defined AD as the transition from a cognitively unimpaired state (i.e., cognitively normal: CN) to a state of mild cognitive impairment (MCI) to AD dementia[9]. However, these clinical syndromic definitions are neither specific[10,11] nor sensitive[12,13] to the underlying pathology of AD. In contrast, the recently proposed NIA-AA 2018 biological framework of AD offers an objective classification framework using biomarkers taken at a single time point[8].

To classify individuals as AD within this biological framework, continuous biomarkers are categorically assigned as either positive or negative. An individual with positive biomarkers for Aβ and tau is defined as having AD and positive biomarkers for neurodegeneration are used to provide information about disease stage[8]. To dichotomise Aβ biomarkers large AD cohort studies have shown that Aβ PET is distributed bimodally[14] with tracer and study specific probabilistic thresholds derived[15]. However, these single threshold values do not account for individuals who are below thresholds of β-amyloid positivity but may nevertheless follow subsequent AD trajectories[16,17], limiting the sensitivity of β-amyloid alone as a predictor of future biomarker changes. Further, spatiotemporal patterns of tau are shown to be strongly linked to both future neurodegeneration and cognitive decline[18]. A recent study showed four distinct spatiotemporal profiles of tau burden in predominantly symptomatic AD[19], proposing clinically meaningful topographies of tau burden. Further evidence in early AD (i.e., asymptomatic and mildly impaired) cohorts suggests converging patterns of primary tau seeding (measured in-vivo by longitudinal PET)[20–23]. These studies suggest that tau initially accumulates within the medial temporal cortex then spreads to the inferolateral temporal lobe and superior and medial regions of the parietal cortex prior to severe cognitive impairment[20–23]. Thus, slowing rates of tau accumulation within these primary regions could serve as a biomarker outcome of interest for clinical trials that typically use change in cognition as primary outcome. In line with these findings, a recent trial of an amyloid-lowering therapy used tau levels to select participants and rate of change in regional tau accumulation as a secondary trial outcome[24]. However, clinical trials remain hindered by sample heterogeneity that reduces sensitivity in assessing treatment outcome[25]. Thus, innovative modelling approaches are needed to integrate continuous topographic patterns of Aβ, tau and neurodegeneration to accurately stratify patients for trial inclusion, with potential to reduce sample heterogeneity and increase trial efficacy.

Recent advances in machine learning allow us to develop predictive models of neurodegenerative disease that mine multimodal datasets including measurements of clinical syndrome, cognition and neuropathology from large patient cohorts[26]. To date, most machine learning models in AD utilise information from rich longitudinal cohorts that were initiated under the framework of clinical syndromic definitions[27]. These studies have primarily focused on discrete changes in syndromic diagnosis[28] and probabilistic estimates of time to conversion to AD based on longitudinal changes in clinical labels[29–36]. Modelling approaches that utilise both longitudinal syndromic labels and continuous biomarker information have strong potential to improve prediction of longitudinal biological processes in AD, bridging the gap between biological and syndromic frameworks.

Here, we employ a trajectory modelling approach based on machine learning[37] to quantify the multivariate relationships between key biomarkers (Aβ, tau, medial temporal atrophy)—in line with the biological framework of AD—and APOE 4 (4 allele of the Apolipoprotein E gene) genotype, the major genetic risk factor for late onset AD[38]. We train our model using discrete longitudinal changes in syndromic definitions to separate individuals who are Clinically Stable from those who are in the earliest stages of AD (i.e., CN or MCI at baseline but Clinically Declining). Extending beyond this binary classification, we derive a continuous prognostic index that quantifies the distance of an individual from the Clinically Stable prototype. We test whether this trajectory modelling approach predicts longitudinal change in cognition and biomarkers (i.e., future tau accumulation) based on baseline non-tau data (Aβ, medial temporal atrophy, APOE 4) over the short timeframes that are typical of AD clinical trials (i.e., 1–3 years) (Fig. 1). We demonstrate that our prognostic index: (a) predicts individualised rates of future tau accumulation even before symptoms occur, (b) re-stratifies populations at greatest risk of accumulating tau in the future. We validate our prognostic index not only in a sample of CN and MCI patients that was not included in constructing the index (i.e. training the model), but also on an independent data set from asymptomatic individuals. Finally, we demonstrate that our prognostic index: (a) is more sensitive compared to baseline syndromic diagnosis and Aβ positivity, (b) reduces the sample size required to determine future tau accumulation, suggesting strong potential of our trajectory modelling approach for application in the design of clinical trials.

## Results

**Deriving a prognostic index from multimodal biomarkers**. We used a trajectory modelling approach based on the Generalised Matrix Learning Vector Quantisation (GMLVQ) machine learning framework (GMLVQ-scalar projection[37]) to generate a prognostic index as a single numerical descriptor (scalar projection) from three biological markers measured at baseline: cortical Aβ measured using PET, medial temporal grey matter density measured using T1-weighted MRI and APOE 4 genotype (presence of one or two alleles). This approach derives a continuous prognostic index by training a machine learning model with classes determined by longitudinal syndromic labels which are assigned independent of baseline biomarker status.

We trained a GMLVQ model on baseline (defined as the date of Aβ PET scan) data from the Alzheimer's Disease Neuroimaging Initiative (ADNI) 2/GO cohort. We determined two classes for training the algorithm defined by baseline and longitudinal syndromic labels independent of biomarker status: (a) Clinically Stable ($n = 100$): CN individuals who remain stable for 4+ years following baseline, (b) Clinically Declining ($n = 156$): individuals with unstable diagnosis (i.e., those with MCI or CN diagnosis at baseline who are subsequently diagnosed with dementia or those who had reverted from a diagnosis of dementia prior to baseline to MCI at baseline). An additional 181 individuals diagnosed with AD (which we refer to as Alzheimer's Clinical Syndrome) were included to cover the full spectrum of longitudinal AD diagnoses. These individuals received a stable diagnosis of AD dementia across follow-ups and were used as a reference population of late-

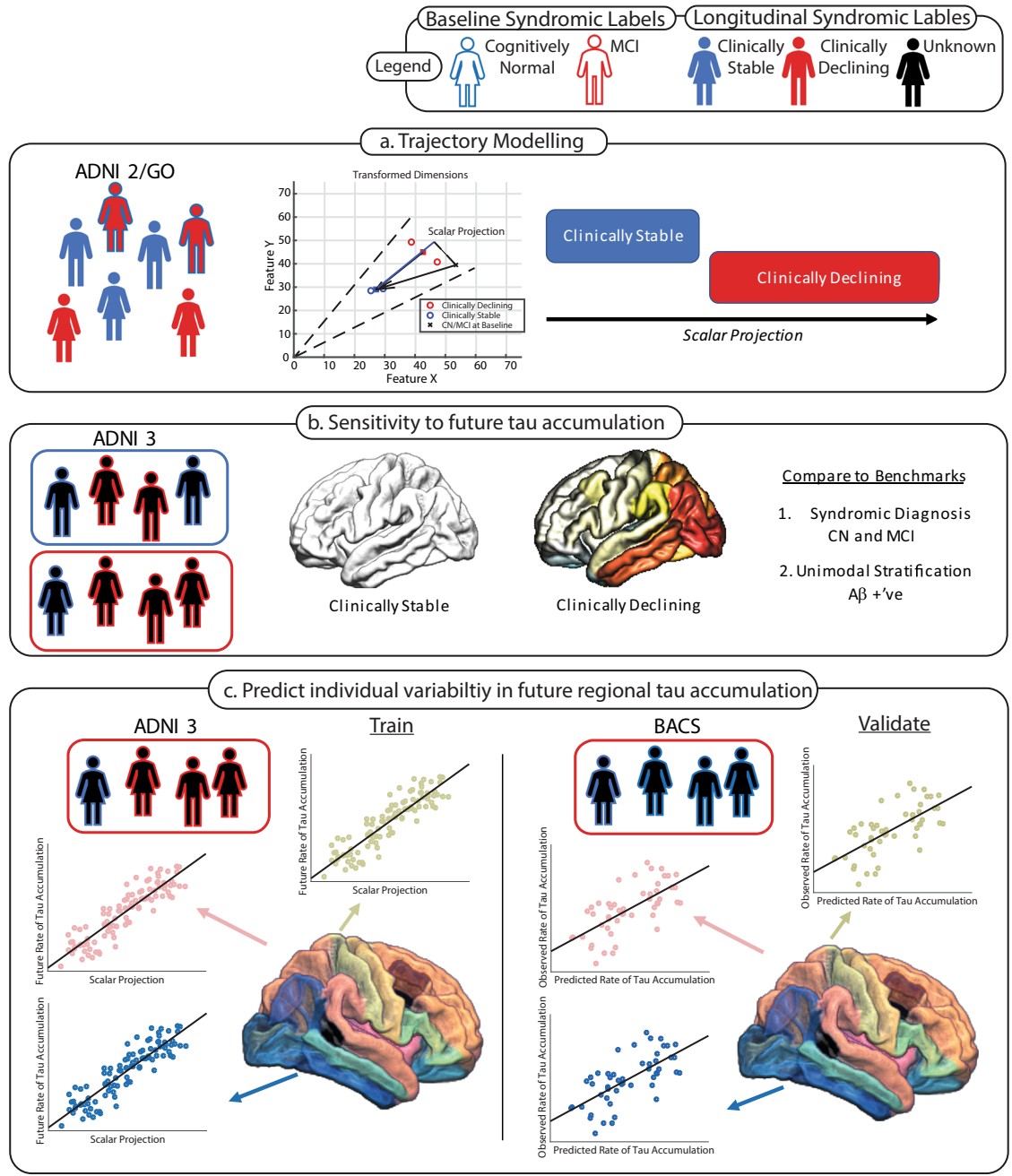

**Fig. 1 Study Workflow.** Workflow and analyses performed to generate, benchmark and validate predictions of future tau accumulation. The fill colour of the schematic cohort participant indicates the longitudinal syndromic label assigned; where: red filled individuals are Clinically Declining, blue filled individuals are Clinically Stable and black filled individuals denote unknown longitudinal diagnoses; these individuals form a test sample for the algorithm. The outline of the schematic cohort participant indicates the baseline syndromic label; where: red outlines are MCI at baseline and blue outlines are cognitively normal at baseline. **a** Training of the GMLVQ-scalar projection model on ADNI2/GO participants based on longitudinal syndromic definitions. The centre figure is a cartoon example indicating how the scalar projection is derived to separate Clinically Stable vs. Clinically Declining (*Supplementary Methods*). **b** ADNI 3 individuals are classified as either Clinically Stable (blue box) or Clinically Declining (red box); using this stratification regional future rate of tau accumulation is compared between the two groups. Two benchmark classifications are used for comparison: 1. Syndromic definition (i.e., CN vs. MCI), 2. Stratification using Aβ status. **c** Final modelling stage using the continuous scalar projection to predict individualised rates of regional future tau accumulation for individuals classified as Clinically Declining (red box). Regression models are learnt to predict future rate of FTP-PET accumulation for each ROI using the ADNI3 sample. These equations are then used to predict future rate of tau accumulation within each ROI for the cognitively normal BACS sample.

stage AD. Comparing the unimodal distributions of the Clinically Stable and Clinically Declining groups shows that the Clinically Declining group has greater AD pathology than the Clinically Stable group across the three biological predictors (Supplementary Figure 1).

We trained our GMLVQ-scalar projection model to learn the multivariate relationship between the three baseline biological predictors (metric tensor) and the location in multidimensional space (prototype) that best discriminates between Clinically Stable and Clinically Declining individuals. We then determined

the distance of each individual from the Clinically Stable prototype along the axis that is predictive of future diagnosis (i.e., prognostic axis from Clinically Stable towards Clinically Declining).

Using random resampling to split the ADNI2/GO sample into training and test sets we demonstrated that our model classifies Clinically Stable vs. Clinically Declining individuals with cross-validated class-balanced accuracy of 88% (sensitivity 87%, specificity 91%). Further, using logistic regression we showed that an individual is less than 50% likely to be classified as Clinically Stable if they have a scalar projection greater than 0.34 (i.e., individuals are more than 50% likely to be classified as Clinically Declining or Alzheimer's Clinical Syndrome). Comparing the scalar projection to the three biological markers showed that our multimodal prognostic index captures predictive variance in each of the unimodal predictors (scalar projection vs. A$\beta$: $R^2 = 82.8\%$, $p < 0.0001$; scalar projection vs. grey matter density: $R^2 = 41\%$, $p < 0.0001$; scalar projection (APOE 4 - /+): t(254)8.5, $p < 0.0001$) (Fig. 2).

Next, we used the model trained on ADNI2/GO data to derive the scalar projection for individuals from two independent cohorts who only have baseline syndromic assessment available: (a) ADNI3 (CN = 72, MCI = 43) (b) Berkeley Aging Cohort Study (BACS) (CN = 56). Participants who were enroled in ADNI3 as roll overs from ADNI2/GO were not used in the initial training of the machine learning model (i.e., Fig. 1a) and are unique to the ADNI3 sample (i.e., Fig. 1b, c). The scalar projection classified 59 ADNI3 participants and 39 BACS participants as Clinically Stable, with 56 ADNI3 and 17 BACS participants classified as Clinically Declining (i.e., scalar projection greater than 0.34) (Fig. 3). This scalar projection was not significantly related to education (BACS: $r(54) = -0.12$, $p = 0.37$, ADNI3: $r(113) = -0.009$, $p = 0.93$) nor sex (BACS: $t(54) = -1.72$, $p = 0.09$, ADNI3: t(113) = −1.13, $p = 0.26$) and showed a weak effect size when correlated to baseline age in BACS ($r(54) = 0.33$, $p = 0.01$) and ADNI3 ($r(113) = 0.36$, $p = 0.0005$).

Further, we tested whether difference in MRI field strength for the BACS sample introduced a systematic bias to the multimodal scalar projection. A two-sample t-test comparing the residual of the fit of the medial temporal grey matter density and the scalar projection showed no significant differences between 1.5 T and 3 T MRI in BACS ($t(54) = -1.79$, $p = 0.08$) (Supplementary Fig. 2), suggesting that our multimodal approach is robust across differences in MRI acquisition.

Finally, for the ADNI3 sample, we tested whether the classification based on the scalar projection derived from baseline multimodal biomarkers is similar to baseline syndromic clinical diagnosis. Comparing the classification of Clinically Stable vs. CN, and Clinically Declining vs. MCI (Fig. 3) showed that agreement was not significantly different from chance (CN($n = 72$): Clinically Stable $n = 40$; Clinically Declining $n = 32$, MCI($n = 43$): Clinically Stable $n = 19$; Clinically Declining $n = 24$, Cohens kappa $\kappa = 0.09$ [−0.0953, 0.274] $p = 0.38$). That is, the clinician-based syndromic diagnosis and the biomarker-derived multimodal scalar projection have poor agreement for differentiation of Clinically Stable and Clinically Declining, suggesting that cross-sectional syndromic definitions of AD are limited in capturing the underlying pathobiology that is predictive of clinical decline.

**Clinically Declining individuals accumulate tau more rapidly than Clinically Stable individuals.** We investigated whether the classification of Clinically Stable vs. Clinically Declining based on baseline biomarkers relates to baseline and future changes in tau

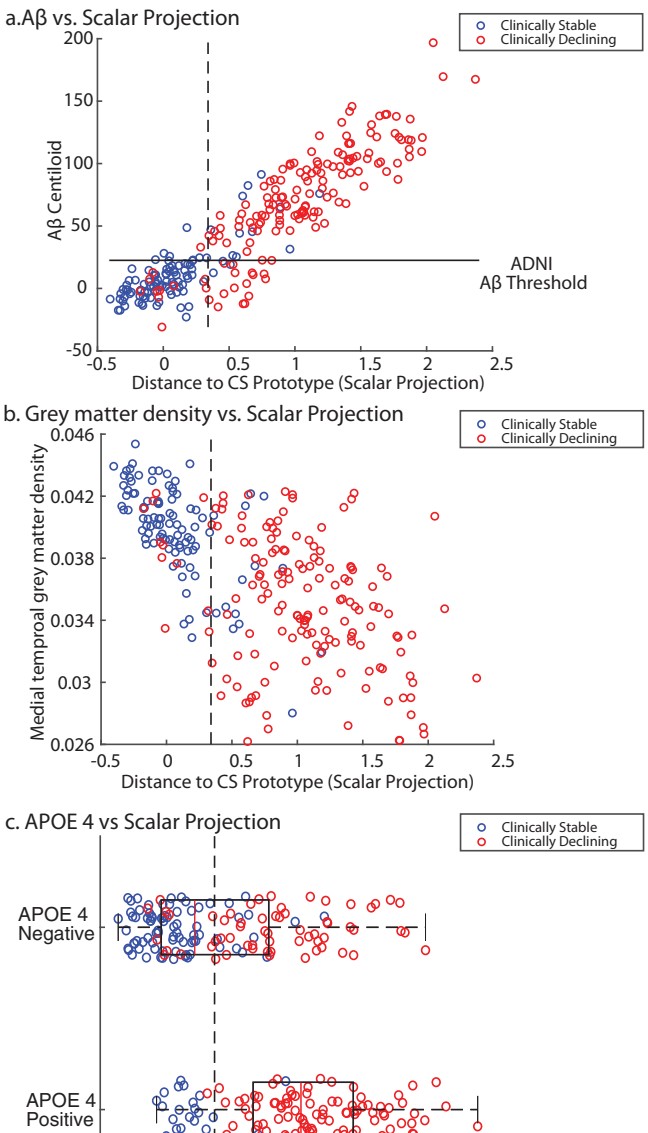

**Fig. 2 Relationship of scalar projection with biological predictors.** ADNI2/GO sample: blue dots indicate individuals in the Clinically Stable group, red dots indicate individuals in the Clinically Declining group. Dashed lines indicate the learnt probabilistic boundary that separates Clinically Stable from Clinically Declining. The scalar projection represents a scalar value from the projection of any sample point along the model-derived prognostic axis, where a scalar projection of 0 represents the median Clinically Stable individual and a value of 1 represents the median Clinically Declining individual. **a** Relationship of scalar projection with FBP centiloids (A$\beta$), the solid horizontal line indicates the ADNI threshold for A$\beta$ positivity (SUVR = 1.11). **b** Relationship of scalar projection with medial temporal grey matter density. **c** Relationship of scalar projection with APOE 4 status (n APOE 4 positive = 116, n APOE 4 negative = 140), the red line is the median of the group, the solid black box represents the 25th to 75th percentile and the dashed black lines represent the range of the data. Source data are provided as a Source Data file.

accumulation. We used [18F]-flourtaucipir PET (FTP-PET) from the ADNI3 sample (Fig. 1b) to extract regional baseline and longitudinal Standardised Uptake Value Ratios (SUVR) values from 36 Desikan–Killiany ROIs. ROIs were grouped together in

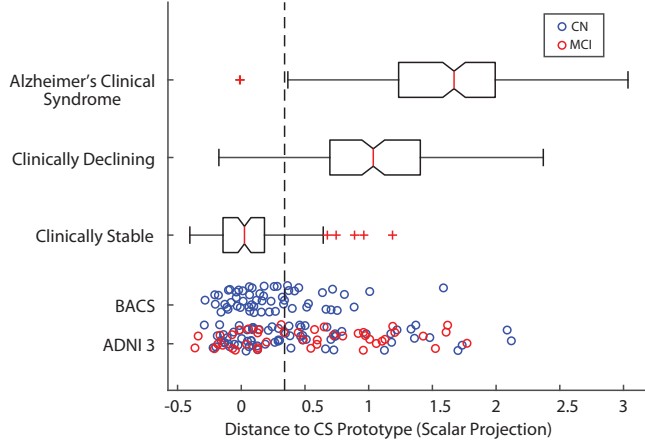

**Fig. 3 Distribution of the scalar projection for the ADNI2/GO, ADNI3 and BACS samples.** The distribution of the scalar projection for individuals from ADNI2/GO in the Clinically Stable ($n = 100$), Clinically Declining ($n = 156$) and Alzheimer's clinical syndrome ($n = 181$) groups (APOE 4 positive: $n = 116$, APOE 4 negative $n = 140$), the red line is the median of the group, the solid black box represents the 25th to 75th percentile and the black horizontal lines represent the range of the data, red crosses are outliers from the distribution, non-overlapping notches indicate significantly different medians ($p < 0.05$). Blue dots indicate individuals from ADNI3/BACS who are cognitively normal (CN) at baseline. Red dots indicate individuals from ADNI3 who have Mild Cognitive Impairment (MCI) at baseline. The dashed vertical black line represents the probabilistic boundary used to classify Clinically Declining vs. Clinically Stable, all individuals to the right of the line are classified as Clinically Declining. Source data are provided as a Source Data file.

order to approximate the topographical distribution of tau in the Braak Staging scheme, as previously described[39]. First, we contrasted baseline tau for Clinically Stable vs. Clinically Declining samples (Supplementary Results: Differences in baseline tau burden Clinically Stable vs. Clinically Declining). These analyses show that the sample classified as Clinically Declining has significantly greater baseline tau than the Clinically Stable sample across the cortex (Braak I mean difference = 0.167 SUVR, $t(113) = 5.6$, $p < 0.001$; Braak II mean difference = 0.074 SUVR, $t(113) = 2.6$, $p = 0.01$; Braak III mean difference = 0.12 SUVR, $t(113) = 4.77$, $p < 0.001$; Braak IV mean difference = 0.08 SUVR, $t(113) = 3.81$, $p < 0.001$; Braak V mean difference = 0.07 SUVR, $t(113) = 3.47$, $p < 0.001$; Braak VI mean difference = 0.041 SUVR, $t(113) = 2.53$, $p = 0.012$) (Supplementary Fig. 3, Supplementary Table 1). This pattern of greater baseline tau for the Clinically Declining sample was also observed in the BACS data set (Supplementary Fig. 4).

Next, we extracted longitudinal rates of regional tau accumulation from the ADNI3 sample within each of the 36 Desikan–Killiany ROIs. For Clinically Stable and Clinically Declining groups we then contrasted the global rate of tau accumulation for Clinically Stable vs. Clinically Declining (i.e., independent samples t-test across ROIs for Clinically Stable vs. Clinically Declining). We observed a significant difference in global tau accumulation when comparing Clinically Stable vs. Clinically Declining ($t(70) = 2.23$, $p = 0.029$), with the Clinically Declining group accumulating global cortical tau 2.3 times faster than the Clinically Stable group (Supplementary Table 2). We observed that there is high specificity to regional tau accumulation using the classification based on the multimodal scalar projection with a near zero interclass correlation coefficient across ROIs ($r = 0.036$ [−0.29 0.36], $F(35,36) = 1.074$ $p = 0.42$). Further, we tested which regions significantly accumulated tau (i.e., rate of

accumulation significantly greater than 0; one sample (i.e., Clinically Stable or Clinically Declining) one tail t-tests within each ROI). We showed that Clinically Declining individuals accumulate tau primarily in Braak stages 4 and 5 ROIs (Supplementary Table 2, Fig. 4). In contrast, the sample classified as Clinically Stable did not show significant future tau accumulation across any cortical regions (Fig. 4d).

We next investigated whether the classification of Clinically Stable vs. Clinically Declining is sensitive to future cognitive change (as measured by future annualised change in Preclinical Alzheimer's Cognitive Composite, i.e., PACC) over the same time period. We observed a significant difference in future cognition between the sample classified as Clinically Stable(mean = 0.13/year) vs. Clinically Declining(mean = −0.86/year) ($t(100) = −2.48$, $p = 0.015$), with the Clinically Declining sample showing significant worsening (i.e., rate of PACC change significantly less than 0) in future cognitive ability (one tail t-test $t(50) = −2.65$, $p = 0.0054$). Taken together, our results show that our modelling approach based on baseline multimodal data is sensitive and specific to baseline tau burden, changes in future tau accumulation and cognitive decline in an independent sample without longitudinal syndromic information (i.e., ADNI3).

Next, we compared a baseline syndromic classification (i.e., CN vs. MCI) to future changes in tau accumulation. We observed a difference in global tau accumulation between CN and MCI groups ($t(70) = 2$, $p = 0.05$), with the MCI population accumulating global cortical tau 1.9 times faster than the CN population. To determine if a classification based on syndromic labels (i.e., CN vs. MCI) is specific to future regional rate of tau accumulation, we calculated the interclass correlation coefficient of future regional rate of tau accumulation across ROIs. A significant interclass correlation coefficient across ROIs ($r = 0.47$ [0.159 0.68], $F(35,36) = 2.69$ $p = 0.002$) suggests poor specificity to regional tau accumulation for stratification based on baseline syndromic definitions. Further, we showed that both CN and MCI samples significantly accumulate tau, with a high degree of overlap across AD susceptible regions in the temporal and posteromedial cortices (Supplementary Table 3, Supplementary Fig. 5). To further quantify this, we calculated the interclass correlation coefficient between a baseline syndromic definition of CN vs. Clinically Declining, and, MCI vs. Clinically Declining. For both syndromic definitions, we observed significant overlap in future regional tau accumulation with Clinically Declining (CN vs. Clinically Declining $r = 0.584$ [0.323 0.763], $F(35,36) = 3.81$ $p < 0.0001$; MCI vs. Clinically Declining $r = 0.86$ [0.751 0.928], $F(35,36) = 13.7$ $p < 0.0001$). Taken together, our results suggest that stratification based on syndromic diagnosis has poorer sensitivity and specificity to future tau accumulation compared to the biological classification of Clinically Stable vs. Clinically Declining based on our prognostic index.

**Modelling comparisons.** We compared our model-derived predictions generated from our GMLVQ-scalar projection approach to alternate prediction frameworks.

First, comparing our GMLVQ binary classification of Clinically Stable vs. Clinically Declining to the standard linear Support Vector Machine (SVM) showed similar accuracy (mean accuracy GMLVQ: 88% SVM: 88%, $t(798) = 1.5$, $p = 0.13$) and 99.13% agreement in the predicted labels for the ADNI3 sample (Supplementary Results: GMLVQ vs. SVM classification). Thus, the SVM classifier corroborates our binary classification results using GMLVQ. We note that this binary classification task serves simply as a step in the derivation of our prognostic index that is at the core of our trajectory modelling approach. In particular, to derive this prognostic index, we need to work in a feature subspace that captures possible signatures of future tau

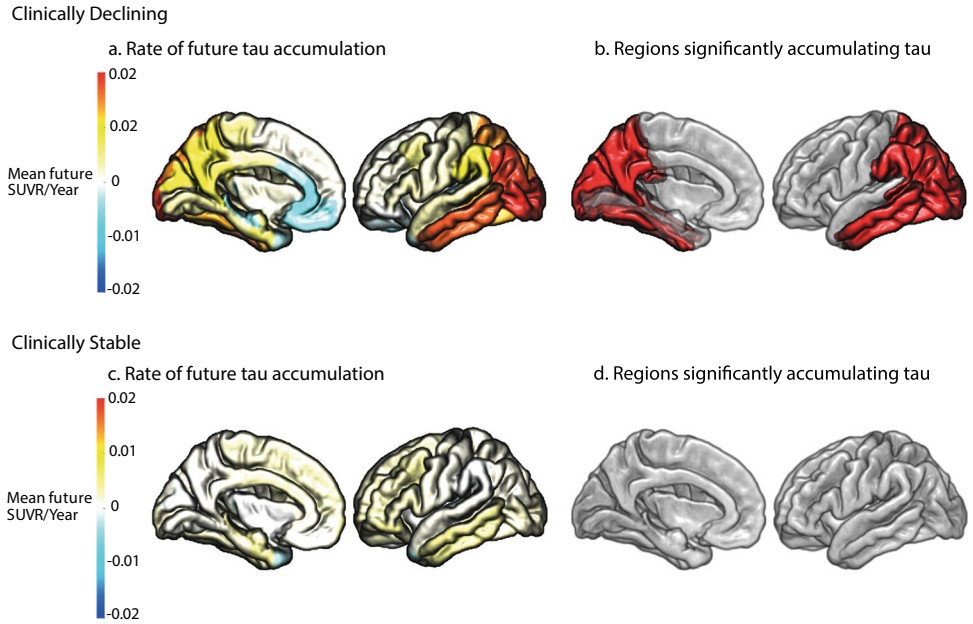

**Fig. 4 Regional future rate of tau accumulation.** Future annualised rate of tau accumulation for ADNI3 individuals across the 36 Desikan–Killiany ROIs. **a** Mean future annualised rate of tau accumulation for individuals classified as Clinically Declining. **b** The regions in red are significantly accumulating tau for individuals classified as Clinically Declining (one-sided t-test (right) against 0, $p < 0.05$ uncorrected). **c** Mean future annualised rate of tau accumulation for individuals classified as Clinically Stable. **d** The regions in red are significantly accumulating tau for individuals classified as Clinically Stable (one-sided t-test (right) against 0, $p < 0.05$ uncorrected). Source data are provided as a Source Data file.

accumulation. Any machine learning classifier that supports this subspace learning could be used. We chose to use GMLVQ over SVM because it naturally provides class prototypes and a subspace endowed with an appropriate metric (distance to the prototypes) that allows us to derive the prognostic index using our scalar projection method.

Second, we compared our trajectory modelling approach based on baseline biomarker data to latent time joint mixed effects models (LTJMM) that have been previously shown to infer disease stage based on longitudinal biomarker data[40]. We tested whether our scalar projection (Fig. 2) which incorporates only baseline pathological burden relates to disease stage (i.e., latent time shift) extracted from the LTJMM that is derived modelling longitudinal tau data. We observed a positive relationship between the LTJMM latent time shift and our prognostic index ($r(113) = 0.42$, $p < 0.0001$), suggesting the scalar projection derived using only baseline data relates to the LTJMM derived disease stage (Supplementary Results GMLVQ-scalar projection vs. LTJMM prediction). We note that LTJMM requires longitudinal data to model disease trajectories and fit individualised parameters, limiting out-of-sample generalisation. In contrast, our modelling approach derives these out-of-sample predictions form baseline data naturally, as we provide individualised indices and features rather than constructing individualised models.

**Fewer patients are required to detect meaningful change in tau accumulation than cognition.** To examine the clinical utility of regional longitudinal tau accumulation as an outcome measure, we contrasted the sample size required to observe change in cognitive decline vs. tau accumulation for the sample classified as Clinically Declining based on the scalar projection. We defined a clinically meaningful change as a 25% reduction in rate of change of either regional tau accumulation or PACC change. For the sample classified as Clinically Declining, we calculated that the required sample size (for a significance level of $p = 0.05$ at a power of $a = 0.8$) to detect a 25% reduction in rate of PACC

change is $n = 917$. However, an average sample size of $n = 637$ is required to detect a 25% reduction in regional future tau accumulation in selected regions (Fig. 4b) at the same power. Thus, using future rate of tau accumulation as a clinical outcome measure compared to future rate of cognitive decline delivers a reduction in required sample size of 31%.

**Using multimodal biomarkers reduces sample size to detect a meaningful change in tau accumulation.** Next, we compared prediction of longitudinal tau accumulation for the sample classified as Clinically Declining ($n = 56$) based on the scalar projection vs. a sample classified based on Aβ alone ($n = 61$ Aβ positive). We observed that the sample classified as Clinically Declining accumulated global cortical tau (i.e., mean rate of tau accumulation across the 36 Desikan–Killiany ROIs) 1.31 times faster than the sample defined only as Aβ positive (Supplementary Table 4).

We next tested for the sample size needed to detect a 25% decrease in rate of future tau accumulation (given significance level of $p = 0.05$ at power of $a = 0.8$) for Clinically Declining vs. Aβ positive samples within regions shown to significantly accumulate tau for individuals classified as Clinically Declining (Fig. 4b). We observed on average a 44% reduction in sample size when stratifying based on Clinically Declining classification ($n = 637$) vs. Aβ positive alone ($n = 1139$). We repeated these calculations using regions in which the Aβ positive sample was shown to significantly accumulate tau (Supplementary Table 4). This showed an average of 38% reduction in sample size when stratifying based on Clinically Declining classification ($n = 581$) vs. Aβ positive alone ($n = 935$) (Fig. 5). These results provide evidence for the clinical utility of stratification of Clinically Declining using our prognostic index derived from baseline multimodal data compared to Aβ status alone.

**Prognostic index predicts individual variability in trajectories of regional tau accumulation in asymptomatic and mildly impaired samples.** Using the ADNI3 sample we fit linear regression equations to test whether the continuous prognostic

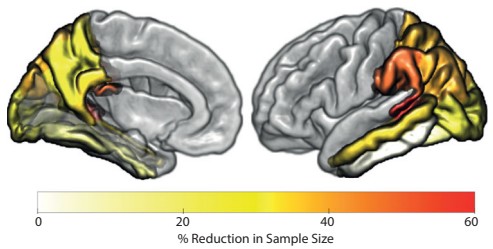

**Fig. 5 Reduction in sample size for patient stratification based on prognostic index vs. Aβ positive.** Percentage reduction of sample size required to observe a 25% reduction in tau accumulation per region for Clinically Declining classification based on multimodal data vs. Aβ status alone. For example, to observe a 25% reduction in tau accumulation in the middle temporal cortex 34% fewer individuals are required when stratifying based on Clinically Declining classification vs. Aβ status alone. In medial and superior-parietal regions (e.g., Precuneus or superior parietal) a sample size reduction of 29% and 39% is achieved when stratifying based on Clinically Declining vs. Aβ status alone. Source data are provided as a Source Data file.

index (derived from the model trained on ADNI2/GO) predicts individual variability in future tau accumulation. Of the 11 regions that were shown to significantly accumulate tau (Fig. 4b) 7 showed a significant relationship between the prognostic index and individual rates of future tau accumulation, explaining up to 21% of variance in the temporal lobe and medial regions of the posterior parietal cortex (Supplementary Table 5). Figure 6a shows an example case of the relationship between our prognostic index and rate of tau accumulation in Fusiform gyrus, a region that is known to be susceptible to early pathological tau deposition in AD.

**Predicting future tau accumulation based on cognitive data**. To investigate the predictive power of cognitive data we re-ran our classification experiments using data from multiple neuropsychiatric tests as input features (ADAS-Cog, MOCA Total, MMSE Total, RAVLT Total). This cognitive classification model reliably separated Clinically Stable vs. Clinically Declining (86% class-balanced accuracy). Further, using the cognitive scalar projection derived in ADNI3, we show that this prognostic index separates individuals who will accumulate tau in the future (Supplementary Results: Cognitive Classification Model, Supplementary Fig. 6). These results demonstrate that stratification based on future tau accumulation is possible using cognitive (non-biomarker) data. Next, we related the cognitive scalar projection to future regional tau accumulation within ROIs that were shown to significantly accumulate tau (Supplementary Fig. 6). We did not observe a significant relationship between the cognitive scalar projection and individual variability in future tau accumulation in these ROIs (Supplementary Results: Cognitive Classification Model). Thus, our trajectory modelling based on cognitive data separates individuals who will accumulate tau in the future; yet, it is less sensitive in predicting individual variability in future regional tau accumulation.

**Individual trajectories of pathological tau accumulation are accurately predicted in an independent asymptomatic sample.** We generated individualised predictions of future tau accumulation for CN individuals classified as Clinically Declining from the BACS sample. Using the model trained on ADNI2/GO individuals, we derived the scalar projection from baseline multimodal biological data in the BACS sample (Fig. 4). To predict the future rate of tau accumulation in this sample, we used the

linear regression models relating the scalar projection to rate of future tau accumulation derived from the 7 regions that showed significant relationships for the ADNI3 sample (Fig. 6, Supplementary Table 5). Comparing the predicted and real future rate of tau accumulation for BACS Clinically Declining individuals, we observed individualised predictions of future tau accumulation explain up to 41% of variance in the temporal cortex and 22% in regions of the posterior parietal cortex (Table 1 and Fig. 7).

Finally, we tested whether future tau accumulation can be predicted for BACS Clinically Stable individuals based on their scalar projection. No accurate prediction can be made for these individuals, with predicted regional future tau accumulation accounting only for 2.8% on average of the observed variance in future tau accumulation (Table 1). These results suggest that our predictions are robust and specific for staging individuals who are in the asymptomatic phase of AD.

**Potential application in clinical trial design**. Our findings propose that our modelling approach has strong potential for application in clinical trial design. First, we reliably stratify individuals into two samples based on baseline non-tau data: (a) Clinically Stable, with low baseline tau, stable cognition and non-significant tau accumulation, (b) Clinically Declining with high baseline tau, declining cognition and significant future tau accumulation. We propose that the Clinically Declining sample is a good candidate for clinical trials, while individuals in the Clinically Stable sample are not suitable for AD clinical trials as they do not show baseline AD pathology or predictable change in outcome measures (cognition, tau accumulation).

Yet, substantial heterogeneity still exists within the Clinically Declining sample. We demonstrate that our prognostic index that quantifies individualised and continuous distance to the Clinically Stable prototype explains some of this heterogeneity both in the ADNI3 and BACS asymptomatic sample. In Fig. 8, we demonstrate how our prognostic index can be used to re-stratify the Clinically Declining sample. Focussing on the fusiform gyrus as a potential intervention target region we show that a more stringent threshold (Fig. 8, dashed black vertical line) than the probabilistic threshold used in the binary classification (Fig. 8, solid black vertical line) allows us to (a) select individuals with increased rate of tau accumulation (mean rate of accumulation: 0.028 vs. 0.0136 SUVR/Year), (b) reduce sample heterogeneity (variance: 0.00079 vs. 0.0012), (c) increase power to detect change in tau accumulation, reducing substantially the required sample size ($n = 93$ vs. $n = 598$). This more precise patient stratification has potential impact in clinical trial design, by reducing heterogeneity in the treatment and placebo groups that has been shown to hamper statistical power in clinical trials[25].

Finally, our multimodal stratification has greater statistical power to detect a clinically meaningful change in tau accumulation vs. cognitive decline (PACC change $n = 917$ vs. tau accumulation $n = 637$) over the time frame of a standard AD clinical trial (1–3 years), suggesting tau accumulation is a sensitive outcome measure for clinical trials in the earliest stages of AD. Further, our multimodal stratification—in contrast to Aß alone—selects a sample with higher rate and lower variability in tau accumulation, increasing statistical power to detect treatment effects (Aβ positive alone $n = 1139$ vs. Clinically Declining $n = 637$).

**Discussion**
Here, we employ a robust and transparent machine learning approach to combine continuous information across AD biomarkers and predict pathological changes in tau accumulation in asymptomatic and mildly impaired stages of AD. We use well-characterised AD biomarkers (Aβ, medial temporal grey matter

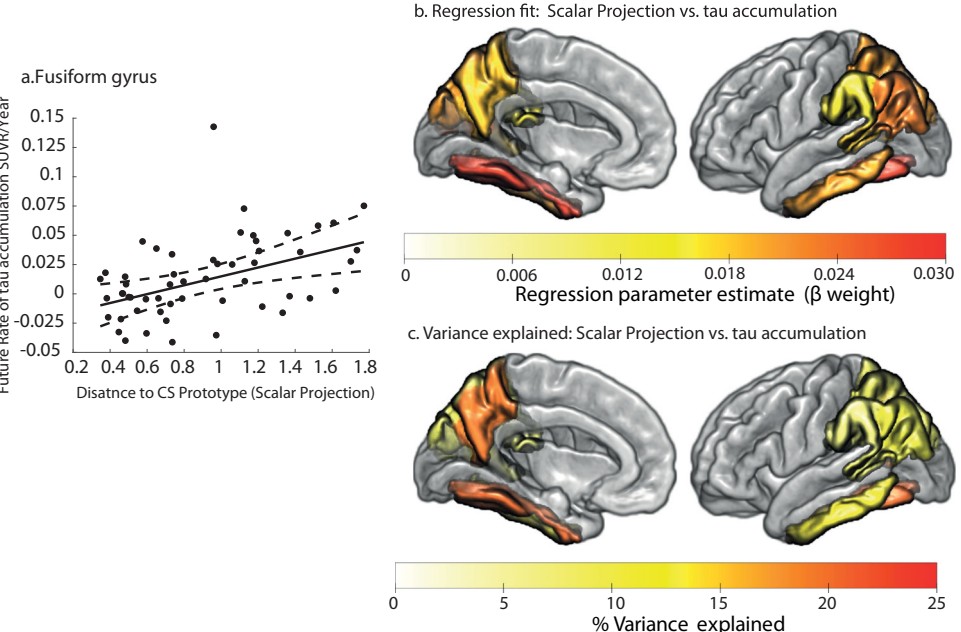

**Fig. 6 Predicting individual variability in future tau accumulation in the ADNI3 Clinically Declining sample. a** Regression fit of the scalar projection with future rate of tau accumulation for the fusiform gyrus (two-tailed linear regression, $p < 0.05$ FWE). The central black line indicates the regression line for the regression fit; the dashed lines indicate the 95% confidence intervals for this regression line. Two outliers based on the scalar projection are not shown for illustrative purposes. **b** Significant regional parameter estimates from linear regressions to predict future rate of tau accumulation for individuals classified as Clinically Declining (two-tailed linear regression, $p < 0.05$ uncorrected). The colour scale indicates the parameter estimate for the slope of the regression fit of the scalar projection with regional future tau accumulation. **c** Percentage of variance explained when using the scalar projection to predict future rate of tau accumulation for individuals classified as Clinically Declining from the ADNI3 cohort. Source data are provided as a Source Data file.

density, APOE 4) to derive a prognostic index, introducing a trajectory modelling approach that extends beyond binary patient stratification based on syndromic labels. Using this prognostic index derived from multimodal baseline data, we predict future tau accumulation, a known pathological driver of AD progression. We demonstrate that this multimodal prognostic index of future tau accumulation is a more sensitive tool for patient stratification than Aβ status alone or clinical syndrome labels. Further, our prognostic index predicts individualised spatially specific changes in tau accumulation in CN individuals, enabling fine stratification and staging at asymptomatic stages of AD (i.e., before clinical symptom occurrence). Our approach advances our understanding of the mechanisms of AD progression and has potential implications for the design of clinical trials.

In particular, using our trajectory modelling approach[37] we derive a prognostic index from baseline biomarkers that accurately classifies individuals as Clinically Declining. Using this prognostic index we show that individuals classified as Clinically Declining will accumulate tau in a topography-specific manner that reflects the initial spreading of tau in early-stage AD (i.e., prior to severe cognitive impairment)[23], accurately reproducing the topography reported in numerous independent cohorts corresponding to the proposed "meta-ROI" for tau quantitation[20–22]. Building on previous work, we show that our multimodal index is more sensitive for predicting tau accumulation compared to a unimodal measure (i.e., Aβ positive alone), demonstrating that individuals who are classified as Clinically Declining accumulate tau at 1.3 times the rate of Aβ positive individuals. Extending beyond binary classifications, we show that our continuous prognostic index predicts individual variability in future regional tau accumulation within regions that are known to be affected in early stages of AD. Further, we show that these individualised predictions generalise to an independent sample of CN individuals before symptoms occur.

Our approach has potential relevance for the design of clinical trials in four main respects. First, we demonstrate that our multimodal modelling approach is more sensitive in capturing early-stage AD-related pathology than a classification based on baseline syndromic labels. The poor sensitivity and specificity of syndromic labels to AD pathology[10–13] has led to the introduction of a biological framework for AD classification[8]. We show that syndromic labels are not consistent with baseline biology that predicts clinical decline (i.e., changes in longitudinal syndromic changes) or future pathological changes (i.e., tau accumulation).

Second, using the rate of tau accumulation as an outcome measure results in 31% reduction in the sample size for detecting a clinically meaningful change at early stages of AD compared to the gold standard cognitive instrument (PACC[41]). This is consistent with previous work showing that a smaller sample size is required to detect a clinically meaningful change in tau accumulation within the "meta-ROI" for tau accumulation than using a cognitive endpoint[22]. Although typically cognitive decline is considered as a primary outcome measure for clinical trials[42], recent trials indicate a potential future role for biomarkers in drug discovery (e.g., Aβ in the case of the recently FDA approved aducanumab). Further, recent trials in early AD participants have investigated downstream effects of Aβ lowering immunotherapies on both cognitive decline and changes in cortical tau burden measured with FTP-PET[24]. As tau is strongly linked to both future neurodegeneration and cognitive decline[18] this makes reduction of future tau accumulation a relevant intervention target and potential outcome measure. This is further evidenced by anti-tau drugs entering the clinical trial pipeline[43,44]. Given the limitation of anti-amyloid interventions in halting clinical decline, simply clearing already deposited tau may be insufficient to stop downstream events[44]. Therefore, measuring changes in tau and cognition simultaneously may be more appropriate for assessing efficacy of anti-tau drug treatments. That is, targeting individuals

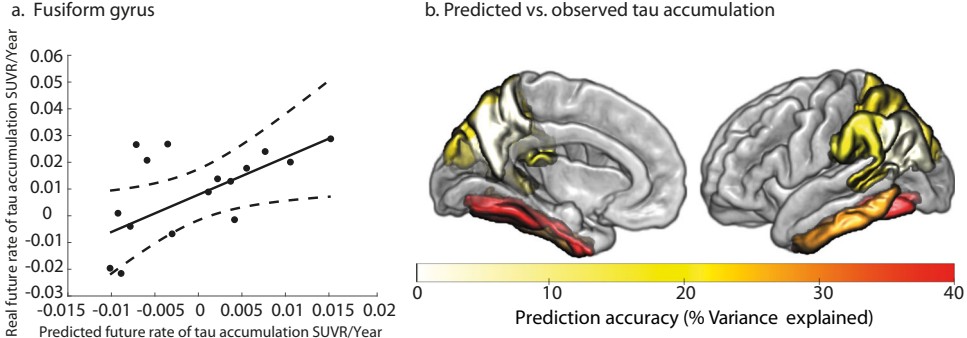

**Fig. 7 Prediction accuracy of future tau accumulation in the BACS Clinically Declining sample.** Comparing predicted tau accumulation to observed tau accumulation for individuals classified as Clinically Declining in the BACS cohort. **a** Fit of the predicted future rate of tau accumulation to the observed tau accumulation within the fusiform gyrus ($R^2 = 41\%$). The central black line indicates the regression line for the fit of predicted vs. observed future rate of tau accumulation; the dashed lines indicate the 95% confidence intervals for this regression line. One outlier based on the scalar projection is not shown. **b** Percentage of shared variance of the predicted regional future tau accumulation and the observed future tau accumulation for Clinically Declining individuals within each of the selected ROIs (Fig. 6a). Source data are provided as a Source Data file.

| Table 1 Predicting regional future annualised tau accumulation in BACS. | | | |
|---|---|---|---|
| **Braak stage** | **Region** | **Clinically Declining ($n = 17$)** | **Clinically Stable ($n = 39$)** |
| | | **Predicted variance explained %** | **Predicted variance explained %** |
| 3 | Fusiform | 40.58 | 1.7 |
| 4 | Inferior temporal | 28.96 | 2.25 |
| 5 | Supramarginal | 19.59 | 1.38 |
| | Inferior parietal | 3.38 | 8.78 |
| | Superior parietal | 22.09 | 5.45 |
| | Precuneus | 2.57 | 0.04 |
| | Bankssts | 21.94 | 0.57 |
| Shared variance of the predicted regional future tau accumulation and the observed future tau accumulation for individuals from the BACS cohort classified as Clinically Stable or Clinically Declining. Source data are provided as a Source Data file. | | | |

with the highest risk of depositing tau rather than those burdened with tau, may increase the likelihood of successfully modifying downstream clinical decline. Our machine learning approach is well suited to address this need as it is sensitive to baseline tau and allows us to select individuals who are predicted to both accumulate tau and have declining cognition.

Third, we show that using our multimodal prognostic index vs. Aß status alone reduces the sample size required to observe a clinically meaningful change in pathological tau accumulation by 44% (Clinically Declining $n = 636$, Aß positive $n = 1139$). This result extends a recent study investigating predictors with the most independent utility in predicting future rate of tau accumulation[45]. This previous work suggests that when considering key AD biomarkers (i.e., APOE 4, Aß and neurodegeneration) Aß status alone is the optimal independent biomarker for stratification to predict future tau accumulation. Our machine learning approach captures predictive disease-related covariance in biomarkers, showing a clear benefit in using multivariate predictors over Aß status alone when stratifying for clinical trials targeting future tau accumulation. The benefit of integrating multimodal biomarkers has been demonstrated in the context of predicting future changes in cognition[46–51]. In particular, previous studies have shown that grey matter atrophy and cortical Aß burden relate to separable patterns of future cognitive decline[46,47,50,51], and longitudinal changes in tau relate to cognitive decline in preclinical AD[18].

Fourth, our trajectory modelling approach shows that there is a linear relationship between baseline non-tau biomarkers and future rates of tau accumulation over a short timespan (typical of a clinical trial). This result has particular relevance to clinical trial design, as it suggests that patient stratification for clinical trials can be optimised to select individuals with appropriate rate of tau deposition, reducing heterogeneity in treatment and placebo groups may lead to erroneous conclusions in clinical trials[25].

Future work is needed to extend our modelling approach and enhance generalisation to diverse data types and populations. First, it is likely that our measure of medial temporal grey matter density captures information specific to typical amnestic AD populations, as it was derived using ADNI data. Additional measures of atrophy may contribute to a larger scale predictive model, as dissociable patterns of tau spreading have been observed for atypical AD syndromes[52].

Second, we focussed on specific well-studied biomarkers (i.e., Aß, medial temporal grey matter density and APOE 4) to make robust predictions, as evidenced by the consistency of our results across samples with different Aß tracers (i.e., FBP in ADNI and PiB in BACS) and MRI field strength. Extending our biomarker modelling approach to integrate less-costly (i.e., plasma) and non-invasive (i.e., cognitive) data has strong potential to determine the most cost-effective approach for stratification at asymptomatic or early stages of AD.

Third, our linear modelling approach, focusing on data from CN and MCI individuals, captures the earliest changes in tau accumulation over a limited timespan (i.e., 3 years). We show that a linear model predicts individual variation in future tau accumulation within multiple samples and over this time frame that is typical of a clinical trial (i.e., early to intermediate pathological stages). This model fits the majority of individual predictions with only a small number of outlier cases due to a high scalar projection (ADNI3: $n = 2$, BACS: $n = 1$).

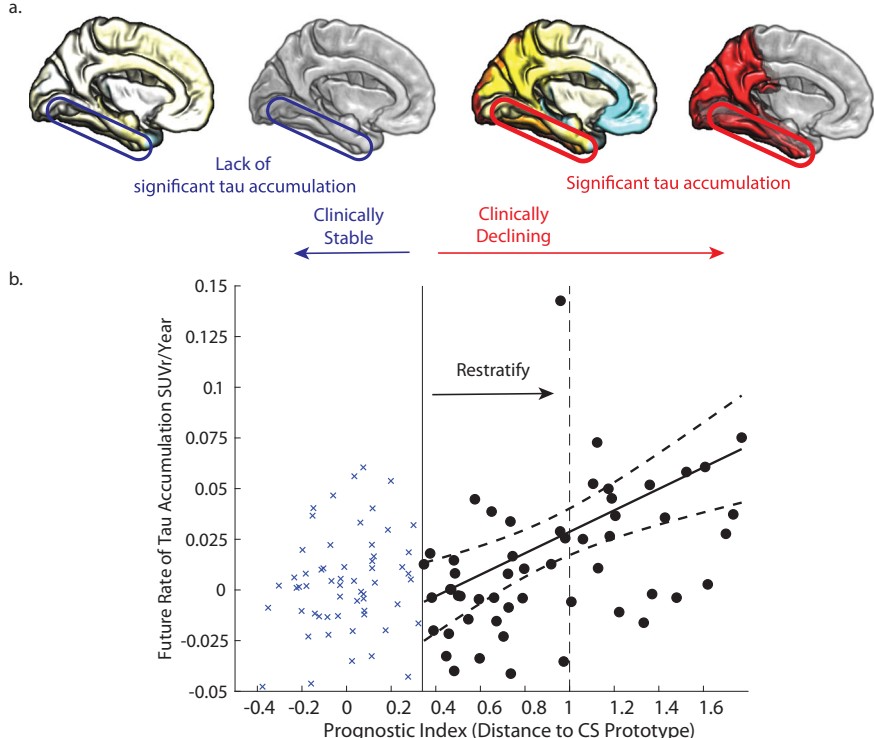

**Fig. 8 Potential application in clinical trial design. a** Cortical maps show average rate of tau accumulation for individuals classified as Clinically Stable vs. Clinically Declining (see Fig. 4). **b** Relationship of the scalar projection with future rate of tau accumulation within the Fusiform gyrus (as shown in Fig. 6a). The solid black vertical line indicates the probabilistic boundary used to perform the binary stratification, blue crosses indicate rate of tau accumulation for the clinically stable group, black circles indicate future rate of tau accumulation for the clinically declining group. Using our prognostic index (i.e., scalar projection) we show that re-stratifying to a more stringent threshold—as indicated by the dashed black vertical line—a new sample can be selected with higher future rates of tau accumulation and lower heterogeneity within the sample.

Fourth, we validated our model—that was trained on data from an AD disease-specific cohort—by testing predictions in an independent CN sample. This provides evidence that our results are not driven by the sampling characteristics of ADNI, suggesting generalisability (albeit in the small BACS sample) of our modelling approach to more diverse groups. Our asymptomatic sample size was limited, as publicly available data from CN participants with longitudinal FTP-PET are scarce. Larger samples with longitudinal data would increase the generalisability and validate the real-world efficacy of our approach.

In sum, our machine learning approach successfully capitalises on longitudinal data to make sensitive and specific predictions of early-stage AD trajectories based on baseline pathophysiology. Our modelling approach provides two key advances: (a) it combines multimodal continuous biological measures to capture trajectories for individuals who may be on the threshold of unimodal biomarker positivity but likely to follow AD related trajectories[17], (b) it harmonises longitudinal data collected using syndromic diagnostic criteria[6,7] (e.g., ADNI[27]) by means of a model-derived prognostic index. Importantly, our approach has translational impact for clinical trial design, as our prognostic index supports more precise patient stratification for inclusion to clinical trials, reducing sample heterogeneity that may lead to erroneous conclusions in clinical trials[25]. Using our prognostic index to select participants within a range of projected tau accumulation has potential to (a) reduce sample heterogeneity that hampers statistical power, (b) target individuals at greatest risk who may benefit the most from clinical intervention (c) decrease the required sample, resulting in more timely and cost-effective clinical trial. Our modelling approach can be tailored to trade off sample size, cost (from subjects screened but rejected

from inclusion), and generalisability for a sample with the highest probability of benefitting from treatment. Our findings highlight the strong potential for machine learning to extract informative disease markers from rich and complex multimodal data and deliver tools with high predictive power for early diagnosis and precise patient stratification.

## Methods

**Study design and participants**. Three separate cohorts were used to generate and test predictive models of regional future tau accumulation. For ADNI, ethical approval was obtained by the ADNI investigators. For BACS, ethical approval was obtained from the institutional review board at Lawrence Berkeley National Laboratory and the University of California, Berkeley. All ADNI and BACS participants provided written informed consent. Primary analysis was performed by investigators not involved in ADNI or BACS recruitment.

Two cohorts were drawn from the ADNI database: ADNI2/GO and ADNI3 (adni.loni.usc.edu). ADNI was launched in 2003 as a public-private partnership, led by Principal Investigator Michael W. Weiner, MD. A major goal of ADNI has been to examine biomarkers including serial magnetic resonance imaging, and positron emission tomography, with clinical and neuropsychological assessment to predict outcomes in MCI and AD.

A third validation cohort was taken from the BACS. This cohort is comprised of a convenience sample of community-dwelling cognitively intact elderly individuals with a Geriatric depression scale[53] score ≤10, Mini mental status examination (MMSE)[54] score ≥25, no current neurological and psychiatric illness, normal functions on verbal and visual memory tests (all scores ≥ −1.5 SD of age-adjusted, gender-adjusted, and education-adjusted norms) and age of 60–90 (inclusive) years. All subjects underwent a detailed standardised neuropsychological test session and neuroimaging measurements, all of which were obtained in close temporal proximity with follow-up every 1–2 years.

Data from 437 individuals from ADNI2/GO were used to train the machine learning model. Individuals were placed into three categories based on their baseline and longitudinal syndromic labels from clinical diagnosis independent of their baseline biomarker status, with baseline defined as the evaluation closest to the first florbetapir (FBP) PET scan acquired in ADNI. Alzheimer's Clinical Syndrome (n = 181, 158 Aß+ at baseline, APOE 4(+/-) = 119/62, Age

mean = 73.7+-std = 6.3 years, Education mean = 16.7+-std = 2.7 years, Sex (M/F) = 107/74): individuals have a stable diagnosis of dementia (in ADNI this corresponds to AD); Clinically Stable (n = 100, 18 Aß+ at baseline, APOE 4(+/-) =21/79, Age mean = 73.7+-std = 6.3 years, Education mean = 16.7+-std = 2.7 years, Sex (M/F) = 51/49): individuals have a baseline diagnosis of CN and retain this diagnosis at follow-up for 4 or more years (mean = 5.7+-std = 1.1 years); Clinically Declining (n = 156, 130 Aß+ at baseline, APOE 4(+/-) = 95/61, Age mean=74.9+-std = 7 years, Education mean = 15.9+-std = 2.7 years, Sex (M/F) = 88/68): individuals have a baseline diagnosis (at date of FBP scan) of either CN (n = 17) or MCI (n = 139) but received a diagnosis of dementia in future clinical evaluation (i.e., progressed to dementia (n = 75)), or had been diagnosed as demented in a clinical evaluation prior to baseline (i.e., reverted (n = 81)). We included individuals in the Clinically Declining group who were MCI at baseline but have received a diagnosis of dementia prior to baseline (i.e., reverted) in this group as we anticipate they are likely affected by AD pathology but are at an earlier stage of AD than the Alzheimer's Clinical Syndrome (i.e., late AD) group. Further, as our machine learning model is designed with limited parameters, when training using noisy diagnostic labels it is optimised to account for target uncertainty without leading to over-fitting. Therefore, including the additional training samples (i.e., 81 individuals who reverted to MCI) will likely improve model training even though their diagnostic labels may have poor reliability.

Data from 115 individuals from ADNI3 were used to test the relationship between the scalar projection and regional future tau accumulation. These individuals were either CN (n = 72) or MCI (n = 43) (61 Aß+ at baseline, APOE 4(+/-)=51/64, Age mean=73.7+-std = 6.9 years, Education mean=16.6+-std = 2.3 years, Sex (M/F) = 58/57) at baseline, defined as the diagnosis closest to the first FTP-PET scan acquired in ADNI3, and have at least one follow-up FTP-PET scan.

Data from 56 community-dwelling individuals from BACS were used to test the accuracy of predictions of regional future tau accumulation. These individuals were CN (n = 56, 30 Aß+ at baseline, APOE 4(+/-) = 17/39, Age mean = 77.2+-std = 5.2 years, Education mean=16.7+-std = 1.8 years, Sex (M/F) = 22/34) at baseline (defined as the diagnosis closest to the first FTP-PET scan acquired in BACS) and have at least one follow-up FTP-PET scan.

**MRI acquisition.** Structural MRIs for the ADNI samples were acquired at ADNI-GO, ADNI2 and ADNI3 sites equipped with 3T MRI scanners using a 3D MP-RAGE or IR-SPGR T1-weighted sequences, as described online (http://adni.loni.usc.edu/methods/documents/mri-protocols). Structural MRIs for the BACS sample were collected on either a 1.5T MRI scanner at Lawrence Berkeley National Laboratory (LBNL) or a 3T MRI scanner at UC Berkeley using 3D MP-RAGE T1- weighted sequences. All ADNI and BACS scans were acquired with voxel sizes of approximately 1 mm × 1 mm × 1 mm.

**PET acquisition.** PET imaging was performed at each ADNI site according to standardised protocols. The FBP-PET protocol entailed the injection of 10 mCi with acquisition of 20 min of emission data at 50–70 min post injection. The FTP-PET protocol entailed the injection of 10 mCi of tracer followed by acquisition of 30 min of emission data from 75–105 min post injection.

For the BACS, PIB PET scans were collected at LBNL. After ~15 mCi tracer injection into an antecubital vein, dynamic acquisition frames were obtained in 3D acquisition mode over a 90 min measurement interval (4 × 15 s frames, 8 × 30 s frames, 9 × 60 s frames, 2 × 180 s frames, 8 × 300 s frames, and 3 × 600 s frames) after X-ray CT. FTP-PET scans were collected following an injection of ~10 mCi of tracer in a protocol highly similar to that used for ADNI with a slightly shorter window of emission data used (80–100 min post injection). All BACS participants were studied on a Siemens Biograph PET/CT.

**Imaging analysis-MRI: medial temporal grey matter density.** All structural MRI pre-processing was performed using Statistical Parametric Mapping 12 (http://www.fil.ion.ucl.ac.uk/spm/). Structural scans were segmented into grey matter, white matter and Cerebrospinal Fluid (CSF). The DARTEL toolbox[55] was then used to generate a study specific template to which all scans were normalised. Following this, individual grey matter segmentation volumes were normalised to MNI space without modulation. The unmodulated values for each voxel represent grey matter density at the voxel location. All images were then smoothed using a 3mm3 isotropic kernel and resliced to MNI resolution 1.5 × 1.5 × 1.5 mm voxel size.

To generate a single index of medial temporal grey matter density we used a voxel weights matrix that was previously derived to generate an interpretable and interoperable disease-specific biomarker[37]. In brief, a feature generation methodology (partial least squares regression with recursive feature elimination (PLSr-RFE)) was used to apply a decomposition on a set of predictors (T1-weighted MRI voxels) to create orthogonal latent variables that show the maximum covariance with the response variable (memory score). Further, we performed recursive feature elimination by iteratively removing predictors (voxels) that have weak predictive value. The PLSr-RFE procedure results in a voxel weights matrix that is used to calculate a single score of AD related medial temporal density. This index of medial temporal grey matter density has been shown to predict memory deficits, relate to individual tau burden and discriminates stable MCI and progressive MCI individuals[37]. To generate an individual's score of medial temporal grey matter density we performed a matrix multiplication of the previously derived voxel weights matrix and each subject's pre-processed T1-weighted MRI scans. Given that this value represents density and not regional volume, the medial temporal grey matter density score is not affected by head size differences (Supplementary Fig. 7).

**Imaging analysis (ADNI)-PET: FBP (florbetapir PET) Aβ.** FBP data were realigned, and the mean of all frames was used to co-register FBP data to each participant's structural MRI. Cortical Standardised Uptake Value Ratios (SUVR)s were generated by averaging FBP retention in a standard group of ROIs defined by FreeSurfer v5.3 (lateral and medial frontal, anterior and posterior cingulate, lateral parietal, and lateral temporal cortical grey matter) and dividing by the average uptake from the whole cerebellum to create an index of global cortical FBP burden (Aβ) for each subject[56]. Finally we converted the SUVR to the centiloid scale[57] using the following conversion taken from the LONI website CL = (FBP SUVR × 196.9) – 196.03 (http://adni.loni.usc.edu/methods/documents/, PET Protocols: ADNI Centiloids). To assign individuals as Aß positive we used the widely published threshold for ADNI FBP; SUVR = 1.1 or CL = 22.5[15].

**Imaging analysis (BACS)-PET: PiB (Pittsburgh Compound B) Aβ.** Distribution volume ratios (DVRs) were generated with Logan graphical analysis on the aligned PIB frames using the native-space grey matter cerebellum as a reference region, fitting 35–90 min after injection.

For each subject, a global cortical PIB index was derived from the native-space DVR image coregistered to the MRI using FreeSurfer (5.3) parcellations using the Desikan–Killiany atlas[58] to define frontal (cortical regions anterior to the precentral sulcus), temporal (middle and superior temporal regions), parietal (supramarginal gyrus, inferior/superior parietal lobules, and precuneus), and anterior/posterior cingulate regions- ROIs combined as a weighted average. There was no partial volume correction performed. Finally we converted PiB DVR values to centiloids using the following conversion CL = (PiB DVR * 142.73) – 141.99.

**Image analysis-PET: FTP (Flortaucipir PET) tau.** FTP data were realigned and the mean of all frames used to co-register FTP to each participant's MRI acquired closest to the time of the FTP-PET. FTP SUVR images were generated by dividing voxel wise FTP uptake values by the average value within a mask of eroded sub-cortical white matter regions[59]. MR images were segmented and parcellated into 72 ROIs taken from the Desikan–Killiany atlas using Freesurfer (V5.3). These ROIs were then used to extract regional SUVR data from the normalised FTP-PET images. Left and right hemisphere ROIs were averaged to generate 36 ROIs for further analysis. We calculated the future annualised rate of tau accumulation for each of the 36 ROIs either by taking the difference between the follow-up and baseline FTP-PET scans divided by the time interval in years from baseline (when only 2 FTP scans were taken), or fitting a linear least squares fit to 3 or more FTP-PET scans and extracting the parameter estimate for the slope of the ROI SUVR vs. time in years from baseline (when 3 or more FTP scans were taken). In the ADNI3 sample the average time between FTP-PET scans is 1.22+- std: 0.38 years with the number of follow-up FTP-PET scans n (2 FTP-PET scans) = 93, n (3 FTP-PET scans) =17, n (4 FTP-PET scans) = 5. In the BACS cohort the average time between FTP-PET scans is 1.8+-std:0.65 years with the number of follow-up FTP-PET scans n (2 FTP-PET scans) =37, n (3 FTP-PET scans) = 19.

**Predictors and outcomes.** Three baseline biological markers related to AD were used as predictors to generate the scalar projection from the machine learning model: (a) Cortical amyloid burden (Aβ) measured using either FBP (ADNI) or PiB (BACS) PET, (b) medial temporal grey matter density derived from the T1-weighted structural MRI and (c) APOE 4 genotype. To quantify cortical amyloid burden we utilised multiple PET tracers. To derive a robust scalar metric for predictions we first harmonised Aß PET values using the centiloid approach. Using this approach it has been shown that FBP and PiB amyloid tracers are interchangeable once scaled linearly onto a common scale (i.e., centiloids)[57].

We have previously shown that training our GMLVQ-scalar projection model on multimodal baseline data predicts future changes in cognition for individuals diagnosed with MCI[37]. Here, we use the same baseline predictors to generate predictions in a new sample of early AD individuals (i.e., individuals who are CN or MCI at baseline). The primary outcome measure for the predictive models is regional future annualised rate of tau accumulation (SUVR/year). To model patient trajectories of future tau accumulation we used longitudinal FTP-PET. The association between antemortem FTP uptake and neurofibrillary tangles load post-mortem has been shown previously[60]. However, FTP retention is associated with significant off-target binding. To mitigate this, we used a reference region from eroded subcortical white matter regions. Previous work has shown that FTP-PET uptake in subcortical regions accounts for 60% of the variation global FTP uptake in healthy amyloid negative CN individuals[61]. A secondary outcome measure is changes in future cognition over the same time scale as the longitudinal FTP scans, as measured by the PACC. To test changes in future cognition for individuals from ADNI3 we used the previously derived ADNI-PACC measure (adni.loni.usc.edu)[41]. Of the 115 ADNI3 individuals with multiple FTP-PET scans 102 individuals had multiple measures of the PACC over a similar time period

(within 6 months of the baseline FTP-PET and the final FTP-PET scan). Future annualised change in PACC is calculated by either taking the difference between the follow-up and baseline PACC scores divided by the time interval in years from baseline (when only 2 PACC scores are available), or fitting a linear least squares fit to 3 or more PACC scores and extracting the parameter estimate for the slope of the PACC vs. time in years from baseline (when 3 or more PACC scores are available). The average time between PACC testing sessions scans is 1.04+-std: 0.44 years with the number of follow-up PACC testing sessions $n$ (2 PACC sessions) = 82, $n$ (3 PACC sessions) = 18, $n$ (4 PACC sessions) = 2.

### Prediction models

*Generalised Matrix Learning Vector Quantisation (GMLVQ)-Scalar Projection.* We previously developed a machine learning approach based on the GMLVQ classification framework: GMLVQ-Scalar Projection[37]. This approach allows us to derive a continuous prognostic metric (i.e., scalar projection) by training a model based on longitudinal diagnostic labels.

Learning Vector Quantisation (LVQ) are classifiers that operate in a supervised manner to iteratively modify class-specific prototypes to find boundaries of discrete classes. In particular, LVQ classifiers are defined by a set of vectors (prototypes) that represent classes within the input space. These prototypes are updated throughout the training phase, resulting in changes in class boundaries. For each training example, the closest prototype for each class is determined. These prototypes are then updated so that the closest prototype representing the same class as the input example is moved towards the input example and those representing different classes are moved further away. The Generalised Matrix LVQ (GMLVQ)[62] extends the LVQ utilising a full metric tensor for a more robust (with respect to the classification task) distance measure in the input space. To do this, the metric tensor induces feature scaling in its diagonal elements, while accounting for task conditional interactions between pairs of features (co-ordinates of the input space) (Supplementary methods: *Generalised matrix learning vector quantisation*). Previously we trained a GMLVQ model with baseline multimodal data (medial temporal grey matter density, Aβ, APOE 4 genotype) and show that the GMLVQ modelling approach classifies MCI patients into subgroups (progressive vs. stable) with high specificity and sensitivity[37].

Extending the binary model, we derived a single prognostic distance measure (scalar projection) that separates individuals based on their longitudinal diagnosis. The GMLVQ-scalar projection approach derives a continuous distance metric from the trained GMLVQ classifier. This continuous distance measure (scalar projection) indicates how far an individual is from the Clinically Stable prototype along the dimension that best separates individuals who have stable (i.e., Clinically Stable) vs. declining (i.e., Clinically Declining) syndromic diagnosis. This allows the model to learn implicitly a continuous prognostic score for an individual that may be predictive of underlying pathophysiological change. Previously we calculated the scalar projection separating individuals who are stable MCI from progressive MCI showing that the continuous value predicts individualised rates of future cognitive decline[37]. Here, we apply the same framework on a new sample, deriving the scalar projection on Clinically Stable vs. Clinically Declining and making individualised predictions of future regional tau accumulation in Clinically Declining populations.

*GMLVQ-scalar projection implementation.* From the training sample, the model learns the multivariate relationship between Aβ, medial temporal grey matter density and APOE 4 (metric tensor $\Lambda$) and the location in multidimensional space that best classifies Clinically Stable vs. Clinically Declining individuals (prototype locations: $\mathbf{w}_{(\text{Clinically Stable, Clinically Declining})}$). For any new subject with Aβ, medial temporal grey matter density and APOE 4 (sample vector: $\mathbf{x_i}$) the scalar projection can be calculated by a series of simple linear equations.

1. Transform the sample vector $\mathbf{x_i}$ and prototypes $\mathbf{w}_{(\text{Clinically Stable, Clinically Declining})}$ into the learnt space via the metric tensor $\Lambda$. Note as the metric tensor is learnt in the squared Euclidean space we transform using the square root of the metric tensor (i.e., $\Lambda^{1/2}$)

$$\mathbf{X_i} = \Lambda^{1/2}\,\mathbf{x_i} \tag{1a}$$

$$\mathbf{W}_{(\text{Clinically Stable, Clinically Declining})} = \Lambda^{1/2}\mathbf{w}_{(\text{Clinically Stable, Clinically Declining})} \tag{1b}$$

2. Centre the co-ordinate system on $\mathbf{W}_{(\text{Clinically Stable})}$ and calculate the orthogonal projection of each vector $\mathbf{X_i}$ onto the vector $\overline{\mathbf{W}_{\text{Clinically Declining}}\mathbf{W}_{\text{Clinically Stable}}}$, in this co-ordinate system.

$$\text{Projection} = \frac{\overline{\mathbf{X_i}\mathbf{W}_{\text{Clinically Stable}}} \cdot \overline{\mathbf{W}_{\text{Clinically Declining}}\mathbf{W}_{\text{Clinically Stable}}}}{\left|\overline{\mathbf{W}_{\text{Clinically Declining}}\mathbf{W}_{\text{Clinically Stable}}}\right|} \tag{2a}$$

3. To normalise the projections with respect to the position of the prototype $\mathbf{W}_{(\text{Clinically Stable})}$, then we divided the projection by the norm of

$\overline{\mathbf{W}_{\text{Clinically Declining}}\mathbf{W}_{\text{Clinically Stable}}}$:

$$\text{Scalar Projection} = \frac{\overline{\mathbf{X_i}\mathbf{W}_{\text{Clinically Stable}}} \cdot \overline{\mathbf{W}_{\text{Clinically Declining}}\mathbf{W}_{\text{Clinically Stable}}}}{\left|\overline{\mathbf{W}_{\text{Clinically Declining}}\mathbf{W}_{\text{Clinically Stable}}}\right|^2} \tag{3a}$$

For a graphical derivation and interpretation see Supplementary Methods: *GMLVQ-Scalar Projection.*

To determine a meaningful threshold of the scalar projection for separating individuals who are Clinically Stable from Clinically Declining individuals we use logistic regression for the ADNI2/GO sample labelled as Clinically Stable, Clinically Declining, and Alzheimer's Clinical Syndrome. This results in a probabilistic boundary based on the scalar projection.

*Determining regions of significant AD-related tau accumulation.* We first classified individuals from ADNI3 as either Clinically Stable or Clinically Declining based on each individual's scalar projection. For the individuals who are classified as Clinically Declining (based on the probabilistic threshold value) we performed a subsequent first level analysis to determine which of the 36 selected ROIs will accumulate tau in the future (i.e., regions with a future annualised rate of accumulation statistically greater than 0).

*Predicting individual variability in regional tau accumulation.* Finally, for regions that pass first level significance (i.e., within regions that significantly accumulating tau $p < 0.05$, uncorrected) we trained a series of regression models using ADNI3 individuals classified as Clinically Declining to test if the scalar projection relates to individual variability in regional future rate of tau accumulation (dependent variable: regional future tau accumulation, independent variable: scalar projection). We then tested these models by making individualised predictions—out of sample—for individuals classified as Clinically Declining from the BACS sample. To test the accuracy of the regional predictions we calculated the shared variance between the observed future accumulation of tau and the model generated prediction using baseline biological data (i.e., scalar projection).

### Statistical analysis

Within-sample accuracy for classifying Clinically Stable vs. Clinically Declining in the ADNI2/GO sample was assessed using random resampling. In brief, we determined within-sample classification accuracy by randomly splitting our sample into test and training data 400 times. To avoid biasing the model in the training phase due to class imbalance in the data (majority class: Clinically Declining = 156 vs. minority class: Clinically Stable = 100), we resampled the data to generate balanced classes (i.e., number of Clinically Declining equals number of Clinically Stable individuals). This resampling process randomly selects half of the individuals in the minority class and the same number of individuals from the majority class as training data; with the remaining sample used as test data. We then averaged the true positive and true negative accuracies across the 400 resampling's to generate a class-balanced cross-validated accuracy.

We used logistic regression to define a probabilistic boundary that separates individuals who are Clinically Stable from Clinically Declining. Using the ADNI2/GO sample we fit a three class (Clinically Stable, Clinically Declining, Alzheimer's Clinical Syndrome) logistic regression to determine the threshold value of the scalar projection. We set the threshold as the probability an individual is less than 50% likely to be Clinically Stable. To determine the regions that will significantly accumulate tau for individuals classified as Clinically Declining we used one-tailed one sample $t$-tests. As we are testing if regions are accumulating tau we use right tail t-tests to determine if the future rate of tau accumulation is significantly ($p < 0.05$, uncorrected) greater than 0 per ROI for individuals classified as Clinically Declining. To compare required sample sizes for different models derived using ADNI3 data we calculated the sample size needed for an arm of a hypothetical clinical trial designed to detect a 25% reduction in annual change (rate of tau accumulation, rate of PACC decline) with a significance of 0.05 and a power of $a = 0.8$. For each comparison, we defined the null hypothesis as the mean and standard deviation of the rate of change calculated from the observed sample, where the alternate hypothesis is a 25% reduction of the mean of the observed sample. For each of the regions that showed significant tau accumulation we fit a robust linear regression (robustfit MATLAB) to predict future tau accumulation using the prognostic index. Setting the dependent variable as future regional tau accumulation and the independent variable as the scalar projection we learnt a series of ROI regression equations.

Rate of tau accumulation $(\text{ADNI3})_{\text{ROI}}$

$= \beta(\text{ADNI3})_{\text{ROI}} * \text{Clinically Declining Scalar Projection(ADNI 3)} + \beta_0(\text{ADNI3})_{\text{ROI}}$ (4)

Finally, using the significant ($p < 0.05$, uncorrected) fits derived from ADNI3 data we generated predictions of tau accumulation for individuals classified as

Clinically Declining in BACS.

Predicted rate of tau accumulation (BACS)$_{ROI}$

$$= \beta(\text{ADNI3})_{ROI} * \text{Clinically Declining Scalar Projection(BACS)} + \beta_0(\text{ADNI3})_{ROI} \quad (5)$$

We tested the accuracy of these predictions in the BACS sample by calculating the shared variance between the predicted future rate of tau accumulation and the observed future rate of tau accumulation after treating for outliers (robust correlation[63]).

**Reporting summary**. Further information on research design is available in the Nature Research Reporting Summary linked to this article.

## Data availability

The summary data generated in this study have been deposited in the University of Cambridge online data repository (https://doi.org/10.17863/CAM.80891). ADNI data is accessible via adni.loni.usc.edu. Additional BACS data are available on request.

## Code availability

Custom code used in this work is available via the University of Cambridge online data repository (https://doi.org/10.17863/CAM.80891).

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

## Acknowledgements

We thank Avraam Papadopoulos for help with computational resources. This work was supported by grants to: Z.K. from the Biotechnology and Biological Sciences Research Council (H012508 and BB/P021255/1), Alan Turing Institute (TU/B/000095), Wellcome Trust (205067/Z/16/Z, 221633/Z/20/Z), Royal Society (INF\R2\202107); Z.K. and W.J.J. from the Global Alliance; W.J.J. US National Institute on Aging (AG034570, AG062542, AG024904). Data collection and sharing for this project was funded by the Alzheimer's Disease Neuroimaging Initiative (ADNI) (National Institutes of Health Grant U01 AG024904) and DOD ADNI (Department of Defense award number W81XWH-12-2-0012). ADNI is funded by the National Institute on Aging, the National Institute of Biomedical Imaging and Bioengineering, and through generous contributions from the following: AbbVie, Alzheimer's Association; Alzheimer's Drug Discovery Foundation; Araclon Biotech; BioClinica, Inc.; Biogen; Bristol-Myers Squibb Company; CereSpir, Inc.; Cogstate; Eisai Inc.; Elan Pharmaceuticals, Inc.; Eli Lilly and Company; EuroImmun; F. Hoffmann-La Roche Ltd and its affiliated company Genentech, Inc.; Fujirebio; GE Healthcare; IXICO Ltd.; Janssen Alzheimer Immunotherapy Research & Development, LLC.; Johnson & Johnson Pharmaceutical Research & Development LLC.; Lumosity; Lundbeck; Merck & Co., Inc.; Meso Scale Diagnostics, LLC.; NeuroRx Research; Neurotrack Technologies; Novartis Pharmaceuticals Corporation; Pfizer Inc.; Piramal Imaging; Servier; Takeda Pharmaceutical Company; and Transition Therapeutics. The Canadian Institutes of Health Research is providing funds to support ADNI clinical sites in Canada. Private sector contributions are facilitated by the Foundation for the National Institutes of Health (www.fnih.org). The grantee organisation is the Northern California Institute for Research and Education, and the study is coordinated by the Alzheimer's Therapeutic Research Institute at the University of Southern California. ADNI data are disseminated by the Laboratory for Neuro Imaging at the University of Southern California. For the purpose of open access, the author has applied for a CC BY public copyright licence to any Author Accepted Manuscript version arising from this submission. Data used in preparation of this article were obtained from the Alzheimer's Disease Neuroimaging Initiative (ADNI) database (adni.loni.usc.edu). As such, the investigators within the ADNI contributed to the design and implementation of ADNI and/or provided data but did not participate in analysis or writing of this report. A complete listing of ADNI investigators can be found at: https://adni.loni.usc.edu/wp-content/uploads/how_to_apply/ADNI_Acknowledgement_List.pdf.

## Author contributions

J.G.: Conceptualisation, formal analysis, investigation, methodology, writing - original draft, writing - review & editing. W.J.J.: Conceptualisation, data curation, investigation, writing - original draft, writing - review & editing. S.B.: Conceptualisation, data curation, formal analysis, investigation, writing - original draft, writing - review & editing. S.M.L.: Conceptualisation, data curation, formal analysis, investigation, writing - original draft, writing - review & editing. P.T.: Conceptualisation, investigation, methodology, writing - original draft, writing - review & editing. Z.K.: Conceptualisation, investigation, methodology, writing - original draft, writing - review & editing.

## Competing interests

W.J.J. has served as consultant to Biogen, Lilly, and Bioclinica. S.M.L. serves on the advisory board for KeifeRx, has received research support from the Alzheimer's Association, and has received compensation from Eisai for speaking. S.B. serves as consultant for Genentech. All other authors declare no competing interests.

## Additional information

# Alzheimer's Disease Neuroimaging Initiative

William J. Jagust [2,3] & Susan M. Landau[2]

