## [Peer review file · Nature Communications]

REVIEWER COMMENTS

Reviewer #1 (Remarks to the Author):

Key aim of this study: predicting spatiotemporal tau accumulation in individuals potentially at risk of Alzheimer's disease. Predicting a map of future tau (PET) accumulation _rates_ in regions of an individual's brain, in vivo.

The two primary audiences for this paper are "methods developers" (computational/statistical researchers) and "applications" researchers in Alzheimer's disease (e.g., patient-facing roles such as clinicians).

Methods: use supervised machine learning (previously published in [19]) to combine relevant features from an early Alzheimer's disease cohort (training data) into a scalar score through linear equations, which is thresholded into a data-driven cut-point to define biomarker-based subgroups of cognitively normal individuals (test data), then calculate group-level associations with future tau accumulation. This study is a straightforward application of the method developed by the authors in [19] to a new test sample, although the test data here is arguably more challenging because individuals are at an earlier stage of disease (cognitively normal here, MCI in [19]).

Confounders: analyses are adjusted for age, education, and gender norms (although, perhaps the authors mean sex rather than gender).

Data: from 2 or 3 cohorts (two phases of ADNI: GO/2 and 3; plus BACS). I say "2 or 3" because there is overlap between the ADNI-GO/2 and ADNI-3 phases. Were the "rollovers" into ADNI-3 excluded from the ADNI-3 portion of experiments? The authors do not make this clear. Sample size is good for the training set (ADNI-GO/2 n=488), but less so for the test sets (ADNI-3 n=115, BACS n=56). I can't help but feel that there might be larger datasets available that would add confidence of generalisability of the findings. Bare minimum is to add a discussion on this limitation. How generalisable do the authors genuinely expect their findings to be? Justify, and quantify if possible.

Findings: group-level associations between the individual scalar scores and future tau accumulation rates. While these are individual predictions, the results are analysed statistically at the group level, which does not align particularly well with the individual-level aims of the study. The results are presented clearly, but the implementation of these results in (clinical) practice is unclear (Line 304 "strong clinical relevance" might be overstating it). The model achieves decent classification performance but, in particular, the outlier cases are not discussed, i.e., where the individual predictions diverge considerably from the average – how are these model predictions converted to clinical decisions for an individual? What are the implications/ramifications for incorrect/poor decisions? Is the model itself to blame? How suitable is a linear model for a disease that likely plays out in a nonlinear fashion? Perhaps the model inputs are inadequate (more biomarkers?)?

That being said, the results around Figure 5 are okay because clinical trials are indeed assessed at the group level. The particular novelty here is in using a spatiotemporal biomarker (admittedly projected into a linear decision space) rather than a simple scalar biomarker. The proof of the pudding would be to apply this in an actual clinical trial (obviously a longer-term future aim!).

In the discussion (line 297) the authors statement about multivariate vs univariate is well-known and not novel. What's missing here is a solid baseline/benchmark model for direct comparison (of apples with apples, as one might say). For example, comparing performance of the classifier here with that of a different multivariate classifier trained on the same multivariate data.

Summary: I find the results interesting on some levels (mostly in terms of the application), if not especially novel (particularly methodologically). Additionally, I have minor question marks over the

data (small sample size in test set; potential train/test overlap in ADNI-GO/2 and ADNI-3).

Detailed list of additional thoughts/changes, in no particular order:

- Please check that no individuals from ADNI-3 were involved in the training set (ADNI-GO/2). If they were, remove them from the test set and rerun the experiments. Clarify in manuscript.
- Discuss individual-level applicability of findings, with particular regard to outliers and clinical decision making.
- Discuss generalisability of findings, with regard to the relatively small sample sizes in the test sets.
- Change the "demented" terminology to "probable AD", or simply "AD".
- Add a brief discussion of differences/similarities between FBP/FTP/PIB PET tracers. Specifically, in relation to the findings. Perhaps off-target binding is a confounder in some regions of interest (perhaps not, which is also worth remarking in the manuscript).
- Line 458. Please start this paragraph by mentioning SPM, so it's clear.
- Discuss clinical utility of predicting `_rates_` of tau accumulation versus raw tau burden. Two individuals could both be accumulating slowly at the same rate, but have completely different tau burden - clearly not the same.

Reviewer #2 (Remarks to the Author):

Thank you for giving me the opportunity to review this very interesting paper. The authors train a series of algorithms with the goal of predicting individualised future regional tau accumulation. This is an important effort, in-part due to circumventing issues in binary categorical classifications often employed. Using a novel GMLVQ technique, early AD is modelled in a multi-dimensional space, within which subgroups are represented by a prototype. Distance from the prototype can be used to represent disease on a continuum (scalar projection), and given new instances with the same feature vector their scalar projection can be calculated. Thresholding the scalar projection enables classification of discrete groups.

For the current paper, first a GMLVQ is trained to model a stable condition (SC) and early AD (EAD) based on the multivariate relationship between amyloid, medial temporal grey matter density and APOE 4. Thus, the scalar projection models prognosis on a continuum between the SC prototype and EAD prototype based on the input variables. Second, the scalar projections for novel data are calculated, and threshold used to classify as EAD or not. For those classified as EAD, brain regions accumulating tau were identified, and the future rate of tau accumulation was predicted for each region using regions of interest and the scalar projection. Finally, future rate of tau accumulation was predicted in an unseen dataset.

Results detail variance explained globally and in regions of interest of future tau accumulation, only found for the EAD group and not SC, supporting this novel approach to stratification that is based on underpinning biology, and characterised using a novel machine learning approach using the scalar projection (or prognostic index). Interestingly for a hypothetical clinical trial sample size required could be reduced by 35% to detect a 25% reduction in future tau accumulation, compared to a 25% reduction in future cognitive decline. A reduction of 30% was found when using EAD classification compared to amyloid positivity alone.

Particular strengths of the paper are the use of multiple datasets to train and test, including out of sample testing; use of resampling methods to create a balanced dataset to train the GMLVQ scalar projection, thus not impacted by confounding variables, and the translational impact of using predictions to decrease required sample size for a hypothetical clinical trial.

I found the methods and results to be exciting. The machine learning and statistical approaches utilised are, as the authors state, robust and transparent. The discussion is very well written and places the findings in context of known research, including a good discussion of the limitations. I think

the manuscript could be improved by building on the introduction, which would aid in the flow of the paper overall, and by making clearer the groups and subgroups of interest throughout, perhaps with a figure. I go into these points in more detail below, along with other minor points.

- 1) I was confused by the groups and subgroups listed within the paper. This begins in the introduction, line 65, which states "In this framework, evidence of A β and pathological tau accumulation is sufficient to establish the diagnosis of AD. Cognitively unimpaired individuals (CN: cognitively normal) are classified as preclinical AD". Line 81 states "...classifies and stages early AD (i.e. CN and MCI)" and line 86 "early AD (i.e. A β positive individuals who are cognitively unimpaired...". I found this explanation a little blunt - we do know that not all of those with AD pathology will develop clinical AD (e.g. Bennett et al., 2006) - but understand that this is a definition based on evidence of AD pathology, as opposed to a clinical definition. However, I think the introduction could be built upon to better describe this focus on pathology/ biology, using terms elsewhere in the paper, which would also aid in the flow. The term 'asymptomatic' is used readily elsewhere, as is reference to CN and MCI as syndromic labels - given the interesting results regarding the mismatch between the biological stratification in the paper and these syndromic labels, I feel bringing this into the introduction earlier in the paper would aid the reader.
- 2) Overall, I think the paper (barring the discussion) is complicated by heavy use of acronyms for groups, which hinders readability. I think a figure detailing the different groups utilised, and for which analyses, would add clarity. Throughout, even after acronyms have been introduced, sometimes full groups labels are used and sometimes acronyms with no consistency, again reducing readability.
- 3) Relatedly, the 'Demented' group is almost a surprise addition, only briefly mentioned in the methodology, but features quite heavily in results and so could be brought in earlier as a group of interest.
- 4) And, my confusion was further compounded by the Supplementary Materials section GMLVQ - Scalar Projection, in which notation returns to progressive/ stable. I wonder if using EAD throughout would improve readability.
- 5) I felt a brief methodological breakdown of how APOE 4 positivity was defined was missing e.g. does the presence of one or two alleles = positivity?
- 6) Figure 2 caption could be condensed by stating (for example) 'the learnt probabilistic boundary that separates SC from EAD is indicated by a dashed vertical or horizontal line', prior to the a, b and c descriptions, rather than repeating this information.
- 7) Figure 3a uses a dashed black line to represent the threshold for EAD, are the dashed black lines on the box plots for each group related to the threshold information? If not perhaps replace them with solid lines to avoid confusion.
- 8) Line 57 bracket missing after (for reviews...)
- 9) Line 156 typo "that" should be "than"
- 10) Line 242 missing a space after 6b).
- 11) Line 312 - states a reduction in sample size of 46% based on prognostic index alone vs A β status alone. I could not seem to find this analysis in the results section, only for EAD classification vs A β status which was reported as 47% (line 223)? Is the same analysis?
- 12) Line 469 typo "RFE))" should be RFE)
- 13) Line 425 "cognition for individuals diagnosed as MCI". This may be personal preference but I would replace "as" with "with".
- 14) Line 935 typo "ii)an" should be ii) an.
- 15) Line 946 typo "(EAD)in" should be (EAD) in.
- 16) Line 952 typo "6c)." should be 6c.)

Reviewer #3 (Remarks to the Author):

The authors report findings of applying a machine learning approach to MRI volumetry and amyloid PET data and apoE4 carriage status to predict future tau accumulation measured with flortaucipir

(FTP) PET and cognitive decline obtained for cohorts from the ADNI database. The subjects comprised established Alzheimer dementia (AD), early Alzheimer's disease (EAD) and normal stable cases (CS). The machine learning derived a predictive index that stratified individuals based on their future pathological tau accumulation. The hypotheses were that future tau accumulation would provide a better outcome measure compared to changes in cognition and that stratification based on multimodal data compared to β -amyloid alone would reduce the sample size required to detect a clinically meaningful change in tau accumulation. After training their algorithm on the ADNI cohort they then extended their machine learning approach to derive individualised trajectories of future pathological tau accumulation in local early AD patients and an independent sample of cognitively unimpaired individuals. The authors conclude that machine learning provides a robust approach for stratification and prognostication with translation impact for clinical trial design at asymptomatic and early stages of AD.

This is a novel study but I have difficulties with the design of the study. It would appear that all three cohorts contain a mixture of amyloid positive and amyloid negative cases. The CS cases are mainly amyloid negative while the EAD and AD groups are mainly, but not all, amyloid positive. Given this the machine learning is being trained on mixtures of preclinical, prodromal, and clinical AD mixed in with non-AD subjects. This makes interpretation of the findings difficult - the utility of machine learning for predicting outcome in terms of tau trajectory or cognitive deficit would be far clearer if all the cases selected were amyloid positive. A second issue is that an SUVR threshold of 1.1 is chosen for amyloid abnormality with FBP PET. This is low and is likely to lead to false positives - the figure provided suggests 1.2 would still separate AD from normal. Third, the cohort examined after training the machine learning algorithm have amyloid load measured with PiB PET and volumetry assessed 1.5 tesla MRI so are not represented by the training cohorts. Given all these issues it is difficult to accept the study conclusion.

Reviewer #4 (Remarks to the Author):

Giorgi & Kourtzi et al., examined the value of a prognostic index comprising beta-amyloid SUVRs, APOE e4 carrier status and gray matter density values in the medial temporal lobe in individuals who subsequently decline to prodromal or advanced stages of Alzheimer's Disease or remain stable over four years using machine learning and the ADNI2 cohort. They observe that the resulting index is well suited to predict faster future tau accumulation in an independent cohort (ADNI2/BACS) and that clinical definitions of cognitively normal adults or diagnosis of mild cognitive impairment is less predictive of tau accumulation compared to the modelled prognostic index. This is a very clever study design, and indeed a novel approach. Criticism and dampened enthusiasm exist however as the authors at times are not able to make clear what exactly their research question is, which is additionally complicated by the non-stringent use of preclinical AD, cognitively normal non-pathological again. A host of methodological and conceptual questions are needed to be addressed to further judge the suitability for publication. See those listed below:

The authors should define, what they mean with "interactions of beta-amyloid and tau pathology" specifically (line 53). As it is written thus far, it may indicate that dependency, despite the fact that these events occur particularly in early phases of AD may in fact be independent, but dependency occurs later in the disease stage. Please elaborate.

The authors state: Cognitively unimpaired individuals are classified as preclinical AD (line 66). That statement is not true and needs to be revised. Specifically, it should be stated somewhere in the research goal or aim of the study, which individuals are included with regards to biomarker status and how this is defined.

What do the authors mean, when describing that the clinical syndromic definitions are not sensitive to

the underlying AD pathology (line 70 ? Why us biomarker at all, if not specific to the pathology?)

The explanation of APOE e4 allele is highly speculative:

- the authors cite a mouse model that showed this association
- the authors then only cite reviews on the topic, neglecting to acknowledge evidence that showed that APOE carriers showed less elevated in vivo tau pathology compared to APOE non carriers (e.g., Mattson, Ossenkoppele et al., Alz Res Therapy, 2018)

The rationale to include both amyloid positives and negatives in the analysis is not very clear. Specifically, it appears that the authors suggested that amyloid biomarker positivity is a necessary condition for preclinical AD and only in such cases a prognostic index on future tau accumulation would make sense. Please elaborate.

What is meant with the sentence that "MRI data were used for quantitation of PET data"? (line 442)

As different beta-amyloid tracers were used in these different cohorts, it is imperative to put these on the same scale using the centiloid scale.

How would the SC and EAD multimodal scalar projection look like when amyloid was used as a binarized information rather than a continuous one. Although I very much appreciated the continuous approach, it would be more accessible to clinicians to evaluate a prognostic index based on binarized amyloid information.

To assess gray matter density values did the authors consider head size differences?

It is not clear how the threshold of amyloid positivity of SUVR = 1.1 was achieved, in lieu of the longitudinal processing of the baseline amyloid data in ADNI. The SUVR values are expected to be much lower than displayed in Figure 1 and amyloid positivity would be redefined. Please explain.

The author report that when deriving the scalar projection for the independent cohorts ADNI 3 (CN=72; MCI:43) and the BAC (CN=56)) the clinician-based diagnosis and the multimodal scalar projection show poor agreement. How do they interpret this finding? Given that the robustness of the diagnosis has not been evaluated over multiple time points in the independent cohort, I fail to see the additional value gained from this analysis.

Given that the multimodal scalar projection from the training sample rests on the cognitive change compared to cognitive stability over time, I find it surprising that SC and EAD individuals did not significantly vary in cognitive measures over time in the independent testing set. What is your explanation for this finding?

Is the rate of tau accumulation significantly different when using the multimodal scalar projection compared to the CN/MCI diagnosis? The authors just state that the stratification is better using SC and EAD but do not present evidence that sensitivity and specificity measures are indeed better for one over the other.

Although the clinical trial analysis is comprehensive and the "new kid on the block" in the biomarker research field, the impression remains that reducing meaningful cognitive decline in combination with the reduction of tau, will be more important for the design of clinical trials than just reducing tau pathology. Please elaborate on this point.

The authors claim that the multimodal modelling approach may be better than syndromic labels such as preclinical AD, MCI or AD to capture AD related pathology. An issue that arises here is that the current multimodal modelling approach rested on exactly those syndromic definitions of CN and

MCI/AD in the ADNI cohort in the first place to even get to the prognostic index. The authors should elaborate on this argument.

I think it should read diagnosis of dementia not "demented" (e.g., line 414).

Response to reviewers

Reviewer #1

Key aim of this study: predicting spatiotemporal tau accumulation in individuals potentially at risk of Alzheimer's disease. Predicting a map of future tau (PET) accumulation `_rates_` in regions of an individual's brain, *in vivo*.

The two primary audiences for this paper are "methods developers" (computational/statistical researchers) and "applications" researchers in Alzheimer's disease (e.g., patient-facing roles such as clinicians).

Methods: use supervised machine learning (previously published in [19]) to combine relevant features from an early Alzheimer's disease cohort (training data) into a scalar score through linear equations, which is thresholded into a data-driven cut-point to define biomarker-based subgroups of cognitively normal individuals (test data), then calculate group-level associations with future tau accumulation. This study is a straightforward application of the method developed by the authors in [19] to a new test sample, although the test data here is arguably more challenging because individuals are at an earlier stage of disease (cognitively normal here, MCI in [19]).

Confounders: analyses are adjusted for age, education, and gender norms (although, perhaps the authors mean sex rather than gender).

We have replaced 'gender' with 'sex'.

1. Data: from 2 or 3 cohorts (two phases of ADNI: GO/2 and 3; plus BACS). I say "2 or 3" because there is overlap between the ADNI-GO/2 and ADNI-3 phases. Were the "rollovers" into ADNI-3 excluded from the ADNI-3 portion of experiments? The authors do not make this clear.

We thank the reviewer for this suggestion regarding rollovers into ADNI-3. We had included a subset of rollovers that were used to train the original machine learning model. We have now removed these individuals from the ADNI2/GO training set and re-run all experiments. We have revised the Results section and the figures according to the new analyses. These new results are very similar to the results presented in the original submission and all conclusions remain the same.

2. Sample size is good for the training set (ADNI-GO/2 $n=488$), but less so for the test sets (ADNI-3 $n=115$, BACS $n=56$). I can't help but feel that there might be larger datasets available that would add confidence of generalisability of the findings. Bare minimum is to add a discussion on this limitation. How generalisable do the authors genuinely expect their findings to be? Justify, and quantify if possible.

Larger AD longitudinal data sets are available; however, data sets with longitudinal FTP-PET are much less common due to the tracer not being widely used prior to 2015. This problem is exacerbated when targeting cognitively normal individuals with longitudinal FTP. Additional validation data sets, in particular data with diverse disease aetiology, will be required to further test the generalisability and efficacy of our approach. Here, we chose to validate our model predictions in BACS because it comprises an asymptomatic sample rather than an AD research cohort. Thus, our results are not driven by the sampling characteristics of ADNI and may have real-world efficacy in a larger and more pathologically diverse sample. We now discuss this point in the revised manuscript.

In particular the Discussion section (pg 21) writes: *‘Finally, we validated our model—that was trained on data from an AD disease-specific cohort—by testing predictions in an independent cognitively normal sample. This provides evidence that our results are not driven by the sampling characteristics of ADNI, suggesting generalisability of our modelling approach to more diverse groups. Our asymptomatic sample size was limited, as publicly available data from cognitively normal participants with longitudinal FTP-PET are scarce. Larger samples with longitudinal data would increase the generalizability and validate the real-world efficacy of our approach.’*

3. Findings: group-level associations between the individual scalar scores and future tau accumulation rates. While these are individual predictions, the results are analysed statistically at the group level, which does not align particularly well with the individual-level aims of the study. The results are presented clearly, but the implementation of these results in (clinical) practice is unclear (Line 304 "strong clinical relevance" might be overstating it). The model achieves decent classification performance but, in particular, the outlier cases are not discussed, i.e., where the individual predictions diverge considerably from the average – how are these model predictions converted to clinical decisions for an individual? What are the implications/ramifications for incorrect/poor decisions? Is the model itself to blame? How suitable is a linear model for a disease that likely plays out in a nonlinear fashion? Perhaps the model inputs are inadequate (more biomarkers?)?

The manuscript reports a) categorical associations separating individuals into either Clinically Stable or Clinically Declining b) explicit predictions on the individual level for Clinically Declining individuals. We adopt this in accordance with the recent NIA-AA 2018 framework for AD as a biological construct categorised along a continuum. First, we presuppose that our sample is composed of two separate populations (1) those without AD pathology (i.e. Clinically Stable) and (2) those with AD pathology (Clinically Declining + Alzheimer’s Clinical Syndrome). Therefore, we introduce an approach to unmix these populations using a threshold along a single scalar derived from our machine learning approach (i.e. GMLVQ-scalar projection). Following unmixing of the sample, we show that the continuous scalar value for the subpopulation with AD pathology (i.e. Clinically Declining) captures individual-level information. We test and validate this by predicting individualised rates of tau accumulation across the cortex. We present predictions on the individual level in Figures 7 and 8 and calculate the goodness of fit of these individual predictions as the shared variance (R^2) between predicted and real tau accumulation for each individual classified as Clinically Declining.

The focus of our analyses is early-stage stratification for patients with AD pathology. In particular, we developed a modelling approach that is sensitive and specific to the AD topography of tau accumulation. The clinical relevance of this work is mainly related to the design of intervention trials that target tau accumulation. We have now clarified this throughout the revised manuscript

Our modelling approach combines three biomarkers in a linear way; as a result the model has limited freedom to make poor decisions due to its architecture. The reviewer is correct about the non-linear fashion of AD pathological change. The proposed AD cascade suggests that tau

accumulation follows a sigmoidal shape over time, therefore the hypothesised rate of change will follow an inverted U shape (Jack et al., 2013, 2010) with the rate of tau accumulation increasing rapidly prior to severe cognitive impairment (i.e. AD dementia). In our work, we investigated the relationship of the scalar projection with the rate of tau accumulation for individuals who are cognitively normal or MCI. For this sample at earlier AD stages, tau accumulation over time (i.e. rate of tau accumulation) is likely to be prior to the plateau of tau accumulation (i.e. within the linear portion of the rate of tau accumulation curve) that is hypothesised for later stage AD (Jack et al., 2013, 2010). Thus, we tested whether a linear model captures the relationship between the scalar projection and individual rate of future tau accumulation for our early-stage AD sample. Further, we derived the rate of future tau accumulation using the linear least squares fit of regional FTP-PET over time. Given the sampling frequency of FTP-PET for our sample it is difficult to fit a higher order (i.e. quadratic or spline) rate of tau accumulation as only 5 individuals in our sample had more than 3 visits. For the ADNI cohort, the number of follow-up FTP-PET scans n (2 FTP-PET scans) =93, n (3 FTP-PET scans) =17, n (4 FTP-PET scans) =5. For the BACS cohort, the number of follow-up FTP-PET scans n (2 FTP-PET scans) =37, n (3 FTP-PET scans) =19.

For these samples only a small number of outliers were identified (n=2 ADNI 3, n=1 BACS). These individuals could not be specified within our linear prediction framework, particularly for the periods of observation we used, which would be typical of a clinical trial. These outliers fall far from the Clinically Declining mean and have high amyloid burden, medial temporal atrophy and likely APOE4 positivity; therefore, these individuals are closer to individuals with late-stage AD (i.e. Alzheimer's Clinical Syndrome) (Figure 4). These outliers highlight that the clinical syndromic definitions are not sensitive to the severity of AD related pathology as they are either CN or MCI (ADNI 3). Within the context of a clinical trial these individuals have more advanced AD pathology than the majority of early AD individuals and therefore may not be ideal candidates for enrolment into an intervention trial targeting tau accumulation in early-stage AD. A carefully constructed and constrained complex fit may be preferred when predicting future tau accumulation for a sample including cognitively normal, MCI and probable AD dementia individuals. We have now included a discussion on these points in the revised manuscript.

In particular the Discussion section (Pg 20) now writes: *'Fourth, our linear modelling approach, focusing on data from cognitively normal and MCI individuals, captures the linear portion of the hypothesised biomarker trajectory for tau accumulation. This model fits the majority of individual predictions with only a small number of outlier cases due to a high scalar projection (ADNI 3: n=2, BACS: n=1). These outliers represent Clinically Declining individuals with more advanced AD pathology who may not be ideal candidates for clinical interventions targeting early AD pathological changes. Further model development including late AD samples with longitudinal FTP-PET is required to cover the full AD spectrum and accurately predict non-linear trajectories of tau accumulation.'*

Additional biomarkers can be incorporated into the model, e.g. more biomarkers from different PET tracers that include hypometabolism or neuroinflammatory responses i.e. FDG-PET,

PK11195-PET. Further, additional MRI measures can be included in the model as well as data from cognitive measures. Here, we focused on specific well-studied biomarkers (i.e. A β , medial temporal grey matter density and APOE 4) to make robust predictions, as evidenced by the generalisability of our results across samples with different A β tracers (i.e. FBP in ADNI and PiB in BACS) and MRI field strengths. However, additional biomarkers may improve model performance and economic viability. We discuss this point further in the manuscript.

In particular the Discussion section (Pg 19) now writes: ‘*Second, we focussed on specific well-studied biomarkers (i.e. A β , medial temporal grey matter density and APOE 4)—rather than interrogating the predictive power of a wider range of markers—to make robust predictions, as evidenced by the generalisability of our results across samples with different A β tracers (i.e. FBP in ADNI and PiB in BACS) and MRI field strength. Our previous work—consistent with other studies—has shown that neuropsychological data are predictive of MCI progression to dementia due to AD^{36,51–56}, with improved predictive performance when including biological information^{36,57–59}. Further, recent studies have shown that blood based AD biomarkers have substantial predictive power in modelling AD trajectories^{60,61}. Extending our modelling approach to integrate less-costly (i.e. plasma) and non-invasive (i.e. cognitive) data has strong potential for stratification at asymptomatic or early stages of AD.*’

4. That being said, the results around Figure 5 are okay because clinical trials are indeed assessed at the group level. The particular novelty here is in using a spatiotemporal biomarker (admittedly projected into a linear decision space) rather than a simple scalar biomarker. The proof of the pudding would be to apply this in an actual clinical trial (obviously a longer-term future aim!).

We thank the reviewer for acknowledging the novelty of the spatiotemporal biomarker derived in our study. We look forward to the opportunity to test our approaches in a clinical trial.

5. In the discussion (line 297) the authors statement about multivariate vs univariate is well-known and not novel. What's missing here is a solid baseline/benchmark model for direct comparison (of apples with apples, as one might say). For example, comparing performance of the classifier here with that of a different multivariate classifier trained on the same multivariate data.

It is generally accepted that multimodal predictive models are preferred when making classifications in AD. However, a recent study suggests that amyloid status alone may be the optimal stratification for enrolment into a clinical trial specifically targeting future tau accumulation (Jack et al., 2020). Our modelling approach captures predictive covariance (via the metric tensor) highlighting that the multivariate integration of the same predictors presented in the Jack 2020 paper provides a benefit over amyloid status alone. We have now further clarified this point in the revised Discussion section.

In particular the text writes (Pg 18-19): ‘*Finally, our multimodal prognostic index—compared to A β status alone—reduces the sample size required to observe a clinically meaningful change in the stereotypical pattern of pathological tau accumulation by 44%, extending recent work showing that A β status is an optimal independent biomarker for stratification based on future*

*tau accumulation*⁴³. Our modelling approach captures predictive disease related covariance in biomarkers, demonstrating the benefits of using a) multivariate biomarker information compared to A β status alone when predicting future tau accumulation, b) machine learning to model interactive pathophysiological processes in AD. These benefits have been previously demonstrated in the context of predicting future changes in cognition⁴⁴⁻⁴⁹. In particular, previous studies have shown that: a) grey matter atrophy and cortical A β burden relate to separable patterns of future cognitive decline^{44,45,48,49}, b) longitudinal changes in tau relate to cognitive decline in preclinical AD¹⁹. Thus, our results support the benefit of combining continuous values of A β and medial temporal grey matter density for prognostication, demonstrating the potential of our approach to inform the design of clinical trials targeting pathophysiological changes at the earliest stages of AD.'

Direct comparison of diverse machine learning approaches remains challenging as cross-validation methodology, sample sizes and sample heterogeneity have a significant effect on model performance metrics. Our modelling approach focuses on both continuous predictions and discrete classification of individuals. Thus, our predictions are not directly comparable to those of models performing only binary classifications based on syndromic diagnosis. Further, our multivariate approach (GMLVQ-scalar projection) extracts continuous predictions from a discrete class of classifiers (GMLVQ) enabling us to both classify and continuously stage individuals. To compare performance of our modelling approach we compared our multimodal baseline stratification with two common approaches to stratify individuals in the early stages of AD: 1) syndromic definitions and 2) amyloid status. This comparison focusses on the data used to perform stratification (i.e. biological vs. syndromic and multimodal vs. unimodal) and demonstrates that the multimodal baseline stratification using biological data has higher predictive power than these standard baseline classification schemes. We have now indicated that these schemes serve as model benchmarks in Figure 1.

We further discuss this point in the revised manuscript.

In particular the Discussion section (pg 19-20) writes: *'Third, our work focussed on the GMLVQ-scalar projection machine learning framework. Machine learning-guided modelling in AD is a rapidly expanding field with most studies investigating binary changes in syndromic labels from baseline (for review: ^{27,62-65}). Direct comparison of diverse machine learning approaches remains challenging, as cross-validation methodology, sample sizes and sample heterogeneity have a significant effect on model performance metrics (i.e. accuracy or receiver operator characteristics) ⁶⁶. Prediction challenges across a range of different tasks offer a more unbiased approach for determining the efficacy of prediction models (e.g. TADPOLE ⁶⁷). However, as our modelling approach makes continuous predictions and stages individuals using an adapted discrete classification framework (GMLVQ), it is not directly comparable to binary classification models that make predictions based on syndromic diagnosis. To validate our approach, we compared our multimodal classification against two standard stratification approaches for AD; a) baseline syndromic labels (i.e. CN and MCI) and, b) unimodal stratification by A β positivity. This allows us to draw conclusions that are relevant for patient stratification and the design of clinical trials; that is, a) determining treatment groups based*

on syndromic labels may result in variability in pathological state across groups and, b) unimodal stratification based on A β alone is underpowered compared to the multimodal stratification derived based on our prognostic index.'

6. Summary: I find the results interesting on some levels (mostly in terms of the application), if not especially novel (particularly methodologically). Additionally, I have minor question marks over the data (small sample size in test set; potential train/test overlap in ADNI-GO/2 and ADNI-3).

We thank the reviewer for their interest in our results and hope that we have successfully addressed their concerns with regards to the data used.

Detailed list of additional thoughts/changes, in no particular order:
7. Please check that no individuals from ADNI-3 were involved in the training set (ADNI-GO/2). If they were, remove them from the test set and rerun the experiments. Clarify in manuscript.

We have removed the roll-overs from the data used to train the scalar projection model.

We chose to retain the rollovers in the ADNI 3 tau sample and remove them from the ADNI2/GO training sample for the GMLVQ scalar projection model. This is due to the relatively smaller sample size of individuals with longitudinal FTP-PET. As stated above all results remain the same.

8. Discuss individual-level applicability of findings, with particular regard to outliers and clinical decision making.

Please refer to revised text and response to comment 3 shown above.

9. Discuss generalisability of findings, with regard to the relatively small sample sizes in the test sets.

Please refer to revised text and response in comment 2 shown above.

10. Change the "demented" terminology to "probable AD", or simply "AD".

We have now changed the terminology to Alzheimer's clinical syndrome consistent with current recommendations for clinical diagnosis without biomarkers.

11. Add a brief discussion of differences/similarities between FBP/FTP/PIB PET tracers. Specifically, in relation to the findings. Perhaps off-target binding is a confounder in some regions of interest (perhaps not, which is also worth remarking in the manuscript).

We harmonised the different amyloid tracers using the centiloid approach. Using the centiloid approach Klunk et al show that FBP and PiB amyloid tracers are reasonably interchangeable (Klunk et al., 2015). In addition, off target binding within the white matter is consistently observed across FBP and PiB tracers. The analysis pipelines to derive FBP SUVR and PiB DVR are built to account for this; therefore, we do not anticipate that this affects our results.

We believe that it is unlikely that our results are confounded by FTP off target binding, as FTP PET measures largely correspond to pathological spreading of AD related tau (Schöll et al., 2016) and we used an unbiased selection of ROIs. In particular, we made no a priori selections of composite regions containing structures with high off target binding (i.e. striatum and hippocampi). Further, a study by Lowe et al. 2019 highlights that the association between antemortem FTP uptake and NFT load postmortem is highly similar with or without partial volume correction (Lowe et al., 2020). In light of this we did not correct for partial volume effects.

To account for potential off target binding in healthy cognitively normal individuals we used a reference region from eroded subcortical white matter regions. Using uptake values from three subcortical ROIs (Putamen Thalamus ChPlex) Baker et al showed that 60% of the variation in non-partial volume corrected global FTP uptake in healthy amyloid negative cognitively normal individuals is removed (Baker et al., 2019). Therefore, we favoured a subcortical reference region over the standard cross sectional reference region of the inferior cerebellar grey matter to account for potential off target binding that may affect cognitively normal non-AD individuals. We now address these points in the revised Methods section (pg 29-30).

In particular the text now writes: *'To quantify cortical amyloid burden we utilised multiple PET tracers. To derive a robust scalar metric for predictions we first harmonised A β PET values using the centiloid approach. Using this approach it has been shown that FBP and PiB amyloid tracers are interchangeable once scaled linearly onto a common scale (i.e. centiloids)⁷². . . . To model patient trajectories of future tau accumulation we used longitudinal FTP-PET. The association between antemortem FTP uptake and neurofibrillary tangles load post-mortem has been shown previously⁷⁴. However, FTP retention is associated with significant off target binding. To mitigate this, we used a reference region from eroded subcortical white matter regions. Previous work has shown that FTP-PET uptake in subcortical regions accounts for 60% of the variation global FTP uptake in healthy amyloid negative cognitively normal individuals⁷⁵.*

12. Line 458. Please start this paragraph by mentioning SPM, so it's clear.

We have now included this. In particular the Methods (Pg 26) now write: *'All structural MRI pre-processing was performed using Statistical Parametric Mapping 12 (<http://www.fil.ion.ucl.ac.uk/spm/>).*

13. Discuss clinical utility of predicting _rates_ of tau accumulation versus raw tau burden. Two individuals could both be accumulating slowly at the same rate, but have completely different tau burden - clearly not the same. We thank the reviewer for this suggestion. We discuss this point in the revised manuscript (pg 4).

In particular the revised text writes: *'Further, reducing cognitive decline is often considered as a primary outcome measure for interventions targeting asymptomatic AD populations¹⁸;*

yet, targeting upstream pathological changes has potential to benefit clinical translation. In particular, tau is strongly linked to both future neurodegeneration and to cognitive decline¹⁹, making reduction of future tau accumulation a potential intervention target, as evidenced by anti tau drugs entering the clinical trial pipeline^{20,21}. Recent evidence suggests that there are consistent patterns of tau spread (measured in-vivo by longitudinal FTP-PET) across early AD (i.e. prior to severe cognitive impairment) cohorts²²⁻²⁴. That is, tau initially accumulates within the temporal cortex then spreads to the superior and medial regions of the parietal cortex prior to severe cognitive impairment²²⁻²⁴. Thus, slowing rates of tau accumulation within these key regions, similar to slowing rates of cognitive decline, could be a meaningful biomarker outcome. Further, as evidenced by failures of anti-amyloid interventions to halt clinical decline, simply clearing already deposited proteinopathies may be insufficient to stop downstream events²¹. Therefore, targeting individuals with the highest risk of depositing tau rather than those burdened with tau, may increase the likelihood of successfully modifying downstream clinical decline.'

Reviewer #2

Thank you for giving me the opportunity to review this very interesting paper. The authors train a series of algorithms with the goal of predicting individualised future regional tau accumulation. This is an important effort, in-part due to circumventing issues in binary categorical classifications often employed. Using a novel GMLVQ technique, early AD is modelled in a multi-dimensional space, within which subgroups are represented by a prototype. Distance from the prototype can be used to represent disease on a continuum (scalar projection), and given new instances with the same feature vector their scalar projection can be calculated. Thresholding the scalar projection enables classification of discrete groups.

For the current paper, first a GMLVQ is trained to model a stable condition (SC) and early AD (EAD) based on the multivariate relationship between amyloid, medial temporal grey matter density and APOE 4. Thus, the scalar projection models prognosis on a continuum between the SC prototype and EAD prototype based on the input variables. Second, the scalar projections for novel data are calculated, and threshold used to classify as EAD or not. For those classified as EAD, brain regions accumulating tau were identified, and the future rate of tau accumulation was predicted for each region using regions of interest and the scalar projection. Finally, future rate of tau accumulation was predicted in an unseen dataset.

Results detail variance explained globally and in regions of interest of future tau accumulation, only found for the EAD group and not SC, supporting this novel approach to stratification that is based on underpinning biology, and characterised using a novel machine learning approach using the scalar projection (or prognostic index). Interestingly for a hypothetical clinical trial sample size required could be reduced by 35% to detect a 25% reduction in future tau accumulation, compared to a 25% reduction in future cognitive decline. A reduction of 30% was found when using EAD classification compared to amyloid positivity alone.

Particular strengths of the paper are the use of multiple datasets to train and test, including out of sample testing; use of resampling methods to create a balanced dataset to train the GMLVQ scalar projection, thus not impacted by confounding variables, and the translational impact of using predictions to decrease required sample size for a hypothetical clinical trial.

I found the methods and results to be exciting. The machine learning and statistical approaches utilised are, as the authors state, robust and transparent. The discussion is very well written and places the findings in context of known research, including a good discussion of the limitations. I think the manuscript could be improved by building on the introduction, which would aid in the flow of the paper overall, and by making clearer the groups and subgroups of interest throughout, perhaps with a figure. I go into these points in more detail below, along with other minor points.

We thank the reviewer for their suggestion of a figure to help readability and their interest in the results and methods presented. We have now included a figure with the groups and analysis that were performed (Figure 1). We have also rewritten the introduction to clarify our approach and motivation of the study.

1) I was confused by the groups and subgroups listed within the paper. This begins in the introduction, line 65, which states “In this framework, evidence of A β and pathological tau accumulation is sufficient to establish the diagnosis of AD. Cognitively unimpaired individuals (CN: cognitively normal) are classified as preclinical AD”. Line 81 states “...classifies and stages early AD (i.e. CN and MCI)” and line 86 “early AD (i.e. A β positive individuals who are cognitively unimpaired...)”. I found this explanation a little blunt - we do know that not all of those with AD pathology will develop clinical AD (e.g. Bennett et al., 2006) - but understand that this is a definition based on evidence of AD pathology, as opposed to a clinical definition. However, I think the introduction could be built upon to better describe this focus on pathology/biology, using terms elsewhere in the paper, which would also aid in the flow. The term ‘asymptomatic’ is used readily elsewhere, as is reference to CN and MCI as syndromic labels – given the interesting results regarding the mismatch between the biological stratification in the paper and these syndromic labels, I feel bringing this into the introduction earlier in the paper would aid the reader.

We thank the reviewer for this suggestion. We have now re-written the introduction to re-focus the paper and included the suggested figure. In addition, we have made a clear distinction between the baseline syndromic definitions (CN and MCI), longitudinal syndromic definitions used as classes to build the machine learning model (i.e. Clinically Stable and Clinically Declining) and how these relate to the underlying biology.

2) Overall, I think the paper (barring the discussion) is complicated by heavy use of acronyms for groups, which hinders readability. I think a figure detailing the different groups utilised, and for which analyses, would add clarity. Throughout, even after acronyms have been introduced, sometimes full groups labels are used and sometimes acronyms with no consistency, again reducing readability.

We thank the reviewer for these suggestions, in the revised manuscript we have tried to be consistent in our use of terminology and reduce the use of acronyms. In addition, we have included the suggested figure (Figure 1) for clarity.

3) Relatedly, the ‘Demented’ group is almost a surprise addition, only briefly mentioned in the methodology, but features quite heavily in results and so could be brought in earlier as a group of interest.

We now introduce this group earlier and more clearly in the results section.

In particular the results (pg 7) writes: *‘An additional 181 individuals diagnosed with AD (which we refer to as Alzheimer’s Clinical Syndrome) were included to cover the full spectrum of longitudinal AD diagnoses. These individuals received a stable diagnosis of AD dementia across follow-ups and are used as a reference population of late-stage AD.’*

4) And, my confusion was further compounded by the Supplementary Materials section GMLVQ – Scalar Projection, in which notation returns to progressive/ stable. I wonder if using EAD throughout would improve readability.

We have now amended the supplementary materials to carry through the terminology of clinically stable and clinically declining

5) I felt a brief methodological breakdown of how APOE 4 positivity was defined was missing e.g. does the presence of one or two alleles = positivity?

APOE 4 positivity was defined as the presence of one or two alleles. We have now included this in the Results (Pg 6) section of the manuscript: '*APOE 4 genotype (presence of one or two alleles)*'

6) Figure 2 caption could be condensed by stating (for example) 'the learnt probabilistic boundary that separates SC from EAD is indicated by a dashed vertical or horizontal line', prior to the a, b and c descriptions, rather than repeating this information.

We have now amended the figure caption. In particular the caption now writes:

'Figure 3: Relationship of scalar projection with biological predictors. ADNI2/GO sample: Blue dots indicate individuals in the Clinically Stable group, red dots indicate individuals in the Clinically Declining group the dashed vertical line indicates the learnt probabilistic boundary that separates Clinically Stable from Clinically Declining. a. Relationship of scalar projection with FBP centiloids ($A\beta$), the solid horizontal line indicates the ADNI threshold for $A\beta$ positivity ($SUVR=1.11$). b. relationship of scalar projection with medial temporal grey matter density. c. relationship of scalar projection with APOE 4 status.'

7) Figure 3a uses a dashed black line to represent the threshold for EAD, are the dashed black lines on the box plots for each group related to the threshold information? If not perhaps replace them with solid lines to avoid confusion.

We have now amended this figure accordingly.

Typos

We thank the reviewer for their considerate revision of our text and have amended all the typos that were identified.

8) Line 57 bracket missing after (for reviews...)

We have now amended the introduction accordingly.

9) Line 156 typo "that" should be "than"

We have now amended the results accordingly.

10) Line 242 missing a space after 6b).

We have now amended the results accordingly.

11) Line 312 – states a reduction in sample size of 46% based on prognostic index alone vs $A\beta$ status alone. I could not seem to find this analysis in the results section, only for EAD classification vs $A\beta$ status which was reported as 47% (line 223)? Is the same analysis?

We have now clarified this in the Discussion, accurately reporting the reduction in sample size based on the revised analysis. In particular the Discussion (pg 18) now writes: *‘Finally, our multimodal prognostic index—compared to A β status alone—reduces the sample size required to observe a clinically meaningful change in the stereotypical pattern of pathological tau accumulation by 44%.’*

12) Line 469 typo “RFE))” should be RFE)

We have now amended the methods accordingly.

13) Line 425 “cognition for individuals diagnosed as MCI”. This may be personal preference but I would replace “as” with “with”.

We have now amended the methods accordingly.

14) Line 935 typo “ii)an” should be ii) an.

We have now amended the figure caption for figure 7 accordingly.

15) Line 946 typo “(EAD)in” should be (EAD) in.

This text has now been removed from the figure caption for figure 8.

16) Line 952 typo “6c).” should be 6c.)

We have now amended the figure caption for figure 8 accordingly.

Reviewer #3

The authors report findings of applying a machine learning approach to MRI volumetry and amyloid PET data and apoE4 carriage status to predict future tau accumulation measured with flortaucipir (FTP) PET and cognitive decline obtained for cohorts from the ADNI database. The subjects comprised established Alzheimer dementia (AD), early Alzheimer's disease (EAD) and normal stable cases (CS). The machine learning derived a predictive index that stratified individuals based on their future pathological tau accumulation. The hypotheses were that future tau accumulation would provide a better outcome measure compared to changes in cognition and that stratification based on multimodal data compared to β -amyloid alone would reduce the sample size required to detect a clinically meaningful change in tau accumulation. After training their algorithm on the ADNI cohort they then extended their machine learning approach to derive individualised trajectories of future pathological tau accumulation in local early AD patients and an independent sample of cognitively unimpaired individuals. The authors conclude that machine learning provides a robust approach for stratification and prognostication with translation impact for clinical trial design at asymptomatic and early stages of AD.

1. This is a novel study but I have difficulties with the design of the study.

We thank the reviewer for acknowledging the novelty of our study. We have now re-written the introduction to discuss the aim and structure of the study, emphasising bridging the gap between the biological and syndromic frameworks of AD.

2. It would appear that all three cohorts contain a mixture of amyloid positive and amyloid negative cases. The CS cases are mainly amyloid negative while the EAD and AD groups are mainly, but not all, amyloid positive. Given this the machine learning is being trained on mixtures of preclinical, prodromal, and clinical AD mixed in with non-AD subjects. This makes interpretation of the findings difficult - the utility of machine learning for predicting outcome in terms of tau trajectory or cognitive deficit would be far clearer if all the cases selected were amyloid positive.

We agree that an AD predictive model defined only by biomarkers should include amyloid positivity. However, the aim of our study was to build a predictive model using labels based on longitudinal clinical syndromic definitions that are blind to biomarker status (i.e. amyloid positivity), as this best reflects the clinical situation in most cases. In particular, our study design aims to extract a predictive continuous biomarker from a model trained to predict longitudinal decline in clinical diagnosis. Using this continuous multimodal biomarker we show that predicting future pathological changes (i.e. future rate of tau accumulation) is improved over A β positivity defined using a strict threshold.

Further, determining amyloid positivity using an SUVR threshold is inherently probabilistic, therefore some sub threshold individuals theoretically will be amyloid positive. In light of this, including slightly subthreshold amyloid negative individuals is preferable for modelling as it ensures that our model is not overly sensitive to the threshold of amyloid positivity.

Finally, among the sample of amyloid negative individuals in the Clinically Declining group (n=26), there are individuals well below the SUVR threshold of positivity and therefore are likely afflicted by non-AD pathology. The inclusion of these individuals in the Clinically Declining group is due to an inherently noisy syndromic clinical definition. Our results demonstrate that these syndromic labels are poor descriptors of baseline biology and future tau accumulation. Our modelling approach is not biased by the inclusion of these individuals, thanks to its low number of degrees of freedom, in contrast to highly parameterised machine learning models that may be negatively affected by this noise in outcome label (i.e. target uncertainty). Please see also response to reviewer 4 comment 16 for more detail on how the model extracts a sensitive prognostic marker from noisy labels. For these reasons, we chose to include these individuals, as removing them would diverge from our study design which focusses on training a model using noisy clinical diagnoses to derive a sensitive and specific multimodal biomarker. We have now clarified this point in the revised manuscript.

In particular, the Discussion (pg 21) writes: *‘Despite these potential limitations, our machine learning approach successfully capitalises on longitudinal data to make sensitive and specific predictions of early-stage AD trajectories based on baseline pathophysiology. Further, our approach provides two key advances: a) combines multimodal continuous biological measures to capture trajectories for individuals who may be on the threshold of unimodal biomarker positivity but likely to follow AD related trajectories¹⁷, b) harmonises longitudinal data collected using syndromic diagnostic criteria^{6,7} (e.g ADNI²⁶) by combining continuous biomarkers into a biologically informative prognostic index in an interpretable and clinically meaningful way.’*

Further, the text in the Supplementary Methods (pg 13) writes: *‘The GMLVQ-scalar projection approach addresses three inherent issues with prognostic models in AD research. 1.) The GMLVQ-scalar projection approach is able to account for target uncertainty. This is achieved by having a model that learns a low-parameter task-dependent scaling matrix (metric tensor), and only two locations in hyperdimensional space (prototypes). These univariate (diagonal) and multivariate (off diagonal) relationships are learnt to separate the two classes (Clinically Stable vs Clinically Declining) as best possible from a global perspective (vs local metric tensors) without over constraining the predictor data. Similarly, by defining only one position in this learnt space that best determines if a person is Clinically Stable or Clinically Declining, the model must ignore subtle differences for any given target, learning a broad location that best describes Clinically Stable / Clinically Declining populations. By not over constraining the data, this type of model will not be sensitive enough to overfit based on subtle difference in diagnostic criteria.’*

3. A second issue is that an SUVR threshold of 1.1 is chosen for amyloid abnormality with FBP PET. This is low and is likely to lead to false positives - the figure provided suggests 1.2 would still separate AD from normal.

The threshold of SUVR = 1.1 is the standard positivity threshold used in ADNI and was chosen a priori, rather than based on the data used in this study. This value equates to a centiloid value approximately 20-25 (depending on processing method), which is also a standard published

range for amyloid positivity as documented on the ADNI website <http://adni.loni.usc.edu> and elsewhere. We now reference this within the manuscript (Joshi et al., 2012; Navitsky et al., 2018).

To address the differences in amyloid PET tracers (FBP & PiB), we have now converted both cohorts to centiloids and re-run all analyses (Klunk et al., 2015). The results are highly similar to those reported in the previous submission and all conclusions remain the same. We previously addressed the differences in amyloid tracers in our original submission by variance normalising within each cohort by the mean and standard deviation of cognitively unimpaired individuals. We have now replaced this analysis with conversion to centiloids.

We believe that it is unlikely that our results are confounded by MRI field strength between samples. Within the BACS sample 13 of the 56 individuals had 1.5T MRI from which the MTL grey matter density value was estimated. To determine the effects of MRI field strength, we investigated the contribution of MTL grey matter density to the multivariate and pathologically predictive scalar projection. First, we tested if the MTL value captures similar variance of the scalar projection in BACS as the MTL value in the ADNI2/GO training sample. Supplementary Figure 1a (below), shows that the MTL grey matter score and the scalar projection have a shared variance of $R^2=34\%$. This value is similar to the shared variance of MTL grey matter density and the scalar projection in ADNI2/GO training sample ($R^2=40\%$). Next, we calculated the error of the fit for each participant and compared these errors within the BACS sample using 1.5T or 3.T. Supplementary Figure 1b (below) shows that the error of the fit is distributed evenly about zero highlighting that there doesn't appear to be any systematic effect of MRI field strength. Finally, we performed a two sample t-test between the error of the fit using for MTL grey matter density measured using 1.5T MRI and 3T ($t(54)=0.394$, $p=0.695$). As there was no significant difference between the two distributions, it is unlikely that any systematic differences were introduced by using 1.5T MRI data. We consider the robust nature of these field-independent findings to be a strength of our method. We address these points further in the revised manuscript.

In particular the Results section (pg 9) writes: *'Further, we tested whether difference in MRI field strength for the BACS sample introduced a systematic bias to the multimodal scalar projection. A two sample t-test comparing the residual of the fit of the medial temporal grey matter density and the scalar projection showed no significant differences between 1.5T and 3T MRI in BACS ($t(54)=0.394$, $p=0.695$) (Supplementary Figure 1), suggesting that our multimodal approach is robust across differences in MRI acquisition.'*

The Discussion section (Pg 19) writes: *'Second, we focussed on specific well-studied biomarkers (i.e. $A\beta$, medial temporal grey matter density and APOE 4)—rather than interrogating the predictive power of a wider range of markers—to make robust predictions, as evidenced by the generalisability of our results across samples with different $A\beta$ tracers (i.e. FBP in ADNI and PiB in BACS) and MRI field strength.'*

The Methods section writes: ‘To quantify cortical amyloid burden we utilised multiple PET tracers. To derive a robust scalar metric for predictions we first harmonised A β PET values using the centiloid approach. Using this approach it has been shown that FBP and PiB amyloid tracers are interchangeable once scaled linearly onto a common scale (i.e. centiloids) ⁴⁰.

a. Effect of field strength on MTL contribution to Scalar Projection

b. Differences in Field Strength

Supplementary Figure 1. Effect of MRI field strength on scalar projection (BACS) a. highlights that there is a contribution of the MTL grey matter density score to the multimodal scalar projection. The black line indicates the linear best fit of these two variables. Black dots represent individuals who are scanned using 3T MRI and red dots are individuals who are scanned using 1.5T MRI. **b.** Shows the residual of the fit of the MTL grey matter density score and the scalar projection for individuals scanned on 1.5T vs 3T MRI. The redline is the median

of the fit, the solid black box represents the 25th to 75th percentile and the dashed black lines represents the range of the data.

4. Third, the cohort examined after training the machine learning algorithm have amyloid load measured with PiB PET and volumetry assessed 1.5 tesla MRI so are not represented by the training cohorts. Given all these issues it is difficult to accept the study conclusion.

We appreciate that variability in data across training and validation groups is a challenge for machine learning modelling. However, we provide evidence that our approach makes robust and meaningful predictions in a sample that has slight variations in how data are collected i.e. BACS vs ADNI 3 (1.5T and PiB vs. FBP). We believe that this variability strengthens rather than biases our results for the following main reasons. Previous work comparing the reliability of different PET tracers has shown that PiB and FBP can be used interchangeably when the appropriate scaling has been conducted (Klunk et al., 2015). Further, we show that there are no systematic differences between 1.5T and 3T data in the relationship of MTL density score to our prognostic index. Finally, this variability is expected to add noise to the results, biasing our findings towards the null rather than resulting in false positive findings, thus strengthening the predictive power of our approach.

Reviewer #4

Giorgio & Kourtzi et al., examined the value of a prognostic index comprising beta-amyloid SUVRs, APOE e4 carrier status and gray matter density values in the medial temporal lobe in individuals who subsequently decline to prodromal or advanced stages of Alzheimer's Disease or remain stable over four years using machine learning and the ADNI2 cohort. They observe that the resulting index is well suited to predict faster future tau accumulation an independent cohort (ADNI2/BACS) and that clinical definitions of cognitively normal adults or diagnosis of mild cognitive impairment is less predictive of tau accumulation compared to the modelled prognostic index. This is a very clever study design, and indeed a novel approach.

1. Criticism and dampened enthusiasm exist however as the authors at times are not able to make clear what exactly their research question is, which is additionally is complicated by the non-stringent use of preclinical AD, cognitively normal non-pathological again. We thank the reviewer for this suggestion. We have now clarified the research question in our revised introduction. We have also taken care to have consistent terminology dissociating between syndromic labels and biological classifications.

2. A host of methodological and conceptional question are needed to be addressed to further judge the suitability for publication See those listen below: The authors should define, what they mean with "interactions of beta-amyloid and tau pathology" specifically (line 53). As it is written thus far, it may indicate that dependency, despite the fact that these events occur particularly in early phases of AD may in fact be independent, but dependency occurs later in the disease stage. Please elaborate. We have now amended the introduction, removing this statement which was not relevant to our approach.

3. The authors state: Cognitively unimpaired individuals are classified as preclinical AD (line 66). That statement is not true and need to be revised. Specifically, it should be stated somewhere in the research goal or aim of the study, which individuals are included with regards to biomarker status and how this is defined. We have now included the aims of our study in the introduction and explained that we did not use biomarker status for assigning individuals to the training sample; instead, groups were assigned based on longitudinal changes in clinical syndromic labels.

4. What do the authors mean, when describing that the clinical syndromic definitions are not sensitive to the underlying AD pathology (line 70 ? Why us biomarker at all, if not specific to the pathology?
By "clinical syndrome" we mean a diagnosis of normal, MCI, or probable AD dementia, traditionally deployed without taking into account biomarker status -note that the current research framework is targeted to research and most clinicians do not widely employ biomarkers-. Previous work interrogating the sensitivity and specificity of these clinical

syndromes, i.e. a diagnosis of probable AD ante mortem, shows that diagnoses of probable AD have a sensitivity to postmortem AD pathology between 70.9% and 87.3% and a specificity between 44.3% and 70.8% (Beach et al., 2012). In contrast, PET biomarkers of β -amyloid and tau have been shown to have high sensitivity and specificity (over 90%) for the relevant brain pathology.

5. The explanation of APOE ϵ 4 allele is highly speculative:

- the authors cite a mouse model that showed this association
- the authors then only cite reviews on the topic, neglecting to acknowledge evidence that showed that APOE carriers showed less elevated in vivo tau pathology compared to APOE non carriers (e.g., Mattson, Ossenkopp et al., Alz Res Therapy, 2018)

We agree that the relationship between APOE and tau is not straightforward. We did not include APOE in our algorithm simply because of its relationship to tau, but rather because it is associated with AD pathological processes. We have removed the statement in question and clarified our choice.

In particular the Introduction (pg 5) writes: *'Here, we employ this trajectory modelling approach to quantify the multivariate relationships between key biomarkers that underlie the pathogenesis of AD: A β , tau and neurodegeneration, together with measurement of the major genetic risk factor for late onset AD, the ϵ 4 allele of the Apolipoprotein E gene³⁷.*

6. The rationale to include both amyloid positives and negatives in the analysis is not very clear. Specifically, it appears that the authors suggested that amyloid biomarker positivity is a necessary condition for preclinical AD and only in such cases a prognostic index on future tau accumulation would make sense. Please elaborate.

We have now clarified why we retain amyloid negative individuals; that is our training sample is defined using longitudinal syndromic definitions independent of biomarker status. Further, we have removed the misleading reference from the introduction. Please also see our response to reviewer 3 comment 2 for more details on our motivation to retain amyloid negatives.

7. What is meant with the sentence that "MRI data were used for quantitation of PET data"? (line 442)

PET data were coregistered to MR images and regions of interest determined on the MR images used to extract the PET data for quantitation. We have now clarified this in the revised Methods section describing the PET analysis.

In particular the text (Pg 28-29) writes: *'FBP data were realigned, and the mean of all frames was used to co-register FBP data to each participant's structural MRI.'* ... *'For each subject, a global cortical PIB index was derived from the native-space DVR image coregistered to the MRI using FreeSurfer (5.3) parcellations using the Desikan-Killiany atlas⁶⁵ to define frontal (cortical regions anterior to the precentral sulcus), temporal (middle and superior temporal regions), parietal (supramarginal gyrus, inferior/superior parietal lobules, and precuneus), and anterior/posterior cingulate regions- ROIs combined as a weighted average.'*

8. As different beta-amyloid tracers were used in these different cohorts, it is imperative to put these on the same scale using the centiloid scale. We thank the reviewer for this suggestion. We have now converted both cohorts to centiloids and re-run the analyses. The results are highly similar as the previous submission and all conclusions remain the same. To address this issue in our previous analysis we had variance normalised within each cohort by the mean and standard deviation of cognitively unimpaired individuals. Following the reviewer's suggestion, we now present all data using the centiloid scale.

9. How would the SC and EAD multimodal scalar projection look like when amyloid was used as a binarized information rather than a continuous one. Although I very much appreciated the continuous approach, it would be more accessible to clinicians to evaluate a prognostic index based on binarized amyloid information.

We appreciate the need for discrete clinical decisions based on biomarker information. We favoured an approach to discretise our multimodal index after continuous information from multiple biomarkers (MRI and PET) were integrated into a single continuous scalar, rather than discretising biomarkers into positive or negative. Using the multimodal scalar projection we present a threshold value (>0.34) where there is a greater than 50% chance that an individual will be Clinically Declining. We show the efficacy of this binarized stratification in the ADNI 3 sample. We demonstrate that for this sample the Clinically Declining group will have declining cognition and accumulate pathological tau, whereas the Clinically Stable group showed no progressive AD pathology (i.e. stable cognition and no significant accumulation of tau). Therefore, we present a multimodal prognostic index that when thresholded can guide clinical decision to separate individuals who will have progressive AD pathology and symptomology (i.e. Clinically Declining) vs. those who will not (i.e. Clinically Stable).

We binarised the scalar projection using continuous amyloid values vs. binary values to capture trajectories for individuals who may be on the boarder of positivity. In particular, determining amyloid positivity using an SUVR threshold is inherently probabilistic and some sub-threshold individuals could in theory be amyloid positive. In light of this, including slightly sub-threshold amyloid negative individuals is preferable for modelling, as it ensures that our model is not overly sensitive to the threshold of amyloid positivity.

10. To assess gray matter density values did the authors consider head size differences?

Supplementary Figure 3 shows the relationship between the MTL grey matter density values and total intracranial volume for the ADNI2/GO training sample. The shared variance between these two variables is $R^2=3.7\%$. This result is not surprising as our pre-processing pipeline for the T1 structural scans omitted the modulation of the data following previous work (Radua et al., 2014); thus, we avoided introducing a proxy of volumetric differences in grey matter voxel values. In omitting this step, the MTL metric derived is from the weighted average of voxel values represented by the probability of being GM. This gives a concentration or density value to each voxel rather than volume. Therefore, we do not observe a strong effect of TIV on the MTL density metric used. A more detailed description of the process followed to derive this

metric can be found in our previously published work (Giorgio et al., 2020). We have now addressed this point in the revised manuscript.

In particular the Methods section writes (Pg 27): *‘To generate an individual’s score of medial temporal grey matter density we performed a matrix multiplication of the previously derived voxel weights matrix and each subject’s pre-processed T1 weighted MRI scans. Given that this value represents density and not regional volume, the medial temporal grey matter density score is not effected by head size differences (Supplementary Figure 3).’*

Supplementary Figure 3. Relationship of medial temporal lobe (MTL) grey matter density score and total intracranial volume (TIV). The relationship between the MTL grey matter density values and total intracranial volume for the ADNI2/GO training sample. The shared variance between these two variables is $R^2=3.7\%$.

11. It is not clear how the threshold of amyloid positivity of $SUVR = 1.1$ was achieved, in lieu of the longitudinal processing of the baseline amyloid data in ADNI. The $SUVR$ values are expected to be much lower than displayed in Figure 1 and amyloid positivity would be redefined. Please explain.

As we used only cross sectional amyloid PET we used the widely used threshold of $SUVR = 1.1$ for a reference region taken from the whole cerebellum as described in (Joshi et al., 2012). We had incorrectly written the reference region used in the original submission of the manuscript. We have now clarified this in the revised manuscript. Please also see our response to reviewer 3 comment 3.

In particular the revised Methods text (pg 27) writes: *‘Cortical Standardised Uptake Value Ratios (SUVR)s were generated by averaging FBP retention in a standard group of ROIs defined by FreeSurfer v5.3 (lateral and medial frontal, anterior and posterior cingulate, lateral parietal, and lateral temporal cortical grey matter) and dividing by the average uptake from the whole cerebellum to create an index of global cortical FBP burden ($A\beta$) for each subject 71.’*

12. The author report that when deriving the scalar projection for the independent cohorts ADNI 3 (CN=72; MCI:43) and the BAC (CN=56)) the clinician-based diagnosis and the multimodal scalar projection show poor agreement. How do they interpret this finding? Given that the robustness of the diagnosis has not been evaluated over multiple time points in the independent cohort, I fail to see the additional value gained from this analysis. We agree that longitudinal observations have the potential to form a more robust and confident clinical diagnosis. The comparison focuses on how well baseline syndromic definitions capture baseline biomarker characterisation. Our results show that the baseline syndromic definition is not sensitive to the baseline biology, suggesting that clinical syndromic definitions are not specific to AD biology or future tau accumulation. This finding is relevant for clinical trials as large and expensive trials targeting early biological processes in AD are increasingly aimed at asymptomatic or mildly symptomatic individuals (i.e. CN/MCI). We show that these syndromic labels—used to constrain recruitment into clinical trials—are not consistent with baseline biology that predicts clinical decline (i.e. classifying based on longitudinal syndromic changes) or future pathological changes. Thus, our modelling approach has two key advantages over syndromic guided recruitment: a) includes biologically relevant biomarker data b) makes use of longitudinal clinical observation. This allows us to make predictions both clinically and biologically relevant using a single time point (baseline) observation that provides a more sensitive and specific stratification than syndromic labels. We discuss this point further in the revised manuscript.

In particular the discussion (pg 17-18) now writes: *‘Our approach has potential relevance for the design of clinical trials in three main respects. First, we demonstrate that our multimodal modelling approach is more sensitive in capturing early-stage AD related pathology than a classification based on baseline syndromic labels. The poor sensitivity and specificity of syndromic labels to AD pathology¹⁰⁻¹³ has led to the introduction of a biological framework for AD classification⁸. We show that syndromic labels are not consistent with baseline biology that predicts clinical decline (i.e. changes in longitudinal syndromic changes) or future pathological changes (i.e. tau accumulation). Thus, our multimodal modelling approach has potential impact for drug discovery trials that recruit asymptomatic or mildly symptomatic individuals (i.e. CN/MCI) to target early biological processes in AD¹⁸.’*

13. Given that the multimodal scalar projection from the training sample rests on the cognitive change compared to cognitive stability over time, I find it surprising that SC and EAD individuals did not significantly vary in cognitive measures over time in the independent testing set. What is your explanation for this finding?

We agree with the reviewer and had expected to see a difference in cognition between SC and EAD individuals. Re-analysing the data following the suggestions of reviewer 1 showed a significant difference between the two groups. To explain this difference in our findings we show the distribution of the rate of cognitive decline for the SC and EAD from the initial analysis (Figure 1-response) and the distribution of the rate of cognitive decline for the SC (now referred to as Clinically Stable) and EAD (now referred to as Clinically Declining) from the current analysis (Figure 2-response). The lack of significant differences we initially reported was largely driven by a few outlier cases (Figure 1-response Circled) in the SC group, these outlier cases were on the border of our probabilistic boundary in the previous submission (see figure 1-response below). Re-analysing the data following the suggestions of reviewer 1 showed that the probabilistic boundary shifted and these two cases (circled in Figure 1-response) cross the threshold to be classified as EAD (now referred to as Clinically Declining). Figure 2-response shows differences between future cognition in the revised analysis; the outliers are now removed from the SC distribution and the comparison of the two groups is significantly different $t(100)=2.48$, $p=0.015$. We have now revised the manuscript to include this result.

In particular the Results section (Pg 11) writes: *'We next investigated if the classification of Clinically Stable vs. Clinically Declining is sensitive to future cognitive change (as measured by future annualised change in Preclinical Alzheimer's Cognitive Composite i.e. PACC) over the same time period. We observed a significant difference in future cognition between individuals classified as Clinically Stable(mean=0.13/year) vs. Clinically Declining(mean=-0.86/year) ($t(100)=-2.48$, $p=0.015$), with individuals classified as Clinically Declining showing significant worsening (i.e. rate of PACC change significantly less than 0) in future cognitive ability (one tail t-test $t(50)=-2.65$, $p=0.0054$). Taken together, our results show that a classification of Clinically Stable vs. Clinically Declining using our prognostic index based on baseline multimodal data is sensitive and specific to changes in future tau accumulation and cognitive decline in an independent sample without longitudinal syndromic information (i.e. ADNI 3).'*

Figure 1-response. Differences in future rate of PACC change SC vs EAD: Previous submission. This figure shows the distribution of the future rate of PACC change for our previous submission for SC and EAD groups. The redline is the median of the fit, the solid blue box represents the 25th to 75th percentile and the dashed black lines represents the range of the data and red crosses indicate outliers. The two outlier cases circled are now classified as EAD in the resubmission.

Figure 2-response. Differences in future rate of PACC change SC vs EAD: revised submission. This figure shows the distribution of the future rate of PACC change for the re-submission for SC and EAD groups. The redline is the median of the fit, the solid blue box represents the 25th to 75th percentile and the dashed black lines represents the range of the data

and red crosses indicate outliers. The two outlier cases circled in figure 1 are now classified as EAD and there is now a significant difference in future cognitive change between SC vs EAD.

14. Is the rate of tau accumulation significantly different when using the multimodal scalar projection compared to the CN/MCI diagnosis? The authors just state that the stratification is better using SC and EAD but do not present evidence that sensitivity and specificity measures are indeed better for one over the other.

To statistically compare these two stratification approaches based on sensitivity and specificity for future tau accumulation we now report the interclass correlation coefficient of future regional rate of tau accumulation across ROIs. This has been further clarified in the manuscript.

In particular the revised Results section (pg 11-12) now writes: *'Next, we compared how sensitive a baseline syndromic classification of CN vs. MCI is to future changes in tau accumulation. Averaging the annualised rate of tau accumulation within each of the 36 Desikan-Killiany ROIs for CN and MCI groups we contrasted the global rate of tau accumulation for CN vs. MCI groups (i.e. independent samples t-test across ROIs for CN vs MCI). We observed a marginally significant difference in global tau accumulation between CN and MCI groups ($t(70)=2, p=0.05$), with MCI individuals accumulating global cortical tau 1.9 times faster than CN individuals. To determine if a classification based on syndromic labels (i.e. CN vs. MCI) is specific to future regional rate of tau accumulation, we calculated the interclass correlation coefficient of future regional rate of tau accumulation across ROIs. A significant interclass correlation coefficient across ROIs ($r=0.47 [0.159 0.68]$, $F(35,36)=2.69$ $p=0.002$) suggests poor specificity to regional tau accumulation for stratification based on baseline syndromic definitions. Further, we tested which regions significantly accumulated tau (i.e. rate of accumulation significantly greater than 0; one sample (i.e. CN or MCI) one tail t-tests within each ROI). We showed that both CN and MCI individuals significantly accumulate tau, with a high degree of overlap across AD susceptible regions in the temporal and posteromedial cortices (**Supplementary Table 2, Supplementary Figure 1**). To further quantify this, we calculated the interclass correlation coefficient between a baseline syndromic definition of CN vs. Clinically Declining, and, MCI vs. Clinically Declining. For both syndromic definitions, we observed significant overlap in future regional tau accumulation with Clinically Declining (CN vs. Clinically Declining $r=0.584 [0.323 0.763]$, $F(35,36)=3.81$ $p<0.0001$; MCI vs. Clinically Declining $r=0.86 [0.751 0.928]$, $F(35,36)=13.7$ $p<0.0001$). Taken together, our results suggest that stratification based on syndromic diagnosis has poorer sensitivity and specificity to future tau accumulation compared to the biological classification of Clinically Stable vs. Clinically Declining based on our prognostic index.'*

15. Although the clinical trial analysis is comprehensive and the “new kid on the block” in the biomarker research field, the impression remains that reducing meaningful cognitive decline in combination with the reduction of tau, will be more important for the design of clinical trials than just reducing tau pathology. Please elaborate on this point. We thank the reviewer for this suggestion and agree that ultimately clinical outcomes (i.e. reducing cognitive decline) are the key measure in clinical trials. In our previous work we have already shown that our algorithm predicts future cognitive decline (Giorgio et al., 2020).

Further, existing data have shown strong relationships between tau deposition and clinical outcomes. Using our modelling approach for stratification allows the selection of individuals with progressive disease (i.e. showing both cognitive decline and tau accumulation) at a point in their disease progression trajectory that could be ameliorated with treatment. Given the well-established relationship between tau and cognition, one would anticipate an anti-tau drug that halts tau accumulation to also halt cognitive decline. Evidence to the contrary would be equally important and therefore it would be useful to incorporate both measures.

We have now discussed this further in the revised manuscript.

In particular the Discussion text now writes (pg 18): *‘Second, using the rate of tau accumulation as an outcome measure results in 31% reduction in the sample size for detecting a clinically meaningful change at early stages of AD compared to the gold standard cognitive instrument (PACC⁴²). This is consistent with previous work showing that a smaller sample size is required to detect a clinically meaningful change in tau accumulation within the “meta-ROI” for tau accumulation than using a cognitive endpoint²⁴. Yet, determining the success of a trial based on reduction of tau alone may be suboptimal given the failure of amyloid clearing treatments to halt clinical decline²¹. Therefore, measuring changes in tau and cognition simultaneously may be more appropriate for assessing efficacy of anti-tau drug treatments. Our machine learning approach is well suited to address this need as it allows us to select individuals who are predicted to both accumulate tau and have declining cognition.’*

16. The authors claim that the multimodal modelling approach may be better than syndromic labels such as preclinical AD, MCI or AD to capture AD related pathology. An issue that arises here is that the current multimodal modelling approach rested on exactly those syndromic definitions of CN and MCI/AD in the ADNI cohort in the first place to even get to the prognostic index. The authors should elaborate on this argument.

We thank the reviewer for this comment. To clarify, our findings demonstrate that the multimodal *baseline* index is better than the *baseline* syndromic labels at predicting future tau accumulation and the two baseline stratifications have poor agreement. We have now clarified this throughout the revised manuscript.

As the reviewer points out, these baselines or one-shot clinical appraisals are less robust than measures taken over time. To increase the reliability of our training labels, we used *longitudinal* clinical appraisal to train our classification algorithm. Yet, these outcome labels may still be limited in reflecting the underlying pathology, due to their inherently noisy nature. We believe that our machine learning approach is well suited to account for this for the following reasons. In particular, we derived the scalar projection by training the model based on ‘noisy’ diagnostic labels. However, as our metric learning model has limited freedom (linear low-parameter model), separating continuous target values (i.e. individualised trajectories) into two broad classes forces the model to extract key multivariate relationships in the data that distinguish between target populations, ignoring subtle differences in target values. That is, the model cannot learn a set of parameters that fit subtle differences in target values, e.g. an individual who is assigned a label of Clinically Declining but has no underlying AD pathology will be classified as Clinically Stable. Restricting the model parameter space in this way allows us to

use noisy diagnostic labels to generate a highly sensitive and specific AD biological index. This low parameter model learning based on broad classes results in the model generating predictions that capture ‘hidden’ biological changes that occur when diagnostic categories change. Further this approach does not afford the model sufficient freedom to overfit the data based on binarised labels that do not encompass the rich continuous information that separates individuals either between or within diagnostic classes.

Finally, a key benefit of this approach is that it allows us to train a model that can be used to make transfer predictions across a class of prediction problems, for example. When predicting AD related changes in tau accumulation in cohorts sampled with longitudinal clinical appraisal prior to the addition of FTP-PET. A model that trains on data collected prior to the introduction of FTP-PET and makes predictions in an independent dataset of participants with FTP-PET will support data harmonisation data across cohorts over the last decade. This is discussed further in Supplementary Methods.

17. I think it should read diagnosis of dementia not “demented” (e.g., line 414).

We have now changed this to Alzheimer’s clinical syndrome in the revised manuscript.

References:

- Baker, S.L., Harrison, T.M., Maass, A., Joie, R. La, Jagust, W.J., 2019. Effect of off-target binding on 18F-flortaucipir variability in healthy controls across the life span. *J. Nucl. Med.* 60, 1444–1451. <https://doi.org/10.2967/jnumed.118.224113>
- Beach, T.G., Monsell, S.E., Phillips, L.E., Kukull, W., 2012. Accuracy of the clinical diagnosis of Alzheimer disease at National Institute on Aging Alzheimer Disease Centers, 2005-2010. *J. Neuropathol. Exp. Neurol.* 71, 266–273. <https://doi.org/10.1097/NEN.0b013e31824b211b>
- Giorgio, J., Landau, S., Jagust, W., Tino, P., Kourtzi, Z., 2020. Modelling prognostic trajectories of cognitive decline due to Alzheimer's disease. *NeuroImage Clin.* 102199. <https://doi.org/10.1016/j.nicl.2020.102199>
- Jack, C.R., Knopman, D.S., Jagust, W.J., Petersen, R.C., Weiner, M.W., Aisen, P.S., Shaw, L.M., Vemuri, P., Wiste, H.J., Weigand, S.D., Lesnick, T.G., Pankratz, V.S., Donohue, M.C., Trojanowski, J.Q., Trojanowski, J.Q., 2013. Tracking pathophysiological processes in Alzheimer's disease: an updated hypothetical model of dynamic biomarkers. *Lancet. Neurol.* 12, 207–16. [https://doi.org/10.1016/S1474-4422\(12\)70291-0](https://doi.org/10.1016/S1474-4422(12)70291-0)
- Jack, C.R., Knopman, D.S., Jagust, W.J., Shaw, L.M., Aisen, P.S., Weiner, M.W., Petersen, R.C., Trojanowski, J.Q., Trojanowski, J.Q., 2010. Hypothetical model of dynamic biomarkers of the Alzheimer's pathological cascade. *Lancet. Neurol.* 9, 119–28. [https://doi.org/10.1016/S1474-4422\(09\)70299-6](https://doi.org/10.1016/S1474-4422(09)70299-6)
- Jack, C.R., Wiste, H.J., Weigand, S.D., Therneau, T.M., Lowe, V.J., Knopman, D.S., Botha, H., Graff-Radford, J., Jones, D.T., Ferman, T.J., Boeve, B.F., Kantarci, K., Vemuri, P., Mielke, M.M., Whitwell, J., Josephs, K., Schwarz, C.G., Senjem, M.L., Gunter, J.L., Petersen, R.C., 2020. Predicting future rates of tau accumulation on PET. *Brain* 143, 3136–3150. <https://doi.org/10.1093/brain/awaa248>
- Joshi, A.D., Pontecorvo, M.J., Clark, C.M., Carpenter, A.P., Jennings, D.L., Sadowsky, C.H., Adler, L.P., Kovnat, K.D., Seibyl, J.P., Arora, A., Saha, K., Burns, J.D., Lowrey, M.J., Mintun, M.A., Skovronsky, D.M., 2012. Performance characteristics of amyloid PET with florbetapir F 18 in patients with Alzheimer's disease and cognitively normal subjects. *J. Nucl. Med.* 53, 378–384. <https://doi.org/10.2967/jnumed.111.090340>
- Klunk, W.E., Koeppe, R.A., Price, J.C., Benzinger, T.L., Devous, M.D., Jagust, W.J., Johnson, K.A., Mathis, C.A., Minhas, D., Pontecorvo, M.J., Rowe, C.C., Skovronsky, D.M., Mintun, M.A., 2015. The Centiloid project: Standardizing quantitative amyloid plaque estimation by PET. *Alzheimer's Dement.* 11, 1-15.e4. <https://doi.org/10.1016/j.jalz.2014.07.003>
- Lowe, V.J., Lundt, E.S., Albertson, S.M., Min, H., Fang, P., Przybelski, S.A., Senjem, M.L., Schwarz, C.G., Kantarci, K., Boeve, B., Jones, D.T., Reichard, R.R., Tranovich, J.F., Hanna Al-Shaikh, F.S., Knopman, D.S., Jack, C.R., Dickson, D.W., Petersen, R.C., Murray, M.E., 2020. Tau-positron emission tomography correlates with neuropathology findings. *Alzheimer's Dement.* 16, 561–571. <https://doi.org/10.1016/j.jalz.2019.09.079>
- Navitsky, M., Joshi, A.D., Kennedy, I., Klunk, W.E., Rowe, C.C., Wong, D.F., Pontecorvo, M.J., Mintun, M.A., Devous, M.D., 2018. Standardization of amyloid quantitation with florbetapir standardized uptake value ratios to the Centiloid scale. *Alzheimer's Dement.* 14, 1565–1571. <https://doi.org/10.1016/j.jalz.2018.06.1353>
- Radua, J., Canales-Rodríguez, E.J., Pomarol-Clotet, E., Salvador, R., 2014. Validity of modulation and optimal settings for advanced voxel-based morphometry. *Neuroimage* 86, 81–90. <https://doi.org/10.1016/j.neuroimage.2013.07.084>
- Schöll, M., Lockhart, S.N., Schonhaut, D.R., O'Neil, J.P., Janabi, M., Ossenkoppele, R., Baker, S.L., Vogel, J.W., Faria, J., Schwimmer, H.D., Rabinovici, G.D., Jagust, W.J., 2016. PET Imaging of Tau Deposition in the Aging Human Brain. *Neuron* 89, 971–982. <https://doi.org/10.1016/j.neuron.2016.01.028>

REVIEWER COMMENTS

Reviewer #1 (Remarks to the Author):

The authors have done a decent job of responding to the comments of all reviewers. However, some key question marks remain.

First, a general comment. As pointed out in my first review, I think it's important to consider accumulated tau burden in conjunction with tau accumulation rate. Here the authors talk only of the rate. If tau accumulation rates will indeed be used as clinical trial outcomes, then that's fine, just add a comment early in the paper to make this very clear. If not (and tau burden is also important) then the authors need to rework the manuscript.

Now I'll address the authors' responses to my own comments (starting with numbers provided by the authors):

3.

a) (my numbering) Regarding the regression lines in Figures 7 and 8:

The 95% confidence interval in Fig 7c (parietal) seems to suggest that a line of no trend would explain the data just as well (statistically speaking). Fig 7d is similar, although the correlation is probably significant (for what that's worth: null hypothesis testing was never intended to be confirmatory, but that's a fight for another day).

The question is: how do these results (Figs 7,8) translate to the clinic? Is it that one can say with 95% confidence that there is, *at best*, a weak positive trend? This needs clarifying and making explicit in the manuscript: the reader needs to be convinced that you are providing a useful model, not merely a statistically significant one. Sure, you can use it to reduce the clinical trial sample (great!), but is it really that useful? I want to be convinced, but I'm not yet.

b) "The focus of our analyses is early-stage stratification for patients with AD pathology. In particular, we developed a modelling approach that is sensitive and specific to the AD topography of tau accumulation."

See Vogel et al, medRxiv 2020 where four topographic subtypes of tau accumulation were robustly identified across multiple cohorts and PET tracers. The authors need to convince the reader that their approach is still valuable in that it considers only temporal aspects of an assumed single topography, whereas the evidence suggests multiple topographies/subtypes: spatiotemporal heterogeneity of tau accumulation (Vogel-2020) and grey-matter atrophy (Young et al, Nature Commun 2018).

c) Regarding using a linear model:

"Thus, we tested whether a linear model captures the relationship between the scalar projection and individual rate of future tau accumulation for our early-stage AD sample..."

- I agree that a linear model makes sense on the scale of a few years. (The authors make a strange argument that they're looking at "the linear portion" of the tau curve: any curve is locally linear on a sufficiently short timescale.)

- The challenge I pose to the authors is related to the nonlinear nature over longer times that are of potential clinical interest. The model, as I understand it, does not explicitly link the linear segments from individuals (over short times) to a global nonlinear model of tau accumulation (over longer times). For example, consider a new patient assessed by the model to be in the earliest stage of AD pathology at their baseline visit. Using a linear prediction is sensible in the short-term, but will likely produce highly inaccurate (over)estimates beyond a few years. Perhaps this isn't a massive problem in practice, since the patient might be asked to return for another assessment in, say, one/two years from baseline. At which point, the model-based predictions would be updated. Worth making this point in the revised manuscript, i.e., if you're motivated by short-time applications of no more than a couple of years, such as clinical trials, then that justifies using a linear model. But using a single linear model to cover all levels of pathology (over many years) is less convincing.

d) Revised text on page 19: "as evidenced by the generalisability of our results across samples"
- Please tone this down a bit. Or qualify it with a comment about (small) sample sizes. Perhaps: "as supported by the consistency of our results across (admittedly small) samples."

e) Revised text: "Extending our modelling approach to integrate less-costly (i.e. plasma) and non-invasive (i.e. cognitive) data has strong potential for stratification at asymptomatic or early stages of AD."

- I really like this idea. Is something like this feasible for the authors to implement in ADNI? Using cognitive data, not "new plasma biomarkers" (since they don't exist in ADNI). If it *is* possible to compare how this model performs with different inputs (cognitive test scores versus neuroimaging+biomarkers), such a study would add a lot of value to the paper as an additional supplementary analysis.

4. "We look forward to the opportunity to test our approaches in a clinical trial."

- Have you approached Pharma to get your model included in future trials?

5.

a) I understand now: you are directly comparing to Jack-2020. I think this paragraph needs to be rewritten to make this more obvious. For example, wording such as "A recent paper claimed that AB status alone...We found that..." would avoid the reader being confused (as I was) into thinking that you're saying, simply, that "multivariate is better than univariate".

- Where you write "by 44%", the reader would appreciate the raw numbers also being included, e.g., "from 100 to 66".

b)

"Direct comparison of diverse machine learning approaches remains challenging as cross-validation methodology, sample sizes and sample heterogeneity have a significant effect on model performance metrics"

- This sentence misses the point. Perhaps I was unclear when I suggested that you should compare your results with a benchmark model. For example, this could be SVM or logistic regression for the classification experiments. To make the comparison fair, you would ideally keep all other experimental design choices fixed: input features, cross-validation settings, etc.

"Thus, our predictions are not directly comparable to those of models performing only binary classifications based on syndromic diagnosis"

- To be blunt, this is nonsense. As explained above, and with apologies if I was unclear in my first review, your classifier experiments can be compared with SVM/etc. in place of your model.

Additionally, your prediction results could also be compared with latent-time mixed-effects models such as (Donohue et al., *Alzheimers Dement* 2014; Li et al., *Stat Meth Med Res* 2017; Lorenzi et al., *NeuroImage* 2017), each of which has public source code available.

"To compare performance of our modelling approach we compared our multimodal baseline stratification with two common approaches to stratify individuals in the early stages of AD: 1) syndromic definitions and 2) amyloid status."

- Seems a reasonable comparison, since your aim is to compete with these in clinical trial settings. However, if one can achieve comparable performance with (for example) a SVM (see my points above), then the technical contribution of this paper is diminished considerably. The authors are strongly encouraged to run these experiments to convince themselves and the reader of their model's contribution to knowledge.

"This allows us to draw conclusions that are relevant for patient stratification and the design of clinical trials...variability in pathological state across groups..."

- Sorry to harp on about it, but your method handles temporal variability (disease stage/pathology

severity) but doesn't handle spatial variability (different subtype patterns) in tau accumulation. See the four spatiotemporal subtypes of AD: Vogel-2020.

13. Revised text: "...Recent evidence suggests that there are consistent patterns of tau spread..."
- Recent evidence suggests, quite strongly, that there are four spatiotemporal subtypes of tau accumulation: Vogel-2020 (as mentioned above). Using a much larger tau PET sample size covering early through to late AD, across multiple tau-PET tracers and multiple cohorts, Vogel et al. provided strong evidence for four spatiotemporal subtypes of tau accumulation in Alzheimer's.

Check spelling throughout: I found some minor errors, e.g., clinician, proteinopathies.

Reviewer #2 (Remarks to the Author):

It is clear that the authors have put a lot of work in to editing this manuscript based on reviewer comments, well done. I like the figure you have produced and along with the re-written introduction I find the paper much clearer. Readability is much improved, but could be improved further with section numbering. I particularly like the sections where you compare the sample sizes needed to detect change in tau compared to cognition/ abeta and feel these have improved since the previous manuscript.

I have few minor concerns:

- In the introduction it states "These clinical syndromic definitions have no discrete demarcations on cognitive scales" - do you mean that threshold scores on cognitive testing are not part of the criteria for diagnosis? If so, please make clearer. There are cut-offs on routine cognitive tests e.g. the MMSE, that are used to demarcate MCI and AD, so this statement as is does not hold.

Typos:

- protienopathies should be proteinopathies
- "we tested whether difference in MRI filed strength" should be field strength

Reviewer #3 (Remarks to the Author):

The authors have now revised the manuscript and addressed my concerns satisfactorily. I am happy with revised version.

Reviewer #4 (Remarks to the Author):

The authors have done a great and satisfactory job in responding to all critical comments raised. Specifically the inclusion of Figure 1 and the cleaning up the introduction with focusing on the goals have greatly improved the accessibility of the article itself.

Responses to Reviewers

Reviewer 1.

- 1. First, a general comment. As pointed out in my first review, I think it's important to consider accumulated tau burden in conjunction with tau accumulation rate. Here the authors talk only of the rate. If tau accumulation rates will indeed be used as clinical trial outcomes, then that's fine, just add a comment early in the paper to make this very clear. If not (and tau burden is also important) then the authors need to rework the manuscript.*

We thank the reviewer for this suggestion. In designing clinical trials for AD several outcome measures are often taken into consideration. Although reduction in cognitive decline is invariably a primary outcome, the recent Donanemab trial included change in tau as a secondary outcome measure (Mintun et al., 2021). A range of therapeutic interventions, and in particular amyloid-lowering immunotherapies, aim to slow down the rate of tau accumulation. Our model is well suited to address the needs of clinical trial stratification, as it is sensitive to both change in cognition and change in tau from baseline. This will become particularly relevant as anti-tau interventions enter the clinical trial phase, where halting the rate of accumulation and clearing already accumulated tau are of particular interest.

Following the reviewer's suggestion, we investigated whether our baseline multimodal stratification using non-tau biomarkers is sensitive to baseline tau burden. Using the model derived stratification of Clinically Stable vs. Clinically Declining, we contrasted the baseline tau burden between the two groups. We observed that individuals who are classified as Clinically Declining have significantly more tau burden than those classified as Clinically Stable. For the ADNI 3 sample, this was observed across the cortex as well as in Braak stage ROIS (Braak I mean difference = 0.167 SUVR, $t(113)=5.6$, $p<0.001$; Braak II mean difference = 0.074 SUVR, $t(113)=2.6$, $p=0.01$; Braak III mean difference = 0.12 SUVR, $t(113)=4.77$, $p<0.001$; Braak IV mean difference = 0.08 SUVR, $t(113)=3.81$, $p<0.001$; Braak V mean difference = 0.07 SUVR, $t(113)=3.47$, $p<0.001$; Braak VI mean difference = 0.041 SUVR, $t(113)=2.53$, $p=0.012$) (**Supplementary Figure 3, Supplementary Table 1**). The same stratification in the BACS samples showed that the BACS Clinically Declining group has increased tau compared to the Clinically Stable group (Braak I mean difference=0.0445 SUVR; Braak II mean difference=0.0274 SUVR; Braak III mean difference=0.0201 SUVR; Braak IV mean difference=0.0233 SUVR; Braak V mean difference=0.0324 SUVR; Braak VI mean difference= 0.0368 SUVR) (**Supplementary Figure 3**). Please note that these comparisons use the preferred cross sectional reference region of the inferior cerebellum to derive SUVR. Further, as there is no partial volume correction the signal from Hippocampus (Braak II) may be unreliable.

Here, we show that our model-derived stratification of Clinically Stable vs. Clinically Declining using baseline non-tau biomarkers is sensitive to both baseline tau burden as well as future rate of tau accumulation. Therefore, we show that our single stratification approach is sensitive to: 1. Baseline Tau, 2. Future change in cognition 3. Future change in tau accumulation, providing converging evidence that our stratification approach is relevant to a wide range of clinical trial outcome measures. Further, our results provide a quantitative link

between continuous measures of baseline medial temporal atrophy, amyloid burden and APOE 4 status, and baseline tau as well as future changes in tau and cognition. As our stratification approach is sensitive to a range of AD pathology and AD related changes, it is relevant for stratification in future clinical trials (particularly those with multiple outcome measures; i.e. clearing tau, halting tau accumulation and halting cognitive decline). Finally, we provide evidence that machine learning tools are well suited to combining interactive biomarkers in a meaningful way that predicts changes in biomarkers that are not explicitly modelled (i.e. tau, cognition).

In particular the text now writes (Introduction)

“Thus, slowing rates of tau accumulation within these primary regions common to the different spatiotemporal profiles, similar to slowing rates of cognitive decline, could serve as an attractive biomarker outcome.”

(Results)

*“First, we contrasted baseline tau for Clinically Stable vs Clinically Declining individuals (**Supplementary Results: Differences in baseline tau burden Clinically Stable vs. Clinically Declining**). These analyses show that individuals classified as Clinically Declining have significantly greater baseline tau across the cortex (Braak I mean difference = 0.167 SUVR, $t(113)=5.6, p<0.001$; Braak II mean difference = 0.074 SUVR, $t(113)=2.6, p=0.01$; Braak III mean difference = 0.12 SUVR, $t(113)=4.77, p<0.001$; Braak IV mean difference = 0.08 SUVR, $t(113)=3.81, p<0.001$; Braak V mean difference = 0.07 SUVR, $t(113)=3.47, p<0.001$; Braak VI mean difference = 0.041 SUVR, $t(113)=2.53, p=0.012$) (**Supplementary Figure 2, Supplementary Table 1**). This pattern of increased baseline tau was also observed in the BACS Clinically Declining sample (**Supplementary Figure 3**).”*

(Discussion)

“cognitive decline is considered as a primary outcome measure for clinical trials (Sperling et al., 2014), recent trials indicate a potential future role for biomarkers in drug discovery (e.g. A β in the case of the recently FDA approved aducanumab). Recent trials in early AD participants have also investigated downstream effects of A β lowering immunotherapies on both cognitive decline and changes in cortical tau burden measured with [18 F]-flourtaucipir PET (FTP-PET) (Mintun et al., 2021). As tau is strongly linked to both future neurodegeneration and cognitive decline (Hanseeuw et al., 2019) this makes reduction of future tau accumulation a relevant intervention target and potential outcome measure. This is further evidenced by anti tau drugs entering the clinical trial pipeline (Cummings et al., 2019; Long and Holtzman, 2019). Given the failures of anti-amyloid interventions to halt clinical decline, simply clearing already deposited tau may be insufficient to stop downstream events (Long and Holtzman, 2019). Therefore, measuring changes in tau and cognition simultaneously may be more appropriate for assessing efficacy of anti-tau drug treatments. That is, targeting individuals with the highest risk of depositing tau rather than those burdened with tau, may increase the likelihood of successfully modifying downstream clinical decline.”

(Supplementary Results)

“Differences in baseline tau burden for Clinically Stable vs. Clinically Declining

Using the model derived classification of Clinically Stable vs. Clinically Declining, we contrasted the baseline tau burden between the two groups. FTP data were realigned, and the mean of all frames was used to coregister FTP to each participant's MRI acquired closest to the time of the FTP-PET. FTP standardised uptake value ratio (SUVR) images were normalised to inferior cerebellar grey matter (Baker et al., 2017). MR images were segmented and parcellated into 72 ROIs taken from the Desikan-Killiany atlas using Freesurfer (V5.3). These ROIs were then used to extract regional SUVR data from the cerebellar normalised FTP-PET images. Left and right hemisphere ROIs were averaged to generate 36 ROIs for further analysis. SUVR values in six aggregate Braak staging regions were also derived averaging uptake across individual Freesurfer region of interests (ROIs) comprising each Braak region (Maass et al., 2017).

Contrasting the baseline tau burden for Clinically Stable (n=59) vs. Clinically Declining (n=56) individuals showed increased baseline tau across the cortex for the Clinically Declining group (**Supplementary Figure 2a, Supplementary Table 1**). This overall pattern was consistent across all Braak regions, with a greater difference observed in earlier Braak regions (Braak I mean difference = 0.167 SUVR, $t(113)=5.6, p<0.001$; Braak II mean difference = 0.074 SUVR, $t(113)=2.6, p=0.01$; Braak III mean difference = 0.12 SUVR, $t(113)=4.77, p<0.001$; Braak IV mean difference = 0.08 SUVR, $t(113)=3.81, p<0.001$; Braak V mean difference = 0.07 SUVR, $t(113)=3.47, p<0.001$; Braak VI mean difference = 0.041 SUVR, $t(113)=2.53, p=0.012$) (**Supplementary Figure 2b, Supplementary Table 1**). This pattern of increased baseline tau for the Clinically Declining group was also observed in the BACS sample (Braak I mean difference=0.0445 SUVR; Braak II mean difference=0.0274 SUVR; Braak III mean difference=0.0201 SUVR; Braak IV mean difference=0.0233 SUVR; Braak V mean difference=0.0324 SUVR; Braak VI mean difference= 0.0368 SUVR) (**Supplementary Figure 3**). Note that, as there is no partial volume correction, the signal from Hippocampus (Braak II) may be unreliable.”

a. Difference in baseline tau burden

b. Baseline tau burden within Braak stages

Supplementary Figure 2. ADNI 3 difference in baseline tau burden Clinically Declining vs. Clinically Stable. a. mean difference in baseline tau burden across the 36 Desikan-Killiany in the ADNI 3 sample. b. group differences in baseline tau burden across the six Braak stages, blue boxes show the distribution of baseline SUVR for the clinically stable group, red boxed show the distribution of baseline SUVR for the Clinically Declining group.

Braak		Mean	Mean	t-stat	t-stat	p-val	p-val
-------	--	------	------	--------	--------	-------	-------

Stage		Clinically Stable	Clinically Declining	ROI	Braak	ROI	Braak
1	ENTORHINAL	1.108	1.275	5.596	5.596	<0.001	<0.001
2	HIPPOCAMPUS	1.251	1.325	2.568	2.568	0.012	0.012
3	PARAHIPPOCAMPAL	1.111	1.214	4.366	4.773	<0.001	<0.001
	FUSIFORM	1.193	1.308	4.185		<0.001	
	LINGUAL	1.095	1.152	3.219		0.002	
	AMYGDALA	1.214	1.397	5.058		<0.001	
4	MIDDLETEMPORAL	1.168	1.278	3.576	3.81	0.001	<0.001
	CAUDALANTERIORCINGULATE	1.058	1.098	2.026		0.045	
	ROSTRALANTERIORCINGULATE	1.084	1.125	1.961		0.052	
	POSTERIORCINGULATE	1.098	1.187	3.723		<0.001	
	ISTHMUSCINGULATE	1.095	1.196	3.820		<0.001	
	INSULA	1.116	1.178	3.039		0.003	
	INFERIORETEMPORAL	1.205	1.335	3.890		<0.001	
TEMPORALPOLE	1.103	1.195	3.593	<0.001			
5	SUPERIORFRONTAL	0.995	1.077	3.707	3.469	<0.001	<0.001
	LATERALORBITOFRONTAL	1.194	1.258	2.964		0.004	
	MEDIALORBITOFRONTAL	1.121	1.185	3.118		0.002	
	FRONTALPOLE	0.973	1.022	2.061		0.042	
	CAUDALMIDDLEFRONTAL	1.031	1.143	3.628		<0.001	
	ROSTRALMIDDLEFRONTAL	1.049	1.134	3.107		0.002	
	PARSOPERCULARIS	1.087	1.159	2.786		0.006	
	PARSORBITALIS	1.144	1.186	1.884		0.062	
	PARSTRIANGULARIS	1.115	1.164	2.190		0.031	
	LATERALOCIPITAL	1.095	1.181	2.730		0.007	
	SUPRAMARGINAL	1.087	1.149	2.747		0.007	
	INFERIORPARIETAL	1.130	1.237	3.475		0.001	
	SUPERIORETEMPORAL	1.087	1.149	2.931		0.004	
	SUPERIORPARIETAL	1.028	1.097	3.227		0.002	
	PRECUNEUS	1.112	1.211	3.976		<0.001	
BANKSSTS	1.211	1.316	3.193	0.002			
TRANSVERSETEMPORAL	1.035	1.048	0.725	0.470			
6	PERICALCARINE	1.119	1.168	2.666	2.534	0.009	0.012
	POSTCENTRAL	0.979	0.998	1.179		0.241	
	CUNEUS	1.102	1.146	2.216		0.029	
	PRECENTRAL	1.001	1.040	2.171		0.032	
	PARACENTRAL	1.022	1.074	2.776		0.006	

Supplementary Table 1. ADNI 3 difference in baseline tau burden Clinically Declining vs. Clinically Stable. Measures of baseline regional tau SUVR the Desikan Killiany atlas for ADNI 3 individuals grouped in the 6 Braak stages. The average tau burden and two sample t-t-test statistics describing whether a region has significantly greater tau for individuals classified as Clinically Declining

a. Difference in baseline tau burden

b. Baseline tau burden within Braak stages

Supplementary Figure 3. BACS difference in baseline tau burden Clinically Declining vs. Clinically Stable. a. mean difference in baseline tau burden across the 36 Desikan-Killiany in the BACS sample. b. group differences in baseline tau burden across the six Braak stages, blue boxes show the distribution of baseline SUVR for the clinically stable group, red boxes show the distribution of baseline SUVR for the Clinically Declining group.

2. a) (my numbering) *Regarding the regression lines in Figures 7 and 8: The 95% confidence interval in Fig 7c (parietal) seems to suggest that a line of no trend would explain the data just as well (statistically speaking). Fig 7d is similar, although the correlation is probably significant (for what that's worth: null hypothesis testing was never intended to be confirmatory, but that's a fight for another day). The question is: how do these results (Figs 7,8) translate to the clinic? Is it that one can say with 95% confidence that there is, *at best*, a weak positive trend? This needs clarifying and making explicit in the manuscript: the reader needs to be convinced that you are providing a useful model, not merely a statistically significant one. Sure, you can use it to reduce the clinical trial sample (great!), but is it really that useful? I want to be convinced, but I'm not yet.*

We test longitudinal biomarker prediction with potential translational impact for the design of clinical trials, rather than implementation in routine clinical practice.

First, our work provides evidence that combining continuous baseline multimodal non-tau biomarkers using machine learning predicts future individualised rates of tau accumulation. Using simple linear regression models that over the time spans typical of clinical trials (i.e. 1-3 years), we determine a linear subspace that allows us to infer risk of future tau accumulation across regions susceptible to tau accumulation in asymptomatic and early AD. This linear subspace relates to individual variability in future tau accumulation within the primary seeding regions of tau spread in early AD. We present individual data for two aggregate regions within the parietal and temporal cortex. In addition, we calculate these trend lines within individual ROIs in ADNI 3 and generate explicit predictions of individualised rates of tau accumulation in the BACS sample, with R^2 values that explain 40% of the variance. Our individualised prediction results within these ROIs show a positive relationship in ADNI 3 demonstrating that larger scalar projection values, indicating higher distance from the Clinically Stable prototype, relate to faster rates of tau accumulation. The same relationship was also observed in the BACS asymptomatic sample. Note that the linear subspace that captures degrees of freedom needed to span individual variability in future tau accumulation has been learnt solely on ADNI 2. Making reliable predictions on independent cohort samples (ADNI 3, BACS) by applying the same learned subspace suggests generalised validity of the learnt feature interactions coded in the subspace basis.

Second, common stratification approaches are constrained by unimodal stratifications based on β -amyloid alone or by clinical diagnoses. Our results show that these stratification approaches are limited. Using multimodal biomarkers, we generate a stratification that has greater sensitivity to future changes in tau accumulation than typical unimodal stratification approaches. In particular our multimodal modelling approach shows higher statistical power and reduced sample sizes for stratification based on future rates of tau accumulation. Optimising group membership has the potential to reduce sample size and costs incurred in phase 3 human trials. Further, determining groups based on 1. baseline tau, 2. future changes in cognition and 3. rate of tau accumulation in early AD regions has the potential to reduce

the heterogeneity in treatment groups, increasing the sensitivity of clinical trials in assessing drug efficacy.

Finally, our findings demonstrate that using the scalar projection value derived from modelling multimodal baseline data, we can re-stratify individuals based on their individualised predicted rates of future tau accumulation (i.e. no, slow, intermediate or rapid anticipated accumulation of tau). These results have relevance to clinical trials, as higher multimodal biomarker severity relates to increased rate of change in tau accumulation in asymptomatic and early AD. Therefore, trials may increase statistical power to detect treatment effects (i.e. reduction in future tau accumulation) if they select samples with larger scalar projections (i.e. higher pathological multimodal biomarker values). Further, matching individuals at the same baseline pathological state, particularly in regard to anticipated change of biomarkers, is highly relevant given recent results of the DIAN-TU study into anti-amyloid interventions. This study suggests that selecting a narrower range of baseline disease severity may have increased statistical power to observe treatment effects in halting downstream effects (i.e. cognitive decline) in dominantly inherited AD (Salloway et al., 2021).

In particular the text now writes (Discussion)

“Further, our trajectory modelling approach shows that there is a linear relationship between baseline non-tau biomarkers and future rates of tau accumulation over a short timespan (typical of a clinical trial) . This result has particular relevance to clinical trial design, as it suggests that patient stratification for clinical trials can be optimised to select individuals with the greatest potential treatment effect (i.e. greatest reduction in rate of tau accumulation). Thus, our findings demonstrate: a) the benefit of combining continuous values of $A\beta$ and medial temporal grey matter density for prognostication, b) the potential of our approach to inform the design of clinical trials targeting pathophysiological changes at the earliest stages of AD. “

3. *b) "The focus of our analyses is early-stage stratification for patients with AD pathology. In particular, we developed a modelling approach that is sensitive and specific to the AD topography of tau accumulation." See Vogel et al, medRxiv 2020 where four topographic subtypes of tau accumulation were robustly identified across multiple cohorts and PET tracers. The authors need to convince the reader that their approach is still valuable in that it considers only temporal aspects of an assumed single topography, whereas the evidence suggests multiple topographies/subtypes: spatiotemporal heterogeneity of tau accumulation (Vogel-2020) and grey-matter atrophy (Young et al, Nature Commun 2018).*

We address this point in our response to point 13 below.

4. *c) Regarding using a linear model: "Thus, we tested whether a linear model captures the relationship between the scalar projection and individual rate of future tau accumulation for our early-stage AD sample..."*

- I agree that a linear model makes sense on the scale of a few years. (The authors make a strange argument that they're looking at "the linear portion" of the tau curve: any curve is locally linear on a sufficiently short timescale.)
- The challenge I pose to the authors is related to the nonlinear nature over longer times that are of potential clinical interest. The model, as I understand it, does not explicitly link the linear segments from individuals (over short times) to a global nonlinear model of tau accumulation (over longer times). For example, consider a new patient assessed by the model to be in the earliest stage of AD pathology at their baseline visit. Using a linear prediction is sensible in the short-term, but will likely produce highly inaccurate (over)estimates beyond a few years. Perhaps this isn't a massive problem in practice, since the patient might be asked to return for another assessment in, say, one/two years from baseline. At which point, the model-based predictions would be updated. Worth making this point in the revised manuscript, i.e., if you're motivated by short-time applications of no more than a couple of years, such as clinical trials, then that justifies using a linear model. But using a single linear model to cover all levels of pathology (over many years) is less convincing.

Our aim is to build a predictive model that captures the earliest AD related changes in tau accumulation. We believe that a linear model is appropriate because of a) the sampling characteristics of the data available (i.e. over a time span of a few years), b) the data we use to stratify populations for a typical AD trial (i.e. over a time span of a few years). As the reviewer points out, our current linear model is not appropriate to model the non-linear trajectories over a longer time span particularly as topologies start to separate into clinical tau phenotypes. We have now clarified that our aim is to predict future rate of tau accumulation over a timescale typical of AD clinical trials and determine: 1. the optimal data to stratify early and asymptomatic AD individuals, 2. whether a multimodal biomarker index can be used to determine who is at greatest risk of accumulating tau in early pathological AD regions.

We have now clarified throughout the text that our work is aimed at application over short time spans common in early AD clinical trials. In particular the text now writes:

(Introduction)

"We test whether our modelling approach predicts longitudinal change in biomarkers (i.e. future tau accumulation) using baseline non-tau data over the short timeframes that are typical in clinical trials (i.e. 1-3 years) at asymptomatic and mildly impaired stages of AD. ... suggesting potential benefits of our multimodal biological stratification for the design of clinical trials that aim to reduce primary pathological tau spread at the earliest stages of AD."

(Discussion)

"Fourth, our linear modelling approach, focusing on data from cognitively normal and MCI individuals, captures the earliest changes in tau accumulation on a time-span typical of a clinical trial (i.e. only a few years). We show that a linear model predicts individual variation in future tau accumulation, within multiple samples and over the timeframe of a clinical trial (i.e. early to intermediate pathological stages)."

5. *d) Revised text on page 19: 'as evidenced by the generalisability of our results across samples'
- Please tone this down a bit. Or qualify it with a comment about (small) sample sizes. Perhaps: 'as supported by the consistency of our results across (admittedly small) samples.'*

We have revised the text and comment on the small sample size in the validation data set. In particular the text (Discussion) now writes

“This provides evidence that our results are not driven by the sampling characteristics of ADNI, suggesting generalisability (albeit in the small BACS sample) of our modelling approach to more diverse groups.”

6. *e) Revised text: 'Extending our modelling approach to integrate less-costly (i.e. plasma) and non-invasive (i.e. cognitive) data has strong potential for stratification at asymptomatic or early stages of AD.'
- I really like this idea. Is something like this feasible for the authors to implement in ADNI? Using cognitive data, not 'new plasma biomarkers' (since they don't exist in ADNI). If it *is* possible to compare how this model performs with different inputs (cognitive test scores versus neuroimaging+biomarkers), such a study would add a lot of value to the paper as an additional supplementary analysis.*

We have now run an additional set of experiments using cognitive data from the ADNI sample. Using the GMLVQ-Scalar projection model we trained our classifier to separate individuals who are Clinically Stable (n=99) vs. Clinically Declining (n=156) from the ADNI2/ GO sample. We used 4 cognitive test scores to build our model; ADAS Cog, MOCA Total, MMSE Total, RAVLT Total. All cognitive assessments were taken within one year of the baseline A β scan. Our model achieved a cross validated class balanced classification accuracy of 86% (determined using random resampling of test data). Next, we derived the scalar projection score for the ADNI 3 sample with longitudinal tau data available. Of the 115 individuals 110 (41 MCI, 69 CN) had cognitive assessments within one year of their baseline tau scan.

The model classified 44 individuals as Clinically Declining and 66 as Clinically Stable. Comparing the agreement between the classifier and clinical diagnosis (**Supplementary Results Table 1**) showed fair agreement between the clinical diagnoses and the machine learning derived classification Cohen's kappa = 0.3295, 95% CI [0.1461, 0.5129] z = 3.4388 p = 0.0006.

Next, we tested whether the classification of Clinically Declining vs. Clinically Stable using the cognitive scalar projection is sensitive to future tau accumulation. We observed that the cognitive model separates individuals who will accumulate tau in the future (**Supplementary Figure 4.**). Further, we observed that there is a low inter-class correlation coefficient for the average rate of tau accumulation for Clinically Declining vs Clinically Stable ($r=0.07$ [-0.26, 0.38], $F(35,36)= 1.1492$ $p=0.34$), suggesting specificity to future rates of tau accumulation using a stratification based on cognitive data.

Finally, we tested whether the scalar projection derived from modelling the cognitive data relates to individual variability in future rate of tau accumulation. Unlike the scalar projection derived from modelling biological data, we did not observe a relationship between the scalar projection derived from cognitive data and individual variability in future rate of tau accumulation within ROIs that were shown to significantly accumulate tau (**Supplementary Results Table 2.**). Taken together, we show that a binary stratification based on cognitive data is sensitive to future tau accumulation. However, the individualised scalar projection score derived from modelling cognitive data is not a sensitive metric for fine scale stratification based on future tau accumulation. Thus, our cognitive model predicts whether individuals will accumulate tau but does not determine who will accumulate tau at a slow, intermediate or rapid rate.

In particular the text now writes:

(Results)

“Predicting future tau accumulation based on cognitive data

*To investigate the predictive power of cognitive data we re-ran our classification experiments using data from multiple neuropsychiatric tests as input features (ADAS-Cog, MOCA Total, MMSE Total, RAVLT Total). This cognitive classification model reliably separated Clinically Stable vs. Clinically Declining (86% class balanced accuracy). Further, using the cognitive scalar projection derived in ADNI 3, we show that this prognostic index separates individuals who will accumulate tau in the future (**Supplementary Results: Cognitive Classification Model**). These results demonstrate that stratification based on future tau accumulation is possible using cognitive (non-biomarker) data. Next, we related the cognitive scalar projection to future regional tau accumulation within ROIs that were shown to significantly accumulate tau (**Supplementary Figure 5**). We did not observe a significant relationship between the cognitive scalar projection and individual variability in future tau accumulation in these ROIs (**Supplementary Results: Cognitive Classification Model**). Thus, our trajectory modelling based on cognitive data separates individuals who will accumulate tau in the future; yet, it is less sensitive in predicting individual variability in future regional tau accumulation.”*

(Discussion)

“Further, we demonstrate that model-derived stratification using either biological or cognitive data determines which individuals will accumulate tau in the future. Yet, modelling biological rather than cognitive data predicts individualised future rates of tau accumulation (i.e. whether an individual will accumulate tau slowly or rapidly). Extending our biomarker modelling approach to integrate less-costly (i.e. plasma) and non-invasive (i.e. cognitive) data has strong potential to determine the most cost-effective approach for stratification at asymptomatic or early stages of AD.”

(Supplementary Results)

“Cognitive Classification Model

Using the GMLVQ-Scalar projection model we trained our classifier to separate Clinically Stable (n=99) vs Clinically Declining (n=156) individuals from the ADNI2/ GO sample. We used 4 cognitive test scores as predictors to train our model: ADAS Cog, MOCA Total,

MMSE Total, RAVLT Total. All cognitive assessments were taken within one year of the baseline A β scan. Our model achieved a cross validated class balanced classification accuracy of 86% (determined using random resampling of test data). Next, we derived the scalar projection score for the ADNI 3 sample with longitudinal tau data available. Of the 115 individuals 110 (41 MCI, 69 CN) had cognitive assessments within one year of their baseline tau scan.

The model classified 44 individuals as Clinically Declining and 66 as Clinically Stable. When comparing the agreement between the classifier and clinical diagnosis (**Supplementary Results Table 1**) we observed a fair agreement between the clinical diagnoses and the machine learning derived classification Cohen's kappa = 0.3295, 95% CI [0.1461, 0.5129] z = 3.4388 p = 0.0006.

Next, we tested whether the classification of Clinically Declining vs Clinically Stable using the scalar projection derived from the cognitive model is sensitive to future tau accumulation. This analysis showed that the cognitive model separates individuals who will accumulate tau in the future (**Supplementary Figure 5, Supplementary Results Table 2**). Further, there was a low interclass correlation coefficient for average rate of tau accumulation between Clinically Declining and Clinically Stable individuals ($r=0.07$ [-0.26, 0.38], $F(35,36)= 1.1492$ $p=0.34$).

	Clinically Stable	Clinically Declining
CN	50	19
MCI	16	25

Supplementary Results Table 1. Clinician vs. cognitive classifier Confusion matrix. inter-rater reliability when diagnosing Clinically Declining based on biological predictors (i.e. scalar projection) or a clinical diagnosis based on syndromic definitions (i.e. CN or MCI).

Predicting future regional tau accumulation based on the cognitive scalar projection

To test if the scalar projection derived from modelling the cognitive data is related to individual variability in future rate of tau accumulation we ran multiple regression models within the regions that were shown to significantly accumulate tau (**Supplementary Figure 5, Supplementary Results Table 2**). Unlike the biological scalar projection derived from modelling biological data, the cognitive scalar projection derived from modelling cognitive data did not show a significant relationship to individual variability in future rate of tau accumulation within any of the ROIs that were shown to significantly accumulate tau (**Supplementary Results Table 3**). Although the binary stratification based on cognitive data predicts future tau accumulation, the individualised score captured by the scalar projection derived from modelling cognitive data is shown to be less sensitive for fine scale stratification based on future tau accumulation. Although our cognitive model predicts whether individuals will accumulate tau in the future, it does not determine whether an individual will accumulate tau at a slow, intermediate or rapid rate.”

People classified as Clinically Declining

People classified as Clinically Stable

Supplementary Figure 5. Regional future annualised rate of tau accumulation across the 36 Desikan Killiany ROIs. Classification of Clinically Declining vs Clinically Stable individuals using cognitive data from the ADNI 3 sample. a. Mean future annualised rate of tau accumulation for Clinically Declining. b. The regions in red are significantly predicted to ($p < 0.05$ uncorrected) accumulate tau for Clinically Declining individuals. c. Mean future annualised rate of tau accumulation for Clinically Stable (CN). d. The regions in red are significantly ($p < 0.05$ uncorrected) predicted to accumulate tau for Clinically Stable individuals.

Braak Stage	Region	Clinically Stable (n=44)			Clinically Declining (n=66)		
		mean accumulation (SUVr/Year)	Significantly Accumulating Tau		mean accumulation (SUVr/Year)	Significantly Accumulating Tau	
			t-stat	p-val		t-stat	p-val
1	ENTORHINAL	0.004	1.059	0.147	0.008	1.228	0.113
2	HIPPOCAMPUS	0.005	1.422	0.080	-0.006	-1.279	0.896
3	PARAHIPPOCAMPAL	0.003	0.908	0.184	0.006	1.180	0.122
	FUSIFORM	0.004	1.267	0.105	0.013	2.627	0.006
	LINGUAL	0.000	0.122	0.452	0.005	1.136	0.131
	AMYGDALA	0.004	0.994	0.162	0.004	0.714	0.240
4	MIDDLETEMPORAL	0.006	1.566	0.061	0.019	3.642	0.000
	CAUDALANTERIORCINGULATE	0.000	-0.149	0.559	0.000	-0.010	0.504
	ROSTRALANTERIORCINGULATE	-0.003	-1.327	0.905	-0.003	-0.643	0.738
	POSTERIORCINGULATE	0.001	0.408	0.342	0.007	2.207	0.016
	ISTHMUSCINGULATE	0.002	0.734	0.233	0.008	2.399	0.010
	INSULA	-0.001	-0.411	0.659	0.004	1.067	0.146
	INFERIORETEMPORAL	0.006	1.563	0.061	0.018	3.299	0.001
4	TEMPORALPOLE	-0.005	-1.083	0.858	-0.002	-0.373	0.644
5	SUPERIORFRONTAL	0.002	0.603	0.274	0.004	1.144	0.129
	LATERALORBITOFRONTAL	0.002	0.580	0.282	-0.001	-0.158	0.562
	MEDIALORBITOFRONTAL	-0.001	-0.307	0.620	-0.006	-1.645	0.946
	FRONTALPOLE	-0.001	-0.212	0.584	0.008	1.233	0.112
	CAUDALMIDDLEFRONTAL	0.001	0.373	0.355	0.012	3.004	0.002
	ROSTRALMIDDLEFRONTAL	0.001	0.326	0.373	0.005	1.231	0.113
	PARSOPERCULARIS	0.003	1.040	0.151	0.005	1.704	0.048
	PARSORBITALIS	0.003	0.564	0.287	0.004	0.816	0.210
	PARSTRIANGULARIS	0.002	0.488	0.314	0.004	1.095	0.140
	LATERALOCIPITAL	0.002	0.441	0.330	0.025	3.458	0.001
	SUPRAMARGINAL	0.001	0.462	0.323	0.011	2.810	0.004
	INFERIORPARIETAL	0.004	1.063	0.146	0.020	3.552	0.000
	SUPERIORETEMPORAL	-0.001	-0.201	0.580	0.007	1.924	0.030
	SUPERIORPARIETAL	0.002	0.415	0.340	0.018	3.482	0.001
	PRECUNEUS	0.002	0.768	0.223	0.008	2.436	0.010
	BANKSSTS	0.006	1.740	0.043	0.011	2.330	0.012
5	TRANSVERSETEMPORAL	-0.005	-1.457	0.925	-0.004	-0.936	0.823
6	PERICALCARINE	0.001	0.225	0.411	0.004	1.029	0.155
	POSTCENTRAL	-0.001	-0.445	0.671	0.007	1.669	0.051
	CUNEUS	-0.001	-0.247	0.597	0.012	2.282	0.014
	PRECENTRAL	0.000	-0.048	0.519	0.006	1.646	0.054
	6	PARACENTRAL	0.003	0.941	0.175	0.005	1.311
	Mean	0.001			0.007		

Supplementary Results Table 2. Regional future annualised rate of tau accumulation
Measures of future regional annualised rate of tau accumulation taken from the Desikan Killiany atlas for ADNI 3 individuals within the 6 Braak stages. The mean future annualised rate of tau accumulation and test statistics describing whether a region significantly

Braak Stage	Region	Beta Estimate (SUVR/Year)	t-stat	p-val
3	FUSIFORM	0.021	1.414	0.165
4	MIDDLETEMPORAL	0.019	1.005	0.321
	POSTERIORCINGULATE	-0.004	-0.340	0.736
	ISTHMUSCINGULATE	0.001	0.126	0.900
	INFERIORETEMPORAL	0.018	0.942	0.351
5	CAUDALMIDDLEFRONTAL	0.013	1.152	0.256
	PARSOPERCULARIS	0.002	0.218	0.829
	LATERALOCIPITAL	0.009	0.396	0.694
	SUPRAMARGINAL	0.007	0.531	0.598
	INFERIORPARIETAL	0.024	1.312	0.197
	SUPERIORETEMPORAL	-0.008	-0.608	0.547
	SUPERIORPARIETAL	0.015	0.923	0.361
	PRECUNEUS	0.002	0.178	0.859
	BANKSSTS	0.022	1.471	0.149
6	CUNEUS	-0.009	-0.513	0.611

Supplementary Results Table 3. Fitting individual variability in regional future annualised rate of tau accumulation Parameter estimates and associated statistics for the robust regression equations using the cognitive scalar projection to predict regional future tau accumulation for individuals from ADNI 3 classified as Clinically Declining.

7. *"We look forward to the opportunity to test our approaches in a clinical trial." - Have you approached Pharma to get your model included in future trials?*

We have established collaborations with Pharma to facilitate the translation of our machine learning tools into clinical trials. However this upcoming work is not in the remit of this paper.

8. a) *I understand now: you are directly comparing to Jack-2020. I think this paragraph needs to be rewritten to make this more obvious. For example, wording such as "A recent paper claimed that Aβ status alone...We found that..." would avoid the reader being confused (as I was) into thinking that you're saying, simply, that "multivariate is better than univariate". - Where you write "by 44%", the reader would appreciate the raw numbers also being included, e.g., "from 100 to 66".*

We have clarified this paragraph in the revised manuscript. In particular, we show that using the multimodal prognostic index vs. Aβ status alone reduces the sample size required to observe a clinically meaningful change in pathological tau accumulation by 44% (Clinically

Declining n=636, Aβ positive n=1139). This result offers a complimentary conclusion to a recent study by Jack and colleagues investigating predictors with the most independent utility in predicting future rate of tau accumulation (Jack et al., 2020). Jack et al concluded that when considering key AD biomarkers (i.e. APOE 4, Aβ and neurodegeneration) Aβ status alone is the optimal independent biomarker for stratification to predict future tau accumulation. Our machine learning approach captures predictive disease related covariance in biomarkers (via the metric tensor), demonstrating that there is a benefit in using multivariate predictors over Aβ status alone when stratifying for clinical trials targeting future tau accumulation. This provides evidence in support of using machine learning to model disease trajectories based on biomarkers of interactive pathophysiological processes in AD.

In particular the text now writes (Discussion):

“This result complements a recent study by Jack and colleagues investigating predictors with the most independent utility in predicting future rate of tau accumulation(Jack et al., 2020). Jack et al concluded that when considering key AD biomarkers (i.e. APOE 4, Aβ and neurodegeneration) Aβ status alone is the optimal independent biomarker for stratification to predict future tau accumulation. Our machine learning approach captures predictive disease related covariance in biomarkers (via the metric tensor), showing a clear benefit in using multivariate predictors over Aβ status alone when stratifying for clinical trials targeting future tau accumulation.”

9. *"Direct comparison of diverse machine learning approaches remains challenging as cross-validation methodology, sample sizes and sample heterogeneity have a significant effect on model performance metrics" - This sentence misses the point. Perhaps I was unclear when I suggested that you should compare your results with a benchmark model. For example, this could be SVM or logistic regression for the classification experiments. To make the comparison fair, you would ideally keep all other experimental design choices fixed: input features, cross-validation settings, etc.*

We thank the reviewer for clarifying this point. Following the reviewer’s suggestion, we compared the established GMLVQ classification framework with a linear SVM using MATLAB statistics and machine learning toolbox. We fixed all experimental design choices (i.e. targets, input features and cross validation settings). Comparing performance of the two low parameter linear classifiers showed the same classification performance in the ADNI2/GO training sample (**Response Figure 1**) (GMLVQ: Average Accuracy 88% SVM: Average Accuracy 88%, paired t-test across cross folds $t(798)=1.5$, $p=0.13$) with a 99.13% overlap in the predicted labels in the ADNI 3 sample. These results suggest that the two low parameter linear classifiers perform comparably in the same classification task.

The similar performance between algorithms is likely due to two main reasons. First, both classifiers are linear and low parameter as there are only three input features used to classify two classes. As the training sample size (n=256) far exceeds the free parameters of each model, neither approach is prone to overfitting. Second, the training classes used are specifically constructed to have the best chance in finding a robust decision boundary. That is, we used multiple clinical appraisals to determine if a training target was stable cognitively normal vs. cognitively normal or MCI at baseline but received a diagnosis of Dementia. Here,

we did not present the model with uncertain classes (i.e. MCI at baseline but cognitively normal or MCI at follow-up), increasing the likelihood of each classifier extracting a robust decision plane.

Despite these similarities between SVM and GMLVQ, GMLVQ has a clear advantage: it learns the metric tensor providing the subspace based on which individualised projection indices (scalar projection values) can be calculated.

Response Figure 1. Comparing classification performance of SVM and GMLVQ. Boxplot showing the distribution of class balanced classification accuracies across resampling for the SVM (left) and GMLVQ (right).

10. Thus, our predictions are not directly comparable to those of models performing only binary classifications based on syndromic diagnosis'' - To be blunt, this is nonsense. As explained above, and with apologies if I was unclear in my first review, your classifier experiments can be compared with SVM/etc. in place of your model.

Please see above for comparison of GMLVQ and SVM classifiers

11. Additionally, your prediction results could also be compared with latent-time mixed-effects models such as (Donohue et al., Alzheimers Dement 2014; Li et al., Stat

Meth Med Res 2017; Lorenzi et al., NeuroImage 2017), each of which has public source code available.

We thank the reviewer for this suggestion. We have now compared our prediction results with the latent time joint mixed effects model (LTJMM) introduced in Li et al. 2017 and applied in Li et al 2018(Li, Iddi, Thompson, & Donohue, 2019; Li et al., 2018). We used the public source code for this model from <https://bitbucket.org/mdonohue/ltjmm/src/master/> to compare our predictions.

First, we ran the LTJMM on the longitudinal FTP-PET measures in the ADNI 3 sample including as covariates APOE 4, MTL atrophy and FBP PET SUVR taken at the baseline FTP-PET scan. We modelled longitudinal FTP-PET accumulation in the 7 ROIs that showed significant regression fits between the scalar projection and future tau accumulation (**Figure 7**). We found that the mean posterior estimates of the rate of tau accumulation for the 7 ROIs are closely associated with the observed rate of tau accumulation (**Response Figure 2**).

Next, we extracted the latent time shift (delta) derived from the LTJMM to investigate if the model-derived disease stage is related to the scalar projection. We observed a significant relationship between the LTJMM latent time shift and the scalar projection $r(113)=0.42, p<0.0001$ (**Response Figure 3**). This demonstrates that the scalar projection derived from only *baseline* biomarker data is related to the LTJMM disease stage derived from both *baseline and longitudinal* biomarker information.

These analyses suggest that our prognostic index of disease severity (i.e. scalar projection derived from baseline data) relates to the latent time shift derived from the LTJMM. However, a key difference is that our scalar projection approach uses baseline non-tau data (i.e. APOE 4, medial temporal atrophy and A β) to make predictions of future tau accumulation, while the LTJMM model requires longitudinal FTP-PET data to derive a parameter estimate of rate of tau accumulation.

In particular, from the underlying LTJMM model (shown below) we see that the alpha 1 parameter representing individualised regional rates of tau accumulation is indexed by k - outcome and i- individuals. Therefore, this model requires the observation of longitudinal FTP-PET (k) to fit the rate parameter for each individual (i).

$$y_{ijk} = \mathbf{x}'_{ijk} \boldsymbol{\beta}_k + \gamma_k (t_{ijk} + \delta_i) + \alpha_{0ik} + \alpha_{1ik} t_{ijk} + \varepsilon_{ijk}$$

In contrast, our approach (shown below) derives this rate of future tau accumulation using *only baseline* non- tau biomarkers. That is, we utilise the scalar projection derived from baseline non-tau biomarker data to generate a prediction for the future rate of tau accumulation.

$$\text{Predicted Rate of Tau Accumulation (BACS)}_{ROI} = \beta(\text{ADNI 3})_{ROI} * \text{Clinically Declining Scalar Projection(BACS)} + \beta_0(\text{ADNI 3})_{ROI}$$

Thus, our scalar projection approach makes the following novel methodological contributions: First, we learn a linear subspace on the ADNI 2/GO sample that does not include FTP-PET imaging, showing that the learnt subspace relates to future tau

accumulation in two independent samples with longitudinal FTP-PET imaging. That is, the learnt subspace of baseline biomarkers that distinguishes Clinically Declining vs Clinically Stable allows us to identify individuals who will accumulate tau in the future.

Second, using this linear subspace and prototypical locations of Clinically Stable vs. Clinically Declining individuals we derive a single baseline index of non-tau biomarker severity out of sample. Using this index, we generate a series of predictive equations that are associated with individualised rates of regionally specific pathological tau accumulation.

Third, using the linear subspace -derived from ADNI 2/ GO- and the regional regression equations -derived from ADNI 3- we explicitly predict future rates of tau accumulation for asymptomatic individuals (i.e. BACS sample). To the best of our knowledge this is the first approach that is able to predict out of sample *individual rates of future regional tau accumulation* using *baseline non-tau biomarkers*. As LTJMM uses individualised parameters, out-of-sample predictions (i.e. training on one sample and testing on another) are not possible. Generalisation from ADNI 2/ GO and ADNI 3 to an asymptomatic sample (BACS) provides evidence that our modelling captures relevant patterns underlying future regional tau accumulation that are predictive when weaker signals are considered as in the case of asymptomatic individuals.

Thus, compared to the LTJMM, our approach has two key advantages a) it does not require longitudinal FTP-PET scans to derive the future rate of tau accumulation and b) it makes explicit predictions of future tau accumulation out-of-sample.

Response Figure 3. Comparing LTJMM model estimations for alpha (rate of tau accumulation) and observed rate of tau accumulation.

Response Figure 3. Comparing Latent time shift from LTJMM and GMLVQ-Scalar projection score.

12. *"To compare performance of our modelling approach we compared our multimodal baseline stratification with two common approaches to stratify individuals in the early stages of AD: 1) syndromic definitions and 2) amyloid status."*
- Seems a reasonable comparison, since your aim is to compete with these in clinical trial settings. However, if one can achieve comparable performance with (for example) a SVM (see my points above), then the technical contribution of this paper is diminished considerably. The authors are strongly encouraged to run these experiments to convince themselves and the reader of their model's contribution to knowledge.*

Our modelling approach is based on a well-established binary classification framework (GMLVQ) that is comparable to other linear classification approaches (e.g. SVM; see response to point 9). Yet, the technical contribution of our approach lies in deriving the scalar projection from the learned subspace that allows us to estimate individualised disease trajectories. That is, GMLVQ learns a metric tensor that zooms into a low dimensional subspace to separate the classes. This enables us to construct individual projection indices and to project these scalar values back onto the cortex to predict regional individualised rates of tau accumulation. These predictions can then be applied out-of-sample in the BACS asymptomatic sample, explaining up to 40% of the observed variance in temporal regions.

Importantly, our binary classification approach makes a strong conceptual contribution to the field, delivering 2 key results with potential impact for clinical trial design: 1. multimodal biological stratification outperforms unimodal stratification based on A β , and 2. syndromic definitions that constrain recruitment are not sensitive or specific to AD pathology. Thus, training a low parameter, simple linear classifier on longitudinal clinical labels predictions can be made out-of-sample across a range of different AD related changes (i.e. baseline and

future tau accumulation as well as cognitive decline). As these secondary predictions use data that are not included in the training of the classifier, our approach can be used to harmonise data that was collected using diagnostic criteria to make predictions in data samples with missing data (i.e. absence of FTP-PET in the ADNI2/GO training sample).

Finally, we show that the standard linear SVM package in MATLAB trained under the same conceptual framework (i.e. trained to predict longitudinal changes in diagnoses) performs similar to the GMLVQ classifier, within the ADNI2/GO training sample (GMLVQ: Average Accuracy 88% SVM: Average Accuracy 88%, paired t-test across cross folds $t(798)=1.5$, $p=0.13$) with a 99.13% overlap in the predicted labels in the ADNI 3 sample (see response to point 9). Thus our approach provides a tool that enables researchers to test implicitly learnt predictions out-of-sample by training simple linear and low parameter classifiers on longitudinal labels. The barrier to entry for researchers to follow our conceptual framework may be reduced, as other widely accessible linear classifiers (e.g. SVM) can be used to make the same predictions, increasing the potential impact of our approach.

To the best of our knowledge, no previous work has used baseline non-tau biomarkers to predict regional future rates of tau accumulation. Thus, our approach has both technical and conceptual merit as well as strong translational potential in the context of clinical trials.

In particular the text now writes:

(Results)

“Model comparison

We compared our model derived predictions generated from our GMLVQ-scalar projection approach to alternate prediction frameworks.

*First, comparing our GMLVQ binary classification of Clinically Stable vs. Clinically Declining to the standard linear Support Vector Machine (SVM) showed similar accuracy (mean accuracy GMLVQ: 88% SVM: 88%, $t(798)=1.5$, $p=0.13$) and 99.13% agreement in the predicted labels for the ADNI 3 sample (**Supplementary Results: GMLVQ vs. SVM classification**). Thus, the SVM classifier corroborates our binary classification results using GMLVQ. Yet, the main advantage of GMLVQ is that it learns the metric tensor, providing a subspace based on which the individualised prognostic index (i.e. scalar projection values) is calculated.*

*Second, we compared our trajectory modelling approach based on baseline biomarker data to latent time joint mixed effects models (LTJMM) that have been previously shown to infer disease stage based on longitudinal biomarker data (Li et al., 2019). We tested whether our scalar projection which incorporates only baseline pathological burden relates to disease stage (i.e. latent time shift) extracted from the LTJMM that is derived modelling longitudinal tau data. We observed a positive relationship between the LTJMM latent time shift and our prognostic index ($r(113)=0.42, p<0.0001$), suggesting the scalar projection derived using only baseline data relates to the LTJMM derived disease stage (**Supplementary Results GMLVQ-scalar projection vs. LTJMM prediction**).*

Taken together, our results show that a) our conceptual framework employing machine learning to utilise longitudinal changes in syndromic labels and predict biomarker changes is

corroborated across linear classifiers b) our prognostic index derived using baseline non-tau data, is related to disease stage estimated using longitudinal tau accumulation.”

(Discussion)

“Third, our modelling approach is based on linear subspace learning and makes continuous individual trajectory predictions using an adapted discrete classification framework (GMLVQ). Comparing our metric learning approach to other linear classifiers (i.e. SVM) corroborates our results, suggesting that low parameter machine learning algorithms trained on longitudinal diagnostic labels integrate baseline biomarker data to stratify early AD individuals (i.e. without requiring longitudinal diagnoses). Further, we show that our model derived prognostic index relates to disease stage, as determined using latent time joint mixed effects (LTJMM) models(Li et al., 2019). However, our approach has two main advantages compared to LTJMM: a) it derives future rate of tau accumulation based on non-tau biomarkers (in contrast to LTJMM that requires longitudinal FTP-PET scans) b) makes predictions of future tau accumulation out-of-sample at asymptomatic stages of disease. ... Our approach provides two key advances: a) it combines multimodal continuous biological measures to capture trajectories for individuals who may be on the threshold of unimodal biomarker positivity but likely to follow AD related trajectories(Landau et al., 2018), b) it harmonises longitudinal data collected using syndromic diagnostic criteria(Albert et al., 2011; McKhann et al., 2011) (e.g. ADNI(Petersen et al., 2010)) by means of a model-derived prognostic index. Further, our multimodal trajectory modelling approach has translational impact for clinical trial design compared to standard stratification approaches for AD based on a) baseline syndromic labels (i.e. CN and MCI) and, b) unimodal stratification by A β positivity. Our results propose that: a) determining treatment groups based on syndromic labels may result in variability in pathological state across groups, b) unimodal stratification based on A β alone is underpowered compared to the multimodal stratification derived based on our prognostic index.”

(Supplementary Results)

“Comparing binary and individualised predictions with alternate modelling approaches

GMLVQ vs. SVM classification

We compared the GMLVQ classification results with a linear Support Vector Machine (SVM). The SVM was run using the fitsvm.m function from MATLAB statistics and machine learning toolbox. We fixed all experimental design choices when comparing the two classifiers. That is, for each resample of the data we ran the two classifiers on the same training and hold out data and calculated the class-balanced accuracy on the hold out data. To compare the performance of the two linear classifiers we performed a paired t-test on the class-balanced accuracy across resampling. Comparing the average classification performance in the ADNI2/GO training sample we observed the same model accuracy (GMLVQ: Average Accuracy 88% SVM: Average Accuracy 88%). Further, we did not observe any significant differences in classification performance across resamplings ($t(798)=1.5$, $p=0.13$). Finally, we trained each model on the full ADNI2/GO sample and generated predicted outcome labels for the ADNI 3 sample. We observe a 99.13% agreement in predicted labels for the ADNI 3 sample. Therefore, we conclude that the two low parameter linear classifiers perform comparably in the same classification task. This is

possibly due to the fact that both classification approaches relate to linear subspace learning. First, both classifiers are linear and low parameter as there are only three input features used for separating two classes. As the training sample size ($n=256$) far exceeds the free parameters of each model, neither approach is prone to overfitting. Second, the training classes used were specifically constructed to have the best chance in finding a robust decision boundary. That is, we used multiple clinical appraisals to determine if a training target was stable cognitively normal vs. cognitively normal or MCI at baseline but received a diagnosis of Dementia. Here, we did not present the model with uncertain classes (i.e. MCI at baseline but cognitively normal or MCI at follow-up, increasing the likelihood of each classifier extracting a robust decision plane. Finally, both approaches can be related to subspace learning; that is, GMLVQ determines a basis set and the minimum distance to a prototype within this subspace, while SVM can be interpreted as representing the normal vector of the class separation hyperplane as a subspace of dimensionality one.

GMLVQ-scalar projection vs. LTJMM prediction

We compared the GMLVQ-scalar projection derived from baseline biomarker data with the model derived disease stage from the latent time joint mixed effects models (LTJMM) presented in Li et al. (Li et al., 2019). To run the LTJMM we used the public source code for this model from <https://bitbucket.org/mdonohue/ljmm/src/master/>.

First, we ran the LTJMM on the longitudinal FTP-PET measures in the ADNI 3 sample including as covariates APOE 4, MTL atrophy and FBP PET SUVR taken at the baseline FTP-PET scan. We modelled longitudinal FTP-PET accumulation in 7 ROIs (BANKSSTS, INFERIOR TEMPORAL, FUSIFORM, PRECUNEUS, INFERIOR PARIETAL, SUPRAMARGINAL, SUPERIOR PARIETAL). We found that the mean posterior estimates of the rate of tau accumulation for the 7 ROIs are closely associated with the observed rate of tau accumulation (mean $R^2=0.81$; min $R^2=0.61$; max $R^2=0.92$). We found that the mean posterior estimates of the rate of tau accumulation for the 7 ROIs were closely associated with the observed rate of tau accumulation. Next, we extracted the latent time shift (delta) derived from the LTJMM to investigate if the model derived disease stage relates to the scalar projection. We observed a significant relationship between the LTJMM latent time shift and the scalar projection $r(113)=0.42, p<0.0001$. This highlights that the scalar projection derived from only baseline biomarker data relates to the LTJMM disease stage derived from both baseline and longitudinal biomarker information.”

13. *"This allows us to draw conclusions that are relevant for patient stratification and the design of clinical trials...variability in pathological state across groups..." - Sorry to harp on about it, but your method handles temporal variability (disease stage/pathology severity) but doesn't handle spatial variability (different subtype patterns) in tau accumulation. See the four spatiotemporal subtypes of AD: Vogel-2020.*

Revised text: "...Recent evidence suggests that there are consistent patterns of tau spread..."

- Recent evidence suggests, quite strongly, that there are four spatiotemporal subtypes of tau accumulation: Vogel-2020 (as mentioned above). Using a much larger tau PET sample size covering early through to late AD, across multiple tau-PET tracers and multiple cohorts, Vogel et al. provided strong evidence for four spatiotemporal subtypes of tau accumulation in Alzheimer's.

We thank the reviewer for this suggestion in light of the recent Vogel paper. As the reviewer points out, our spatiotemporal predictions focus on one topography of longitudinal tau accumulation. Our intention is to model future rates of tau accumulation in asymptomatic or mildly symptomatic older adults in an age range typical of late onset AD, rather than account for clinical subtypes of AD or atypical AD variants.

The Vogel et al paper uses the SuStain modelling approach in a large sample of older adults from several large cohorts with FTP-PET to stage individuals in four distinct spatiotemporal subtypes of tau accumulation. It is likely that these spatiotemporal patterns are largely related to clinical syndromes that are most often expressed in earlier onset cases of AD and may not truly represent the earliest accumulation of tau in the asymptomatic and early phases of typical late onset AD. Interestingly, the regions that we showed to accumulate tau prospectively are included in the SuStain results. Despite the differences in the SuStain patterns between subtypes, many of these differences reflect relative involvement of a similar set of regions. The regions in our **Figure 5** overlap substantially with most of the subtype regions in Vogel et al Figure 1.

In contrast, our sample comprises a large number of asymptomatic and early AD participants. In the Vogel et al paper, the initial stage of modelling involves removing tau negative (i.e. S0) participants resulting in 78.5% of all cognitively normal individuals being removed from further subtyping. This reduced the total number of cognitively normal individuals to around a third of the remaining tau positive group, resulting in a sample that was biased towards fitting symptomatic patients. The retained asymptomatic individuals were predominately assigned to earlier SuStain stages. However, the authors show that these individuals (who are primarily asymptomatic or mildly impaired) have a significantly lower probability of falling into a specific subtype (with CN likelihood statistically lower than MCI and AD). With the authors concluding that *“In general, early stage individuals were assigned to subtypes with less confidence,... This provides some evidence that the earliest phases of each subtype may overlap”*.

Thus, the Vogel paper provides evidence for spatiotemporal profiles in predominantly clinical AD rather than asymptomatic and preclinical AD, as 1. the model uncovered spatiotemporal variability in a predominately symptomatic sample and 2. the majority of the asymptomatic sample couldn't be assigned with high probability to any given subtype. This limitation was discussed by the authors in the Vogel paper. *“It is noteworthy that we used a fairly conservative approach to identify “tau-positive” individuals, and that our subtyping was performed primarily on cognitively impaired individuals. It is possible that most variability occurs in later disease stages given that early stage individuals were not confidently assigned to a subtype”*

Further, when investigating the relative involvement in each subtype by cohort, the different spatiotemporal subtypes are less frequent and are expressed less strongly in the ADNI data particularly compared to UCSF and Biofinder data. It is possible that variability in the

assignment to a given subtype is driven by variation in clinical syndromes (i.e. Early onset AD, PCA, lvPPA), as the sampling characteristics of ADNI focus on typical late onset AD, and there is prevalence of atypical early onset AD variants within the UCSF sample. In the Vogel paper (extended data figure 4), the ADNI sample expressed subtype 4 (left temporal) to a limited extent and subtype 3 (posterior) to an even lesser extent which is consistent with the rare presentation of ADNI participants with focal neurobehavioral syndromes.

Finally, the Vogel paper investigates whether a previously published epidemic spreading model predicts the spreading pattern of tau. When investigating the primary seeding region for tau spread across the cortex in early stages (predominantly asymptomatic and early AD individuals), 3/4 subtypes had the same best fit seeding region (MTL). This suggests that the early stages of tau spread may overlap across most spatiotemporal subtypes, originating in temporal regions. A recent publication from the Harvard Aging Brain Study in a group of cognitively normal and mildly symptomatic individuals provides further evidence for a temporal lobe onset of tau pathology (Sanchez et al., 2021). As our investigations focus on early tau spread, our single spatiotemporal profile is not invalidated by the Vogel et al findings. Further our out-of-sample prediction in the asymptomatic BACS sample, provides additional evidence that our approach captures early stage seeding of tau. This enhances the clinical relevance of our approach as our predictions relate to early AD (i.e. prior to widespread tau propagation and progression to clinical AD subtypes) where interventions may be most effective.

To demonstrate that our results capture early AD accumulation of tau we present a figure adapted from three independent early AD samples (Jack et al., 2018; Pontecorvo et al., 2019; Schultz et al., 2018). We show that across these cohorts investigating the earliest changes in tau accumulation we observe the same (predictable) pattern of tau accumulation typical of early AD.

In both the ADNI 3 and BACS samples we uncovered regions that largely replicated findings from previous studies (**Response Figure 4**). Using baseline FTP as a proxy for disease severity, Pontecorvo et al. show that in earlier stages of AD increases in FTP were seen predominantly in the inferior lateral temporal cortex and in the posterior cingulate (Pontecorvo et al., 2019). Further, Schultz et al. show that preclinical AD individuals (amyloid positive cognitively unimpaired) accumulate tau in regions extending from the amygdala, banks of the superior temporal sulcus, entorhinal, fusiform, inferior parietal, inferior temporal cortex parahippocampal gyrus and precuneus (Schultz et al., 2018). This largely supports the definition comprising the ‘meta regions’ of tau accumulation described in Jack et al (Jack et al., 2018).

ADNI 3 variance explained of Scalar Projection vs. tau accumulation

BACS predicted vs. real tau accumulation

Response Figure 4. Regions that significantly accumulate tau in early AD across different cohorts. **a.** variance explained within our single topography. **b.** meta regions defined by Jack et al 2018. Regions comprising the early Alzheimer's disease change and temporal meta-region of interest are indicated in red, blue, aqua, magenta; regions in green are the late AD change meta regions. **c.** longitudinal tau accumulation patterns from Pontecorvo et al. 2019; the heat map is the mean voxel-wise change from baseline to 18 months for β -amyloid+ subjects with intermediate tau deposition at baseline. **d.** difference in tauopathy in cognitively unimpaired β -amyloid+ cohort compared to β -amyloid; taken from Schultz et al. 2018.

In particular the text now writes:

(Introduction)

“Further spatiotemporal patterns of tau are shown to be strongly linked to both future neurodegeneration and cognitive decline(Hanseeuw et al., 2019). A recent study proposes four distinct spatiotemporal profiles of tau burden in predominantly symptomatic AD(Vogel et al., 2021), proposing clinically meaningful topographies of tau burden. Further evidence in early AD (i.e. asymptomatic and mildly impaired) cohorts suggests converging patterns of primary tau seeding (measured in-vivo by longitudinal FTP-PET)(Jack et al., 2018; Pontecorvo et al., 2019; Sanchez et al., 2021; Schultz et al., 2018). These studies show that tau initially accumulates within the medial temporal cortex then spreads to the superior and medial regions of the parietal cortex prior to severe cognitive impairment(Jack et al., 2018; Pontecorvo et al., 2019; Sanchez et al., 2021; Schultz et al., 2018). ...

Finally, we demonstrate the efficacy of our stratification approach against baseline 1) syndromic diagnosis and 2) A β positivity, suggesting potential benefits of our multimodal biological stratification for the design of clinical trials that aim to reduce primary pathological tau spread at the earliest stages of AD.”

(Discussion)

“Using this prognostic index we showed that individuals classified as Clinically Declining will accumulate tau in a topography-specific manner that reflects the initial spreading of tau in early stage AD (i.e. prior to severe cognitive impairment)(Sanchez et al., 2021), accurately reproducing the topography reported in numerous independent cohorts corresponding to the proposed “meta-ROI” for tau quantitation(Jack et al., 2018; Pontecorvo et al., 2019; Schultz et al., 2018). “

Response to reviewer 2.

In the introduction it states "These clinical syndromic definitions have no discrete demarcations on cognitive scales" - do you mean that threshold scores on cognitive testing are not part of the criteria for diagnosis? If so, please make clearer. There are cut-offs on routine cognitive tests e.g. the MMSE, that are used to demarcate MCI and AD, so this statement as is does not hold.

We have now amended the introduction removing this statement. In particular the text (Introduction) now writes:

"However, these clinical syndromic definitions are neither specific(Nelson et al., 2011; Serrano-Pozo et al., 2014) nor sensitive(Murray et al., 2011; Ossenkoppele et al., 2015) to the underlying pathology of AD."

References:

- Jack, C. R., Wiste, H. J., Schwarz, C. G., Lowe, V. J., Senjem, M. L., Vemuri, P., ... Petersen, R. C. (2018). Longitudinal tau PET in ageing and Alzheimer's disease. *Brain*, *141*(5), 1517–1528. <https://doi.org/10.1093/brain/awy059>
- Li, D., Iddi, S., Thompson, W. K., & Donohue, M. C. (2019). Bayesian latent time joint mixed effect models for multicohort longitudinal data. *Statistical Methods in Medical Research*, *28*(3), 835–845. <https://doi.org/10.1177/0962280217737566>
- Li, D., Iddi, S., Thompson, W. K., Rafii, M. S., Aisen, P. S., & Donohue, M. C. (2018). Bayesian latent time joint mixed-effects model of progression in the Alzheimer's Disease Neuroimaging Initiative. *Alzheimer's and Dementia: Diagnosis, Assessment and Disease Monitoring*, *10*, 657–668. <https://doi.org/10.1016/j.dadm.2018.07.008>
- Mintun, M. A., Lo, A. C., Duggan Evans, C., Wessels, A. M., Ardayfio, P. A., Andersen, S. W., ... Skovronsky, D. M. (2021). Donanemab in Early Alzheimer's Disease. *New England Journal of Medicine*, *384*(18), 1691–1704. <https://doi.org/10.1056/NEJMoa2100708>
- Pontecorvo, M. J., Devous, M. D., Kennedy, I., Navitsky, M., Lu, M., Galante, N., ... Mintun, M. A. (2019). A multicentre longitudinal study of flortaucipir (18F) in normal ageing, mild cognitive impairment and Alzheimer's disease dementia. *Brain*, *142*(6), 1723–1735. <https://doi.org/10.1093/brain/awz090>
- Salloway, S., Farlow, M., McDade, E., Clifford, D. B., Wang, G., Llibre-Guerra, J. J., ... van Dyck, C. H. (2021). A trial of gantenerumab or solanezumab in dominantly inherited Alzheimer's disease. *Nature Medicine*, 1–10. <https://doi.org/10.1038/s41591-021-01369-8>
- Sanchez, J. S., Becker, J. A., Jacobs, H. I. L., Hanseeuw, B. J., Jiang, S., Schultz, A. P., ... Johnson, K. A. (2021). The cortical origin and initial spread of medial temporal tauopathy in Alzheimer's disease assessed with positron emission tomography. *Science Translational Medicine*, *13*(577), 655. <https://doi.org/10.1126/scitranslmed.abc0655>
- Schultz, S. A., Gordon, B. A., Mishra, S., Su, Y., Perrin, R. J., Cairns, N. J., ... Benzinger, T. L. S. (2018). Widespread distribution of tauopathy in preclinical Alzheimer's disease. *Neurobiology of Aging*, *72*, 177–185. <https://doi.org/10.1016/j.neurobiolaging.2018.08.022>

Reviewers' comments:

Reviewer #1 (Remarks to the Author):

I again commend the authors on their response to my criticisms, but there are remaining questions/concerns that I detail below. In short, I am not convinced of any considerable technical contribution because an SVM performed as well as GMLVQ (see below point 9 below, primarily). And I remain unconvinced of the application being convincing (see below points 2 and 1, primarily).

1. The authors may have missed my point slightly. Firstly, individuals having different tau burden can have the same rate of accumulation, which means that rate is not a suitable outcome measure (I'd be surprised if the FDA/EMA would consider rate alone). Secondly, the analysis presented in this response appears to be at the group level (Braak stages). My original comment was about the authors making it absolutely clear in their manuscript if they are predicting rate and/or burden. In particular, which (rate/burden) is relevant to the clinical trial application that the authors use to sell this work? I agree that classifying based on rate is potentially useful for enriching trials, but doubt it would be accepted as an outcome measure. Ultimately it will be tau burden itself used as an outcome measure (and likely only a secondary outcome for the time being, based on current FDA guidance), which means that rate/burden cannot be considered in isolation.

Action: modify the manuscript text accordingly, please. Multiple places. Including, e.g.,:

- "This pattern of increased baseline tau was also observed in the BACS Clinically Declining sample (Supplementary Figure 3)."

should read

"This group-level pattern of increased baseline tau was also observed in the BACS Clinically Declining sample (Supplementary Figure 3)."

- Likewise for text pertaining to rates as outcome measures, which I'm not convinced by as my response above hopefully makes clear.

- When referring to tau burden in group-level analyses the correct adjective is "higher" burden (in one group versus the other, e.g., in the Clinically Declining group), not "increased". These analyses (the first numbered response) did not involve assessing change, as far as I can tell. Might seem pedantic, but this is important.

2. Just FYI, my comments here about "translation to the clinic" were a bit vague — I include clinical trials in this. My query about the practical utility of the "slightly significant" (and uncorrected) trends in Figure 7 and especially in Figure 8 remains unanswered. How do these figures relate to the authors' rather grandiose claims about their model being predictive of tau accumulation, when it appears that a flat curve would fit the data almost as well, statistically speaking? What would the FDA/EMA make of these curves if the outcome measure (vertical axis) was to be proposed as a trial outcome? Not much, I suspect.

6. Thanks for running these extra experiments.

9. Thanks for running these extra experiments on an off-the-shelf classifier to benchmark performance. This is an often overlooked, but important task. Since your classifier performed almost identically to the SVM, this strongly suggests that your technical contribution (in terms of the method) is minimal.

- I disagree with the following claim made by the authors in their response: "Despite these similarities between SVM and GMLVQ, GMLVQ has a clear advantage: it learns the metric tensor providing the subspace based on which individualised projection indices (scalar projection values) can be calculated."

To me this sounds similar to the “a generative model is better than a discriminative model” argument, i.e., that classification performance is augmented by some additional contribution to knowledge. This is fine in principle, e.g., if your study aims to contribute to disease understanding. But here it is a direct comparison of two classifiers, with the key selling point (as presented by the authors in the manuscript) being the application of their model to clinical trials. If a trained SVM can be applied with almost identical performance, that detracts considerably from the study’s primary selling point.

The revised discussion of the manuscript should certainly downplay the importance/value/contribution of GMLVQ (which appears diminished based on these SVM results). Have I misunderstood? Is the “metric tensor” somehow super valuable?

See also my original comment (numbered as point 12 in the authors' response) where I said in my previous review that “the technical contribution of this paper is diminished considerably” if an SVM performs similarly to GMLVQ. Based on these new SVM experiments, I would say that the technical contribution of this manuscript has been diminished considerably. In conjunction with my comments above under point 2, I'd say that the overall contribution has been diminished considerably.

11. Thanks for running these extra experiments. However, to be a fair comparison, you should have trained the LTJMM using non-tau data and used it to predict the same outcomes as GMLVQ. Additionally, once trained, an LTJMM can use cross-sectional data to assign latent time and perform subsequent prediction of tau accumulation in a comparable way to the GMLVQ/SVM experiments.

13. Thanks for this detailed clarification.

Response to Reviewer 1

1. *I again commend the authors on their response to my criticisms, but there are remaining questions/concerns that I detail below. In short, I am not convinced of any considerable technical contribution because an SVM performed as well as GMLVQ (see below point 9 below, primarily).*

The key innovation of our work is deriving a prognostic index; that is, a continuous index that quantifies (in a suitable learnt metric) the distance of an individual from the Clinically Stable prototype. Our trajectory modelling approach extends beyond binary patient classifications that have poor sensitivity to baseline disease severity and carry risk of misdiagnosis. We show that our continuous index: a) predicts individualised rates of future tau accumulation from non-tau baseline data even before symptoms occur, b) re-stratifies populations at greatest risk of accumulating tau in the future with potential application for clinical trials (see **Figure 8**). The predictive power of our prognostic index is validated not only against an independent sample that was not used to construct the index (i.e. ADNI 3: Cognitively normal, MCI data), but also (without any modifications) on a completely separate independent data set from asymptomatic individuals (i.e. BACS).

To derive this prognostic index, we need to work in a feature subspace that captures possible signatures of future tau accumulation. Any machine learning classifier that supports this subspace learning could be used. We chose to use GMLVQ over SVM because it naturally provides class prototypes and a subspace endowed with an appropriate metric that allows us to derive the prognostic index using our scalar projection method. It is this prognostic index that provides the key technical contribution of our work; the classifier is simply used as a step in the derivation of our prognostic index. The similarity in performance between these binary classifiers is not surprising—it would be suspicious if it were otherwise—and does not limit the technical contribution of our work. On the contrary, it confirms reproducibility, a key challenge in machine learning applications.

We elaborate on this in our response to point 7 and have revised the manuscript (pg. 5) to clarify the novel technical contribution and innovation of our work.

2. *And I remain unconvinced of the application being convincing (see below points 2 and 1, primarily).*

In the revised manuscript (see Introduction pg. 4-6; Discussion pg. 19-21), we clarify that our modelling approach has potential application in clinical trial design; that is, stratifying patients for inclusion to clinical trials based on our prognostic index that relates to projected rates of tau change reducing heterogeneity and increasing statistical power. In particular, we demonstrate that our ML-derived prognostic index of AD allows us to a) identify individuals who are at greatest risk of accumulating tau in the future, b) reduce the sample size required to determine future tau accumulation. We propose that this prognostic index can be used for patient selection in clinical trials (see **Figure 8** and point 6 below). Should our modelling approach be applied to clinical trials it has the potential to impact drug discovery by a) reducing sample heterogeneity that hampers statistical power, b) targeting individuals at greatest risk who may benefit the most from clinical intervention c) decreasing the required sample size and resulting in more timely and cost-effective clinical trials.

3. 1. The authors may have missed my point slightly. Firstly, individuals having different tau burden can have the same rate of accumulation, which means that rate is not a suitable outcome measure (I'd be surprised if the FDA/EMA would consider rate alone). Secondly, the analysis presented in this response appears to be at the group level (Braak stages). My original comment was about the authors making it absolutely clear in their manuscript if they are predicting rate and/or burden. In particular, which (rate/burden) is relevant to the clinical trial application that the authors use to sell this work?

We agree that tau burden and accumulation rate can be dissociated. We demonstrate that our modelling approach allows us to stratify patients not only based on tau accumulation but also on baseline tau (i.e. tau burden) and future cognitive decline. In particular, we show that the Clinically Declining group has greater baseline tau and future rate of tau accumulation. Further, our results provide quantitative evidence for the advantages of predicting changes in tau accumulation over cognition (a primary outcome measure in AD trials). In particular, we show that over the time frame of a standard AD clinical trial (1-3 years) there is greater statistical power to detect a clinically meaningful change in tau accumulation vs. cognitive decline (PACC change n=917 vs. tau accumulation n=637). These results suggest that tau accumulation could be an attractive outcome measure for clinical trials in the earliest stages of AD.

Predicting biomarker outcomes is important for understanding disease pathophysiology. Further, the value of predicting rate of tau accumulation for clinical trials has been demonstrated by recent trials; for example, the Donanemab trial (Mintun et al., 2021) used change in tau over time as a secondary outcome measure. For more details please see our response to point 4. In the revised manuscript (Introduction pg. 4), we clarify why we chose to focus on individualised future rate of tau accumulation.

Finally, the reviewer appears to have misunderstood our analysis, suggesting that Braak staging was used as group-level outcome. In the analysis investigating baseline tau burden we contrasted baseline tau for Clinically Declining vs. Clinically Stable groups within selected brain regions that overlap with Braak stages. Our results for these AD relevant brain regions show that the Clinically Declining group is burdened with tau at baseline. We have now clarified this in the revised manuscript (pg. 9).

4. I agree that classifying based on rate is potentially useful for enriching trials, but doubt it would be accepted as an outcome measure. Ultimately it will be tau burden itself used as an outcome measure (and likely only a secondary outcome for the time being, based on current FDA guidance), which means that rate/burden cannot be considered in isolation.

The reviewer questions the value of predicting rate of tau accumulation as an outcome measure in clinical trials, as regulators (i.e. FDA/EMA) are interested in cognitive decline as primary outcome. Below, we clarify that our work predicting changes in biomarker (i.e. tau-based) outcomes from baseline data is highly timely for application in clinical trials for the following reasons.

First, earlier this summer the FDA approved aducanumab, the first drug to be approved in 20 years for the treatment of Alzheimer's disease, based on clinical trials showing its effect on a biomarker outcome (i.e. reduction in amyloid-beta). Second, the recent Donanemab trial

(Mintun et al., 2021) used tau-PET measurements to select individuals who had the highest likelihood of responding to an anti-amyloid therapy. Third, the same trial used change in tau-PET over time (i.e. rate of tau accumulation, as used in our modelling approach) as a secondary outcome measure, as increasing tau is thought to be causal in producing effects of amyloid on cognition.

On a more general note, we do not believe it is appropriate for potential FDA/EMA policies to serve as a benchmark for research studies. The reviewer appears to have misunderstood the clinical application of our findings: our results inform clinical trial design. In particular, our ML-derived prognostic index can be used to inform patient selection for clinical trials that use change (in tau or cognition) over time as outcome. In the revised manuscript (see section: *Potential application in clinical trial design; pg. 16-18*) we clarify the value of our modelling approach for application in clinical trials.

5. *Action: modify the manuscript text accordingly, please. Multiple places. Including, e.g.,:*

- *"This pattern of increased baseline tau was also observed in the BACS Clinically Declining sample (Supplementary Figure 3)."*

should read

"This group-level pattern of increased baseline tau was also observed in the BACS Clinically Declining sample (Supplementary Figure 3)."

- *Likewise for text pertaining to rates as outcome measures, which I'm not convinced by as my response above hopefully makes clear.*

- *When referring to tau burden in group-level analyses the correct adjective is "higher" burden (in one group versus the other, e.g., in the Clinically Declining group), not "increased". These analyses (the first numbered response) did not involve assessing change, as far as I can tell. Might seem pedantic, but this is important.*

We thank the reviewer and have revised the text following the reviewer's suggestions.

6. 2. *Just FYI, my comments here about "translation to the clinic" were a bit vague — I include clinical trials in this. My query about the practical utility of the "slightly significant" (and uncorrected) trends in Figure 7 and especially in Figure 8 remains unanswered. How do these figures relate to the authors' rather grandiose claims about their model being predictive of tau accumulation, when it appears that a flat curve would fit the data almost as well, statistically speaking? What would the FDA/EMA make of these curves if the outcome measure (vertical axis) was to be proposed as a trial outcome? Not much, I suspect.*

We assess the statistical significance of our results based on out-of-sample cross-validation including data from different cohorts (BACS, ADNI 3) than the training sample (ADNI 2/GO). This is a stringent and robust validation approach.

We have revised Figures 7 and 8 to focus on regional (rather than aggregate tau across areas) rate of tau accumulation. In particular, the new figures (Figures 6 and 7) show the

relationship between our prognostic index and rate of tau accumulation in Fusiform gyrus, a region that is known to be susceptible to early pathological tau deposition in AD.

For **Figure 6a**, shows a statistically significant relationship when corrected using Bonferroni correction (Beta=0.028, $t=3.425$, $p<0.05(\text{FWE})$, $R^2=21.1\%$). Further, as this regression line represents the least squares fit, a flat line would not fit the data equally well; the regression model learnt that the best fitting line has a positive slope (i.e. Beta=0.028).

Figure 7a demonstrates out of sample prediction of individualised rates of future tau accumulation that account for 41% of the variance in the BACS sample. We generated these predictions out of sample, training a model on a completely different sample (ADNI-2/GO) using different PET tracers and MRI field strengths. This out-of-sample prediction of future tau accumulation in asymptomatic individuals using non tau baseline predictors from a patient sample provides a striking validation and a novel result. These results validate our trajectory modelling approach and provide evidence that our prognostic index captures baseline heterogeneity in disease state that is predictive of future changes in biomarkers (i.e. tau accumulation).

We have now included a new figure (**Figure 8**) that demonstrates how the curve in figure 7a can be used for patient re-stratification in a clinical trial. Focussing on the fusiform gyrus our binary stratification of Stable vs. Clinically Declining shows improved statistical power (i.e. decrease in required sample size) when: a) stratifying based on multimodal data vs. amyloid alone ($n=598$ vs. $n=719$), b) predicting changes in tau vs. cognition ($n=598$ vs. $n=917$). Yet, there is still substantial heterogeneity within the Clinically Declining group. Extending our approach to trajectory modelling using a scalar projection method, we show that our prognostic index explains 21.1% of this heterogeneity. In particular, we show that a more stringent threshold (indicated by the dashed black vertical line) than the probabilistic threshold used in the binary classification (indicated by the solid black vertical line) allows us to: a) select individuals with increased rate of tau accumulation (rate of accumulation: 0.028 vs. 0.0136 SUVR/Year), b) reduce sample heterogeneity (Variance: 0.00079 vs. 0.0012), c) increase power to detect change, reducing required sample size ($n=93$ vs. $n=598$).

Finally, we clarify, that our results inform clinical trial design rather than the assessment of trial outcomes by regulators (see section: *Potential application in clinical trial design*). Our prognostic index of baseline pathological severity is of value for trial design for the following main reasons. First, using a continuous index (i.e. the scalar projection) that relates to future tau accumulation allows us to identify individuals who will have the greatest future rate of change and therefore the best chance for treatment effects. Second, increasing the power to detect change reduces the sample sizes required and the subsequent costs. Finally, matching samples based on baseline severity reduces the chances of making erroneous inferences in intervention trials. In particular, recent studies (e.g. DIAN-TU study into anti-amyloid interventions) suggest that selecting a narrower range of baseline disease severity may result in increased statistical power to observe treatment effects in dominantly inherited AD (Salloway et al., 2021). Further, sample heterogeneity in treatment and placebo groups can lead to incorrectly determining treatment efficacy. A recent simulation study demonstrates how unexplained heterogeneity in treatment and placebo groups may lead to erroneous conclusions in clinical trials (Jutten et al., 2021).

Figure 8. Potential application in clinical trial design. *a.* cortical maps show average rate of tau accumulation for individuals classified as Clinically Stable vs. Clinically Declining (see **Figure 4**). *b.* Relationship of the scalar projection with future rate of tau accumulation within the Fusiform gyrus (as shown in **Figure 6a**). The solid black vertical line indicates the probabilistic boundary used to perform the binary stratification, blue crosses indicate rate of tau accumulation for the clinically stable group, black circles indicate future rate of tau accumulation for the clinically declining group. Using our prognostic index (i.e. scalar projection) we show that re-stratifying to a more stringent threshold—as indicated by the dashed black vertical line—a new sample can be selected with higher future rates of tau accumulation and lower heterogeneity within the sample.

7. Thanks for running these extra experiments on an off-the-shelf classifier to benchmark performance. This is an often overlooked, but important task. Since your classifier performed almost identically to the SVM, this strongly suggests that your technical contribution (in terms of the method) is minimal. I disagree with the following claim made by the authors in their response: "Despite these similarities between SVM and GMLVQ, GMLVQ has a clear advantage: it learns the metric tensor providing the subspace based on which individualised projection indices (scalar projection values) can be calculated." To me this sounds similar to the "a generative model is better than a discriminative model" argument, i.e., that

classification performance is augmented by some additional contribution to knowledge. This is fine in principle, e.g., if your study aims to contribute to disease understanding. But here it is a direct comparison of two classifiers, with the key selling point (as presented by the authors in the manuscript) being the application of their model to clinical trials. If a trained SVM can be applied with almost identical performance, that detracts considerably from the study's primary selling point. The revised discussion of the manuscript should certainly downplay the importance/value/contribution of GMLVQ (which appears diminished based on these SVM results). Have I misunderstood? Is the "metric tensor" somehow super valuable? See also my original comment (numbered as point 12 in the authors' response) where I said in my previous review that "the technical contribution of this paper is diminished considerably" if an SVM performs similarly to GMLVQ. Based on these new SVM experiments, I would say that the technical contribution of this manuscript has been diminished considerably. In conjunction with my comments above under point 2, I'd say that the overall contribution has been diminished considerably.

The aim of our trajectory modelling approach is to derive a nuanced continuous representation of *individual-specific conditions*—expressed as an index that can be used to infer individualised risk of future tau accumulation from non-tau baseline data—rather than simply assign individuals to discrete categories (Clinically Stable vs. Clinically Declining). To construct this prognostic index, we need to work in a subspace of the feature space of non-tau data that captures possible signatures of future tau accumulation, that is, the subspace that supports distinguishing between Clinically Stable and Clinically Declining subjects. Further, we need the reference points of the two conditions (Clinically Stable and Clinically Declining), so that individuals can be assessed based on the degree to which they belong (in their subspace representations) to each of the conditions. That is, to construct our prognostic index we need: (1) a relevant low-dimensional subspace of the feature space, (2) representative prototypes of the two conditions (class prototypes), (3) a quantitative way of assessing the degree of similarity between individuals' feature representations and the class prototypes. Any machine learning methodology capable of providing these three key ingredients could be used to derive our prognostic index. We chose to employ GMLVQ because it naturally provides all the ingredients as fundamental aspects of the trained classification model: (i) learnt class prototypes as representatives of the two core conditions (Clinically Stable vs. Declining); (ii) learnt metric tensor providing both the relevant subspace and the metric (distance) for assessing closeness to the prototypes through our scalar projection method.

In the revised text (pg 12), we clarify that we use the GMLVQ classifier as a stepping-stone to construct our prognostic index. Of course, many classifiers exist and, when employed appropriately, have the potential to achieve similar classification performance. Thus, it is not surprising that SVM has similar performance to GMLVQ. It would be suspicious if it were otherwise! This similarity in SVM and GMLVQ performance simply confirms that our classifier works as expected. The novelty of our approach lies in constructing our prognostic index from the components provided by GMLVQ. We chose GMLVQ over SVM to derive our prognostic index because GMLVQ provides naturally what SVM cannot: class prototypes and subspace endowed with an appropriate metric. The fact that our prognostic index generalises across different samples (as shown by out-of-sample validation on: a) ADNI 3:

Cognitively normal, MCI data, b) asymptomatic individuals from BACS) is strong evidence that it captures patterns related to future tau accumulation.

8. *Thanks for running these extra experiments. However, to be a fair comparison, you should have trained the LTJMM using non-tau data and used it to predict the same outcomes as GMLVQ. Additionally, once trained, an LTJMM can use cross-sectional data to assign latent time and perform subsequent prediction of tau accumulation in a comparable way to the GMLVQ/SVM experiments.*

LTJMM requires longitudinal data to model disease trajectories and fit individualised parameters, limiting out-of-sample generalisation. A recent implementation of LTJMM in predicting AD progression includes the out-of-sample baseline data in the model training (Iddi et al., 2019). This relaxation of the notion of out-of-sample can limit the scope for clinical applications that necessitate predictions for new patient data. In contrast, our modelling approach derives these out-of-sample predictions from baseline data naturally, as we provide individualised indices and features rather than constructing individualised models. The reviewer's request for technical treatments that may alleviate shortcomings of the LTJMM is well beyond the scope of model comparison and our study.

References

- Iddi, S., Li, D., Aisen, P.S., Rafii, M.S., Thompson, W.K., Donohue, M.C., 2019. Predicting the course of Alzheimer's progression. *Brain Informatics* 2019 61 6, 1–18. <https://doi.org/10.1186/S40708-019-0099-0>
- Jutten, R.J., Sikkes, S.A.M., Flier, W.M. Van der, Scheltens, P., Visser, P.J., Tijms, B.M., Initiative, for the A.D.N., 2021. Finding Treatment Effects in Alzheimer Trials in the Face of Disease Progression Heterogeneity. *Neurology* 96, e2673–e2684. <https://doi.org/10.1212/WNL.0000000000012022>
- Mintun, M.A., Lo, A.C., Duggan Evans, C., Wessels, A.M., Ardayfio, P.A., Andersen, S.W., Shcherbinin, S., Sparks, J., Sims, J.R., Brys, M., Apostolova, L.G., Salloway, S.P., Skovronsky, D.M., 2021. Donanemab in Early Alzheimer's Disease. *N. Engl. J. Med.* 384, 1691–1704. <https://doi.org/10.1056/NEJMoa2100708>
- Salloway, S., Farlow, M., McDade, E., Clifford, D.B., Wang, G., Llibre-Guerra, J.J., Hitchcock, J.M., Mills, S.L., Santacruz, A.M., Aschenbrenner, A.J., Hassenstab, J., Benzinger, T.L.S., Gordon, B.A., Fagan, A.M., Coalier, K.A., Cruchaga, C., Goate, A.A., Perrin, R.J., Xiong, C., Li, Y., Morris, J.C., Snider, B.J., Mummery, C., Surti, G.M., Hannequin, D., Wallon, D., Berman, S.B., Lah, J.J., Jimenez-Velazquez, I.Z., Roberson, E.D., van Dyck, C.H., Honig, L.S., Sánchez-Valle, R., Brooks, W.S., Gauthier, S., Galasko, D.R., Masters, C.L., Brosch, J.R., Hsiung, G.-Y.R., Jayadev, S., Formaglio, M., Masellis, M., Clarnette, R., Pariente, J., Dubois, B., Pasquier, F., Jack, C.R., Koeppe, R., Snyder, P.J., Aisen, P.S., Thomas, R.G., Berry, S.M., Wendelberger, B.A., Andersen, S.W., Holdridge, K.C., Mintun, M.A., Yaari, R., Sims, J.R., Baudler, M., Delmar, P., Doody, R.S., Fontoura, P., Giacobino, C., Kerchner, G.A., Bateman, R.J., Formaglio, M., Mills, S.L., Pariente, J., van Dyck, C.H., 2021. A trial of gantenerumab or solanezumab in dominantly inherited Alzheimer's disease. *Nat. Med.*

1–10. <https://doi.org/10.1038/s41591-021-01369-8>

REVIEWERS' COMMENTS

Reviewer #1 (Remarks to the Author):

Thanks for all the clarifications and detailed explanations. I very much appreciate the authors' efforts here. And I hope they agree that my comments have led to an improved manuscript for all readers.

The clarifications make it much easier for the reader to get the key message: this study is motivated by prognostic enrichment of clinical trials, i.e., to identify early "progressors/accumulators" who are on an AD-biomarker trajectory, and to do this *_better than_* current methods. This focusses on dealing with temporal heterogeneity such as detecting sub-threshold AD-pathway individuals who would normally be excluded from a clinical trial due to being "amyloid-negative". The authors do this in a spatially-curated manner by also considering the locations/topography of pathology accumulation.

My concerns have been addressed and I have no further objections to publication, subject to the usual proof-reading to fix remaining typos and some minor changes below.

Minor changes:

- The main change required is to provide the reader with a comment on the clinical/practical meaningfulness for the result on re-stratifying the Clinically Declining sample: how meaningful is the size and difference in tau accumulation rates? (SUVR/year of 0.028 vs 0.014) It seems to me from Table 3 in Pontecorvo et al., Brain 2019 (ref 20 in the manuscript) that the scale and variability of tau accumulation in amyloid-positives is of the same order as these numbers. Please add a comment on the implications for clinical trials.

- Figure 2c: swap axes to match other subfigures, and indeed to match the title of 2c: APOE4 vs Scalar Projection (Scalar Projection should be on the horizontal axis).

Response to Reviewer 1

Thanks for all the clarifications and detailed explanations. I very much appreciate the authors' efforts here. And I hope they agree that my comments have led to an improved manuscript for all readers.

The clarifications make it much easier for the reader to get the key message: this study is motivated by prognostic enrichment of clinical trials, i.e., to identify early "progressors/accumulators" who are on an AD-biomarker trajectory, and to do this better than current methods. This focusses on dealing with temporal heterogeneity such as detecting sub-threshold AD-pathway individuals who would normally be excluded from a clinical trial due to being "amyloid-negative". The authors do this in a spatially-curated manner by also considering the locations/topography of pathology accumulation.

My concerns have been addressed and I have no further objections to publication, subject to the usual proof-reading to fix remaining typo's and some minor changes below.

We are pleased the reviewer is satisfied with our revision. We thank the reviewer for their constructive comments and helpful suggestions.

Minor changes:

1. The main change required is to provide the reader with a comment on the clinical/practical meaningfulness for the result on re-stratifying the Clinically Declining sample: how meaningful is the size and difference in tau accumulation rates? (SUVR/year of 0.028 vs 0.014) It seems to me from Table 3 in Pontecorvo et al., Brain 2019 (ref 20 in the manuscript) that the scale and variability of tau accumulation in amyloid-positives is of the same order as these numbers. Please add a comment on the implications for clinical trials.

We thank the reviewer for this suggestion. To clarify, re-stratifying the clinically declining population using model-derived prognostic index results in a) increased mean tau accumulation rates (0.028 vs. 0.014 SUVR/year), b) reduced variability of tau accumulation rates (Variance: 0.00079 vs. 0.0012). Increasing the mean and reducing the variability of tau accumulation rates results in increased statistical power to detect change in tau accumulation. This impacts the design of clinical trials, as the required sample size for detecting change in tau accumulation is substantially reduced (n=93 vs. n=598).

Direct comparison of tau accumulation values across studies is complicated by differences in the data processing pipelines and the selection of the reference region for estimating SUVR that impacts the SUVR scale and values. For example, the Pontecorvo et. al. Brain 2019 paper uses the PERSI reference region, which is derived using a parametric approach to select a subset of voxels for SUVR normalisation¹. Our study uses a reference region taken from eroded subcortical white matter regions as this region increases sensitivity to longitudinal change in early stages of Alzheimer's disease².

We clarify this point further in the Results section, focusing on an example region of interest. In particular, the text writes:

'Focussing on the fusiform gyrus as a potential intervention target region we show that a more stringent threshold (Figure 8, dashed black vertical line) than the probabilistic threshold used in the binary classification (Figure 8, solid black vertical line) allows us to a) select individuals with increased rate of tau accumulation (mean rate of accumulation: 0.028 vs. 0.0136 SUVR/Year), b) reduce sample heterogeneity (variance: 0.00079 vs. 0.0012), c) increase power to detect change in tau accumulation, reducing substantially the required sample size (n=93 vs. n=598). This more precise patient stratification has potential impact in impact clinical trial

design, by reducing heterogeneity in the treatment and placebo groups that has been shown to hamper statistical power in clinical trials³.

Also, the Discussion section writes:

'Using our prognostic index to select participants within a range of projected tau accumulation has potential to a) reduce sample heterogeneity that hampers statistical power, b) target individuals at greatest risk who may benefit the most from clinical intervention c) decrease the required sample, resulting in more timely and cost-effective clinical trial. Our modelling approach can be tailored to trade off sample size, cost (from subjects screened but rejected from inclusion), and generalisability for a sample with the highest probability of benefitting from treatment.'

2. Figure 2c: swap axes to match other subfigures, and indeed to match the title of 2c: APOE4 vs Scalar Projection (Scalar Projection should be on the horizontal axis).

We thank the reviewer for this suggestion. We have now updated this figure following the reviewer's suggestion.

References:

1. Southekal, S. *et al.* Flortaucipir F 18 Quantitation Using Parametric Estimation of Reference Signal Intensity. *J. Nucl. Med.* **59**, 944–951 (2018).
2. Baker, S. L., Harrison, T. M., Maass, A., Joie, R. La & Jagust, W. J. Effect of off-target binding on 18F-flortaucipir variability in healthy controls across the life span. *J. Nucl. Med.* **60**, 1444–1451 (2019).